# Exploring the relationship between sea ice and phytoplankton growth in the Weddell Gyre using satellite and Argo-float data

Clara Celestine Douglas[1,2], Nathan Briggs[2], Peter Brown[2], Graeme MacGilchrist[3,4], and Alberto Naveira Garabato[1]

[1]Ocean and Earth Science, University of Southampton, Southampton, UK
[2]Ocean BioGeosciences, National Oceanography Centre, Southampton, UK
[3]Atmospheric and Oceanic Science, Princeton University, Princeton, NJ, USA
[4]School of Earth and Environmental Sciences, University of St Andrews, St Andrews, UK

**Correspondence:** Clara Douglas (C.C.Douglas@soton.ac.uk)

**Abstract.** Some of the highest rates of primary production across the Southern Ocean occur in the seasonal ice zone (SIZ), making this a prominent area of importance for both local ecosystems and the global carbon cycle. There, the annual advance and retreat of ice impact light and nutrient availability, as well as the circulation and stratification, thereby imposing a dominant control on phytoplankton growth. In this study, the drivers of variability in phytoplankton growth between 2002-2020 in the Weddell Gyre SIZ were assessed using satellite net primary production (NPP) products alongside chlorophyll-a and particulate organic carbon (POC) data from autonomous biogeochemical floats. Although the highest daily rates of NPP are consistently observed in the continental shelf region (water depths shallower than 2000m), the open ocean region's larger size and longer ice-free season mean that it dominates biological carbon uptake within the Weddell Gyre, accounting for 93-96% of the basin's total annual NPP. Variability in the summer maximum ice-free area is the strongest predictor of inter-annual variability in total NPP across the Weddell Gyre, with greater ice-free area resulting in greater annual NPP, explaining nearly half of the variance ($R^2$=42%). In the shelf region, the return of sea-ice cover controls the end of the productive season. In the open ocean, however, both satellite NPP and float data show that a decline in NPP occurs before the end of the ice-free season ($\sim$ 80 to 130 days after sea-ice retreat). Evidence of concurrent increases in float-observed chlorophyll-a and POC suggest that, later in the summer season, additional factors such as micro-nutrient availability or top-down controls (e.g. grazing) could be limiting NPP. These results indicate that in a warmer and more ice-free Weddell Gyre, notwithstanding compensating changes in nutrient supply, NPP is likely to be enhanced only up to a certain limit of ice-free days.

## 1 Introduction

The seasonal ice zone (SIZ) in the Southern Ocean (SO) plays an important but poorly quantified role in the net uptake and sequestration of carbon into the oceans on a range of timescales (Sigman et al., 2010; Brown et al., 2015; Van Heuven et al., 2014; Bushinsky et al., 2019). In this region, some of the highest rates of primary production across the SO are observed (Arrigo et al., 2008). Consequently, biological activity is considered to be key in determining the region's net carbon sink via the biological carbon pump (Brown et al., 2015; MacGilchrist et al., 2019; Henley et al., 2020). The biological carbon pump

begins with the fixation of carbon through photosynthesis by phytoplankton, which lowers the levels of carbon dioxide in the surface ocean, driving uptake of carbon into the ocean from the atmosphere (Hauck et al., 2015). The fate of this fixed carbon

(recycled in the surface ocean by heterotrophs or exported to depth) determines the strength and sign of the SO carbon sink (Arteaga et al., 2019; Ducklow et al., 2018; Sigman et al., 2010; Boyd et al., 2019). Furthermore, primary production forms the base of a rich and efficient food web in the SIZ, with high productivity regions supporting areas of ecological significance (Hindell et al., 2020), and commercially important species including krill, toothfish and squid (Trebilco et al., 2020).

Sea-ice dynamics play a critical role in primary production in the SIZ by attenuating light and altering stratification, mixing,

and nutrient delivery (McGillicuddy et al., 2015; Gupta et al., 2020; Twelves et al., 2021). Many observational (e.g. Arteaga et al., 2020; Bisson and Cael, 2021; Hague and Vichi, 2021; von Berg et al., 2020; Giddy et al., 2023; Briggs et al., 2018; Uchida et al., 2019; McGillicuddy et al., 2015) and model-based (e.g. Schultz et al., 2021; Taylor et al., 2013; Briggs et al., 2018; Twelves et al., 2021; McGillicuddy et al., 2015; Gupta et al., 2020) studies have enhanced our understanding of these impacts on regional (SO-wide) scales and at small, local scales. However, our understanding of the spatiotemporal relationship between

sea ice and net primary production (NPP) on basin-scales is lacking. Climate models from Coupled Model Intercomparison Project Phase 5 and 6 (CMIP5 and CMIP6) project a decline in Antarctic sea-ice area and concentration as a response to anthropogenic climate change (Casagrande et al., 2023). However, low confidence in projections, due to the complexity of ocean-ice-atmosphere systems, means that exact estimates of decline are uncertain (Casagrande et al., 2023; Meredith et al., 2019). Therefore, in light of these observed and anticipated changes in the climate of the SIZ (Kumar et al., 2021; Ludescher

et al., 2019; Casagrande et al., 2023), the need for a deeper understanding of the relationship between sea ice and NPP is pressing, as changes in sea ice will have concomitant impacts on carbon uptake and ecosystem health. However, the crucial gaps in our understanding of the drivers of NPP in the SIZ mean that large uncertainty remains about the nature and extent of these changes (Campbell et al., 2019; Henley et al., 2020; Kim and Kim, 2021; Pinkerton et al., 2021; Séférian et al., 2020; Henson et al., 2022).

The Weddell Sea is one of a few regions of deep and bottom water formation, and it has largely been thought that the transportation of these water masses to depth was the principal pathway for vast quantities of carbon (taken up through the solubility and biological carbon pumps) to be sequestered from the SO (Jullion et al., 2014; Meredith et al., 2014; Van Heuven et al., 2014; Nissen et al., 2022). However, it has more recently been hypothesised that this may not be the case. Instead, the Weddell Gyre's net $CO_2$ sink may be driven by biological carbon uptake in the offshore central Gyre, where the accumulation

of respired carbon at mid-depths within local Circumpolar Deep Water, and circulation of this enriched water mass out of the Gyre, provide a significant pathway for carbon sequestration north of the Gyre (MacGilchrist et al., 2019). Many SO biological carbon pump studies have focused on coastal and shelf regions, such as highly productive polynyas (e.g. McGillicuddy et al., 2015; Arrigo et al., 2015). However, the offshore (open ocean) area of the Weddell Sea has been shown to contribute greatly to the region's annual primary production, with annual NPP in the offshore (open ocean) marginal ice zone (73.7 g C $m^{-2}$ $a^{-1}$)

exceeding that seen in the shelf marginal ice zone (65.2 g C $m^{-2}$ $a^{-1}$), and almost matching the annual NPP exhibited in the shelf region (77.0 g C $m^{-2}$ $a^{-1}$) (Arrigo et al., 2008). In this way, the Weddell Sea marginal ice zone was identified as the largest and most productive marginal ice zone of all the geographic sectors investigated (25% greater than the Ross Sea marginal ice

zone; Arrigo et al., 2008). As such, the interaction between biology and circulation in the off-shore area of this SO region is particularly important, and it is vital to advance understanding of what drives these processes now to improve prediction of how they may change in the future.

Given the importance of biological carbon uptake in the Weddell Gyre for local ecosystems and the global carbon cycle, this paper characterises and quantifies the basin-scale relationship between sea ice and phytoplankton growth using both satellite net primary production (NPP) products and *in situ* observations of chlorophyll-a (Chl-a) and particulate organic carbon (POC) from Biogeochemical (BGC) Argo floats. In Sect. 2 we describe our data and methods as well as highlight and quantify uncertainties associated with our approach. In Sect. 3 we present a multi-year perspective of NPP in the Weddell Gyre, as well as its variability, before discussing the importance of sea ice and nutrient availability in driving inter-annual variability in NPP in Sect. 4.

## 2  Data and Methods

### 2.1  Study Area

The Weddell Gyre is a cyclonic gyre located in the Atlantic sector of the SO (Fig. 1; Vernet et al., 2019). Water primarily enters the gyre from the east (Circumpolar Deep Water supplied from the Antarctic Circumpolar Current) and leaves toward the north, with the Gyre's circulation and extent determined by wind forcing and topography (Hoppema, 2004; MacGilchrist et al., 2019; Vernet et al., 2019). Sea ice extends across almost the entire Weddell Gyre in the winter and retreats in the summer, encompassing the basin within the SIZ (Fig. 1; Arrigo et al., 2008; Vernet et al., 2019).

The extent of the study area was defined to align with previous studies of the Weddell Gyre (Akhoudas et al., 2021; Jullion et al., 2014; Brown et al., 2014, 2015; MacGilchrist et al., 2019). The region is bounded by the Antarctic continent to the south and west and two hydrographic transects to the north and east: the Antarctic Deep Water Rate of Export (ANDREX) cruises, and the CLIVAR quasi-meridional I6S section at 30 °E (Fig. 1; Bacon and Jullion, 2009; Meredith, 2010; Speer and Dittmar, 2008). For the purposes of this study, the Weddell Gyre was divided into two subregions: a shelf region (defined as the area with bathymetric depth less than 2000m) and an open-ocean region (depths greater than 2000m). The position of the 2000m isobath was extracted from TerrainBase, a 5-minute resolution global elevation database (National Geophysical Data Center/NESDIS/NOAA/U.S. Department of Commerce., 1995).

### 2.2  Satellite Data

#### 2.2.1  Sea Ice

Daily satellite sea ice concentrations over 2002-2020 were taken from the NOAA/National Snow and Ice Data center (NSIDC) Climate Data Record (Version 4, Meier et al., 2021). The data were obtained on the NSIDC polar stereographic 25 x 25 km grid. Eight-day means of sea-ice concentration (SIC) were calculated to match the NPP and Chl-a data temporal resolution. Pixels were defined as ice-free if the SIC was less than 15% (Windnagel et al., 2021). Annual statistics were calculated over

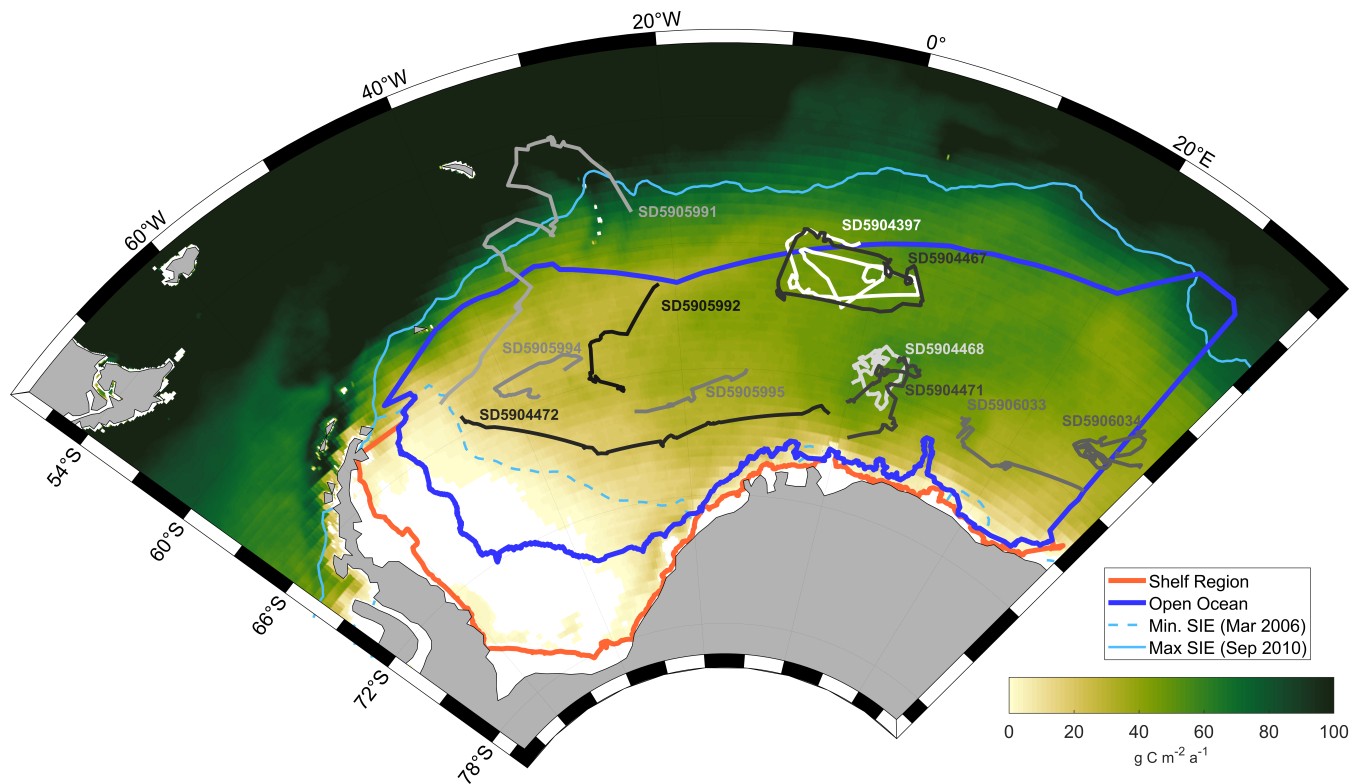

**Figure 1.** Location of the Weddell Gyre and study subregions (dark blue: open ocean; orange: shelf). The 18-year mean area-normalised annual NPP (g C m$^{-2}$) climatology derived from MODIS-Aqua satellite measurements using the Carbon, Absorption and Fluorescence Euphotic-resolving (CAFE) model (Silsbe et al., 2016) is represented by the yellow to green colourmap. White areas represent no data (permanent sea ice present). SOCCOM float trajectories from 11 BGC-Argo floats are shown in greyscale and labelled with WMO ID. Profiles from float SD5905991 located north of the study region were not included in the analysis. The dashed and solid light blue lines denote the summer minimum (March 2006) and winter maximum (September 2010) sea ice extent respectively.

austral years 2003-2020, starting July 2002 and ending June 2020. The sea-ice area (SIA) was calculated by multiplying the
90  SIC within each pixel by its area. Ice-free area (IFA) was calculated as the difference between the total area of the Weddell
Gyre (and sub-regions) and the SIA.

### 2.2.2 NPP and Chlorophyll-a

Satellite-derived NPP products calculated from MODIS (Moderate-Resolution Imaging Spectroradiometer)-Aqua chlorophyll-
a (Chl-a) data were obtained from the Ocean Productivity website (www.science.oregonstate.edu/ocean.productivity/ ; Oregon
95  State University, 2019). We use the NPP product derived using the Carbon, Absorption and Fluorescence Euphotic-resolving
(CAFE) model (Silsbe et al., 2016; Westberry and Behrenfeld, 2013). The CAFE model is an absorption-based model that con-
siders the amount of energy absorbed by phytoplankton, photoacclimation, and the efficiency with which energy is converted

into biomass. CAFE was chosen over other NPP models as it is currently the most comprehensive open-source NPP model and is the most robust in capturing the baseline productivity of the SO (Silsbe et al., 2016; Ryan-keogh et al., 2023; Westberry et al., 2023). The CAFE NPP product is calculated using MODIS-derived products: Chl-a, PAR (photosynthetically available radiation), sea surface temperature; MODIS-derived absorption and backscattering variables configured using the Generalized Inherent Optical Property (GIOP) model; and mixed layer depth provided by the HYbrid Coordinate Ocean Model (HYCOM). Input data to the CAFE model are cloud-filled via spatial and temporal interpolation prior to calculating NPP. The resolution of the NPP data is 8-day averages on a 1/12° latitude-longitude grid (2-4km zonal resolution, 8-10km meridional resolution in this region) for the period 2002-2020. Cloud-filled MODIS-Aqua Chl-a concentration data were also obtained at the same resolution as the NPP data to later compare to BGC-Argo float data as proxies for growth.

Three other NPP models were analysed to assess the robustness of the CAFE results. The Carbon-based Productivity Model (CbPM; Westberry et al., 2008; Behrenfeld et al., 2005) relates NPP to phytoplankton carbon biomass (derived from particulate backscatter received by MODIS-Aqua). Both the Vertically Generalized Production Model (VGPM; Behrenfeld and Falkowski, 1997) and Eppley-adjusted VGPM (eppley; Eppley, 1972) products are calculated based on the relationship between chlorophyll and temperature derived growth rates. Eppley differs from VGPM by parameterising the relationship between temperature and phytoplankton growth rate using the positive exponential function described by Eppley (1972); Morel (1991).

In our analysis, we derive a number of quantities from the basic NPP variable provided; *i.e.* the area-normalized, time-mean NPP at each lat-lon ($x$-$y$) pixel, for each 8-day period, which we label $\mu \equiv \mu(x,y,t)$. The spatially-averaged NPP for each subregion, $i$, is defined thus:

$$\overline{\mu}^{x,y}(i,t) = \frac{1}{\mathcal{A}(i,t)} \int\limits_{\mathcal{A}(i,t)} \mu(x,y,t)\, dA \tag{1}$$

where $dA \equiv dxdy$ is the area increment, and $\mathcal{A}(i,t)$ corresponds to the areal extent of the visible ice-free area in each subregion $i$ at time $t$ (as ascertained from data availability in the NPP product). Note that this area is not exactly equivalent to the IFA derived from the sea-ice data because of the "adjacency effect" of the ice edge on restricting ocean colour data near sea ice (Pope et al., 2017) as well as areas where there are data missing for other reasons (cloud-fill algorithm, input data: See Section 2.4). The total Weddell Gyre region corresponds to the area $A(t) = \sum_i \mathcal{A}(i,t)$. The annual means of the spatially-averaged NPP ($\overline{\mu}^{x,y}$) are derived over the period of the austral year starting in July ($t_s$) and ending in June of the following calendar year ($t_e$).

The "total annual NPP" ($N$) corresponds to the spatially and temporally integrated carbon uptake over each subregion:

$$N(i,\tilde{t}) = \int\limits_{t_s(\tilde{t})}^{t_e(\tilde{t})} \int\limits_{\mathcal{A}(i,t)} \mu(x,y,t)\, dA\, dt \tag{2}$$

where $\tilde{t}$ is a yearly time increment, defined over the austral year. Although we include the time-dependence of the areal extent here for consistency, it is irrelevant in this calculation because ice-covered pixels (where $\mu = 0$) do not contribute to the area integral.

Ocean-colour satellite observations are restricted by the presence of sea ice and when the solar angle/zenith is too low (below
20°, see Sect. 2.4 for more details). The number of days each pixel was visible to the ocean-colour satellite (and therefore had
data recorded) was counted. For the most part, satellite-visible days are approximately equivalent to the number of days from
sea-ice melt/retreat to when the noontime solar zenith angle decreases to less than 20°and restricts ocean colour measurements.

## 2.3 Autonomous floats

We use BGC-Argo float data to evaluate the data recovery attributes of satellite data, estimate associated uncertainties (Section
2.4), and also to assess the seasonal progression of phytoplankton growth in the water column, using Chl-a as a proxy for
photosynthetic potential and particulate organic carbon (POC) as a proxy for biomass. Eleven autonomous BGC-Argo floats
deployed by the Southern Ocean Carbon and Climate Observations and Modelling (SOCCOM) project from 2014 onwards
observed at least one complete annual cycle (from July to June) within the study region (Fig. 1). The floats were programmed
to profile from 1700-2000m to the surface on 10-day intervals. Chl-a data for these floats were obtained from the 21 December
2021 SOCCOM snapshot (https://doi.org/10.6075/J00R9PJW) and interpolated over a 5m vertical grid. SOCCOM provide two
Chl-a data products which differ in their fluorescence to Chl-a calibration. The corrected Chl-a product ($chl\_a\_corr$), which
has a Southern Ocean specific correction applied (Johnson et al., 2017), was used here. Missing surface/shallow Chl-a values
were extrapolated (nearest neighbor) from the shallowest data available for each profile (Appendix Fig.A1).

Mean and depth-integrated estimates of Chl-a were calculated from float profiles by integrating binned Chl-a concentrations
between the surface to 200m and surface to 20m, with the mean estimates for the 0-20m bin intended to line-up with the
approximate optical depth of MODIS-Aqua - i.e. what the satellites likely measured (Fig. 2; Gordon and McCluney, 1975).
Under-ice profiles were identified based on criteria used in the ice avoidance algorithm, which prevents a float ascending to the
surface if the median temperature between 20-50m is less than -1.78 °C (Bisson and Cael, 2021; Klatt et al., 2007).

The seasonal cycles of float-observed Chl-a were assessed alongside the timings of sea-ice retreat, sea-ice return, and the
date when the noontime solar zenith angle drops below 20°, restricting ocean-colour satellite observation, to get a small-scale
(localised) depth-resolved perspective on the drivers of seasonal and annual Chl-a variability. The dates that floats emerged
from under ice and returned to under-ice conditions were determined and the length of the ice-covered and ice-free seasons
were calculated. The average latitude of each float in March was calculated, and based on this location, the date that the solar
angle/zenith would be too low for MODIS-Aqua satellite coverage was determined.

POC concentrations were estimated from optical backscattering data after the removal of spikes due to large particles fol-
lowing (Briggs et al., 2011). "De-spiked" backscattering were averaged in 10 m bins in the upper 50 m and then at 50 m
intervals to 200 m. As with the Chl-a data, missing surface/shallow backscatter values were extrapolated (nearest neighbor)
from the shallowest data available for each profile. Backscatter was converted to POC concentrations using the conversion
co-efficient 3.12 x $10^4$ as proposed in Johnson et al. (2017). The mean and depth-integrated POC in the 0-20m and 0-200m
bins are reported here.

## 2.4  Uncertainty estimates for satellite chlorophyll-a and NPP

Satellite estimates of annual NPP should be considered conservative due to the limitations of ocean-colour satellites and NPP products. We divide these potential negative biases in satellite-derived annual NPP into three categories: 1) sea-ice coverage; 2) low solar angle; and 3) other data gaps. The first two we assess using BGC Argo float data. The third we assess and attempt to correct using satellite-derived sea-ice coverage data. Sea ice restricts ocean-colour satellites from viewing any production taking place under and within sea ice (Arrigo and van Dijken, 2004; Peck et al., 2010; Pope et al., 2017). Spatial coverage of the ocean-colour satellite is also increasingly restricted from early March, due to the decreasing solar angle limiting optical view (Pope et al., 2017). This means that much of the surface ocean is not observed by the MODIS-Aqua satellite at the end of the summer period despite much of the region still being ice-free. It is therefore likely that some NPP is being missed at the end of the growing season. Finally, satellite NPP data contain additional gaps, due to a range of optical limitations, including ice adjacency cloud cover.

Without comprehensive *in situ* measurements of NPP, it is not possible to directly quantify what is missed by satellites. However, BGC-Argo float data can be used to provide insight into these limitations, as they provide sub-surface, year-round, under-ice observations of key parameters related to NPP. Floats data have their own limitations - floats are Lagrangian autonomous observing platforms, so observations reflect both temporal and spatial variability. Additionally, sensor calibrations may vary and sensors sometimes drift towards the end of the float deployment. We did not attempt to estimate water-column integrated NPP from float data, as the floats in the study region lacked PAR (Photosynthetically Active Radiation) sensors, and, as far as we are aware, there are not yet methods for calculating NPP from float data that have been robustly validated for widespread use. Instead, estimates of Chl-a and POC can be used as conservative proxies for phytoplankton growth/biomass to assess the importance of missed phytoplankton. While Chl-a does not directly equate to NPP, satellite Chl-a is used in all satellite NPP products, and as such we can compare float and satellite Chl-a observations. Figure 2 highlights the similarities and dissimilarities in float and satellite Chl-a estimates. The absolute values of Chl-a differ between the satellite and the float because of the way Chl-a is derived from observations on these platforms - satellite Chl-a is derived from reflectance, whereas fluorescence is used to calculate float-observed Chl-a. Nonetheless, the relative changes in Chl-a over the timeseries should be seen in both datasets. In most cases, the satellite Chl-a at each float location peaks and troughs at the same time as the float Chl-a. On some occasions, the spring peak/bloom is not fully seen by the satellites, and in several cases (Figure 2: b) and c) in 2018, f) and k) in 2020), it is missed entirely. As such, the addition of float data in this study expands on the results derived from the satellite products and allows assessment of the seasonal patterns in Chl-a. Notably, it is clear that the satellite Chl-a product does not span the entire ice-free/growing season and the floats are valuable in providing a year-round perspective on the seasonal changes in Chl-a and POC.

We use float data to determine how much of the annual-integrated Chl-a and POC (in the surface 20m) is missed by satellites, reasoning that this offers a reasonable estimate for the fraction of missed NPP. The proportions of annually integrated Chl-a and POC measured by each float observed when the float was (1) under sea ice (using the Klatt et al. (2007) temperature criterion mentioned above) and (2) when the solar angle is less than 20°, limiting satellite view at the end of summer (from

mid-March) are summarised in Table 1. At the float locations, a small proportion of annually integrated Chl-a is present under sea ice (median 10%, range 2-23%), while up to 19% (median 9%) of annually integrated Chl-a is observed by floats during the time when the ocean-colour satellite was unable to view the surface ocean due to low solar angle (Table 1). Up to 30% of the annually integrated POC is recorded under-ice (minimum 7%, median 19%) and an average of 12% (range 4-20%) of the surface POC is recorded after the solar angle is too low.

Ocean-colour satellites are only able to view the surface of the ocean, meaning Chl-a satellite products do not represent the full euphotic zone/water column inventory (Pope et al., 2017). The CAFE algorithm applies a light saturation model for the mixed layer and assumes a co-varying relationship between the phytoplankton absorption coefficient and light saturation below the mixed layer in order to estimate the vertical structure of Chl-a (and NPP; Silsbe et al., 2016). As a result, the subsurface Chl-a inventory may not be accurately resolved and may also miss subsurface Chl-a maximum layers, which contribute substantially
to productivity and promote enhanced carbon export in the SO (Baldry et al., 2020). Again, floats enable assessment of the importance of phytoplankton missed by satellites, in this case, below 20m. The depth integrated Chl-a signal seen in Figure 2 declines later than the surface Chl-a, indicating that concentrations of Chl-a remain elevated at depth for longer than at the surface.

    In addition to missing satellite NPP data under ice and at low solar zenith angle, there are further empty grid points in the
mapped NPP and Chl-a data that mean that the IFA does not equate to the area of the NPP or Chl-a products where data is recorded. NPP in these areas is considered to be zero, leading to an underestimate of the regions' total NPP. Some of these gaps could be attributed to the "adjacency effect" along the ice edge and also where the cloud-fill algorithm did not complete successfully. In addition to this, there is also a disparity in the spatial coverage of the NPP products and the Chl-a input data used to derive CAFE NPP (Table 1). Some of the input data (absorption due to gelbstoff and detritus, absorption due
to phytoplankton and backscatter spectral slope parameter) used in the CAFE algorithm to derive NPP have less extensive spatial coverage than the Chl-a input data. This means that there are some areas in the NPP product that imply there is no NPP occurring despite Chl-a being observed by the satellite. Over the timeseries, between 10-100% (mean 78%, median 85%) of the area with Chl-a data also has coverage in the CAFE NPP product (lowest values occur as the satellite view declines at the start of winter). The spatial coverage of the CAFE product in relation to IFA ranges between 0-94% over the seasonal cycle.
When ice-free and with sufficient solar elevation for ocean-colour observations, on average 47% of the IFA was visible in the CAFE product. This difference between the IFA and area where NPP is observed in the CAFE product is most acute in the shelf region, where 0-65% (mean 10%) of IFA has NPP data when conditions are suitable for ocean-colour observations, and many coastal polynyas that are visible in the input Chl-a data are not translated into the NPP data. Therefore shelf integrated NPP values are expected to be significantly underestimated. To estimate the potential total NPP occurring across the entire IFA in
both the shelf and open ocean regions, data gaps were filled with the mean daily rate for each 8-day time-step within a region. As with the original calculation, the total annual NPP is then calculated by integrating the new 8-day totals for the austral year. The difference between the directly observed (raw) and imputed (gap-filled) estimates can be seen in Figure 3.b-d).

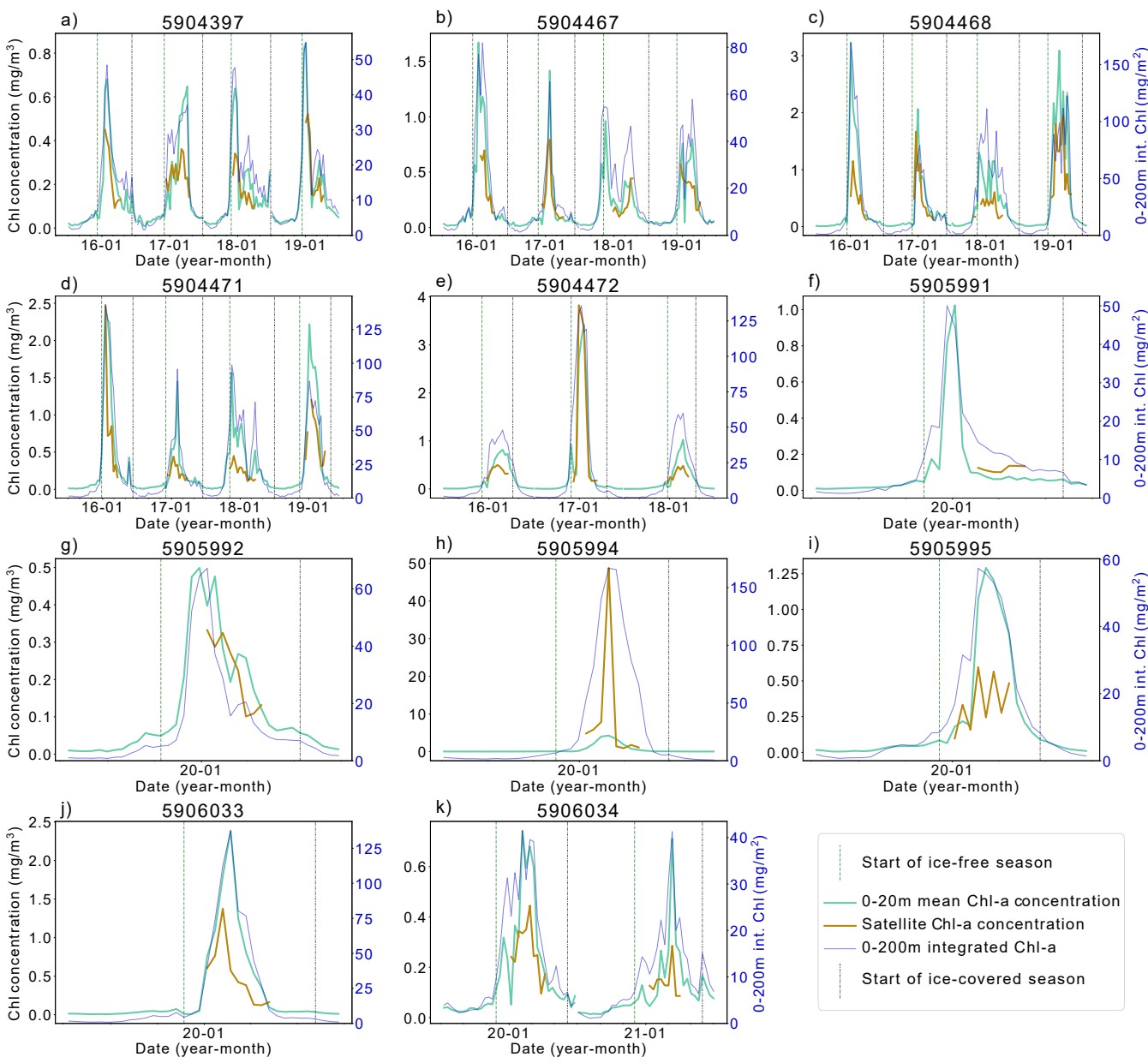

**Figure 2.** Satellite - BGC-Argo float chlorophyll-a (Chl-a) comparison. Left y-axis: mean Chl-a concentration values (1) between 0-20m (green) and (2) at the closest pixel to location of float profile (orange line). Right y-axis: the integrated Chl-a between 0-200m (dark blue line). Vertical dashed green line indicates start of ice-free season; vertical dash dot blue line indicates start of ice-season.

**Table 1.** Estimation of missed ocean-colour satellite NPP data based on complementary BGC-Argo float data and assessment of NPP input data

| Reason | Parameter | Estimate of missed data (%) |
|---|---|---|
| Sea ice | Annual integrated Chl-a within surface (0-20m) layer | 2 - 23% (median 10%) |
| | Annual integrated POC within surface (0-20m) layer | 7 - 30% (median 19%) |
| Low solar angle | Annual integrated Chl-a within surface (0-20m) layer | 1 - 19% (median 9%) |
| | Annual integrated POC within surface (0-20m) layer | 4 - 20% (median 12%) |
| Other data gaps | Fraction of IFA without NPP coverage when noon zenith angle >20°. (Caused by a combination of the "adjacency effect" of ice edge, cloud-fill errors, and poor spatial coverage of input variables (absorption, backscatter) | Weddell Gyre: 0-94% (mean 47%) Shelf: 0-65% (mean 10%) Open ocean: 0-97% (mean 50%) |

# 3  Results

## 3.1  Climatological NPP and Sea Ice

Total annual NPP integrated over the entire Weddell Gyre between 2003 and 2020 averaged ($\pm$ standard deviation) $172\pm34$ Tg C a$^{-1}$ before gap-filling and $269\pm39$ Tg C a$^{-1}$ after gap-filling (adjusting for the missed IFA; see Section. 2.4). Annual area-normalised production was on average $97\pm8$ g C m$^{-2}$ a$^{-1}$. While the open ocean experiences lower daily rates of productivity compared to the shelf region ($376\pm33$ mg C m$^{-2}$ d$^{-1}$ compared to $582\pm99$ mg C m$^{-2}$ d$^{-1}$; Figure 3.a), annual NPP is in fact higher per unit area in the open ocean than in the shelf region ($97\pm8$ mg C m$^{-2}$ a$^{-1}$, $68\pm23$ mg C m$^{-2}$ a$^{-1}$ respectively;

Fig. 3.b). This is due to a longer mean visible ice-free season: The sea-ice product shows that areas in the outer North-East edge of the open ocean are at the outer extent of the SIZ and can be ice-free for entire years, while on average, the whole open ocean region is ice-free for $139\pm13$ days per year. The longest any of the shelf region is ice-free is 157 days, while the mean is $37\pm13$ days. The open ocean also has a far greater area than the shelf region ($50.32$ x $10^5$ km$^2$ compared to $8.81$ x $10^5$ km$^2$, such that the open ocean represents 85% of the Weddell study region). As a result, when integrated over time and area, the open

ocean accounts for a significant majority of the total carbon taken up by phytoplankton in the Weddell Gyre and dominates the inter-annual variability of NPP seen in the region (Fig. 3.b and c). Before imputation, the total annual NPP in the open ocean is $170\pm33$ Tg C compared to $2\pm2$ Tg C in the shelf region (such that the open ocean accounts for $99\pm1$% of the total NPP in the Weddell Gyre). After imputation, annual NPP rises to $255\pm38$ Tg C a$^{-1}$ in the open ocean and $11\pm5$ Tg C a$^{-1}$ in the shelf region. Despite seeing a large increase in shelf estimates following the use of the gap-filling approach, the open ocean

still accounts for 96%$\pm2$% of the imputed Weddell NPP (ranging between 93-96% depending on the NPP model chosen).

As described in Section. 2.4 and summarised in Table. 1, BGC Argo float estimates of Chl-a and POC both under ice and at low sun angle suggest that substantial amount of phytoplankton biomass may be missed by satellite NPP products in the open ocean (medians of 20% Chl and 39% POC for all float locations). We did not attempt to further adjust our NPP estimates for

these missed data, because float biomass proxies are not directly proportional to NPP and because float coverage, while broad,
was not even throughout the Weddell Gyre.

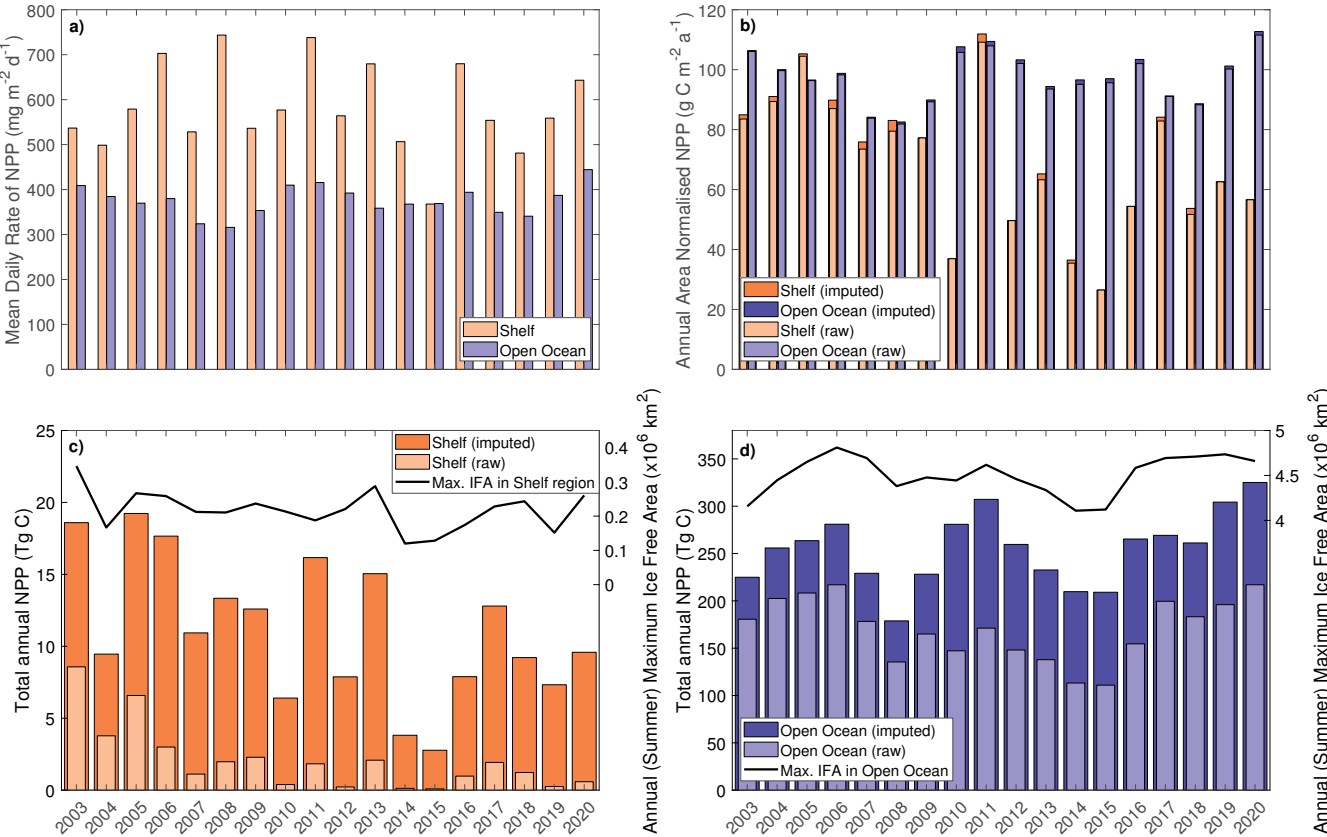

**Figure 3.** CAFE NPP: a) Mean daily NPP. Orange bars represent the shelf region value and blue bars for open ocean values; b) Area-Normalised Annual NPP; c) Shelf Region Total Annual NPP shown on the left axis and Annual (Summer) Maximum Ice Free Area on the right axis; d) Open Ocean Total Annual NPP shown on the left axis and Annual (Summer) Maximum Ice Free Area on the right axis. The light bars represent the raw (directly observed) values integrated from the CAFE NPP product, the dark bars indicate the imputed values calculated to consider data gaps in the spatial coverage of the CAFE product.

The summer maximum IFA for the entire Weddell Gyre averaged ± standard deviation $4.73\pm0.23$ x $10^6$ km$^2$ between 2003 and 2020 ($80\pm4\%$ of the total study region). The mean IFA per year for the entire Weddell Gyre averaged $2.18\pm0.24$ x $10^6$ km$^2$ between 2003 and 2020. On average in the shelf region, summer (maximum) IFA was $2.17\pm0.57$ x $10^5$ km$^2$ ($25\pm7\%$ of the shelf area), the mean IFA was $8.10\pm1.67$ x $10^4$ km$^2$. In the open ocean region region, the average summer (maximum) IFA was $4.50\pm0.22$ x $10^6$ km$^2$ ($89\pm4\%$ of the open ocean area), mean IFA averaged $2.08\pm0.23$ x $10^6$ km$^2$.

A clear seasonal cycle is seen in both NPP and IFA (Fig. 4). When comparing the mean annual cycle of open water area and total NPP, the rise in NPP coincides with the retreat of sea ice and increase in the area of ice-free water (Fig. 4). In both

regions, NPP peaks before the maximum open water area is seen and then declines despite the continued expansion of open water, creating a mismatch between trends of NPP and ice-free area from February.

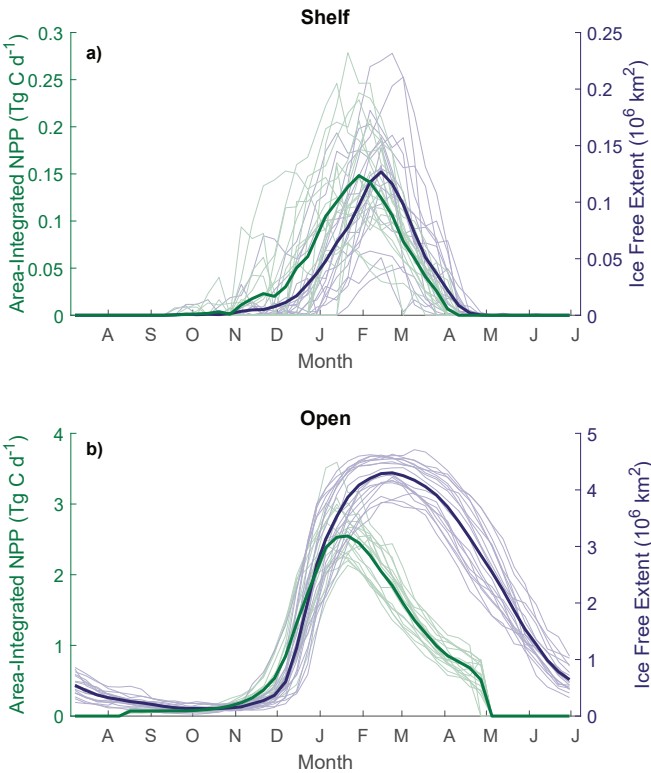

**Figure 4.** Seasonal cycles of total area-integrated gap-filled NPP (green) and ice-free extent (blue) across the (a) shelf region and (b) open ocean. Bold lines represent the 18-year mean, and thin lines show the annual variability.

## 3.2 Inter-annual variability and trends

No secular trends in total annual NPP were observed in the Weddell Gyre or the open-ocean region. Inter-annual variability is large, with production over the austral year in 2006 reaching 220 Tg C, while only 111 Tg C in 2015 (Fig.3.c). Successive NPP minima in 2008 and 2014-2015 may indicate a cyclical pattern with a period of 6-7 years, but a longer timeseries is needed to determine this conclusively. Likewise, no secular trend was observed in any of the sea-ice variables within the Weddell Gyre or open ocean over the study period. Potential causes of variability are multiple, including ice-free area, ice-free days, timing of ice retreat, cloudiness, wind speed and direction, sea surface temperature, vertical nutrient supply, glacial contribution. We investigate a number of these in the discussion below.

In contrast, a trend in NPP is seen in the shelf region. In the CAFE model, imputed NPP declined by 3% per year, $p=0.02$ (Fig. 3.b). A similar rate of decline, although less statistically significant, is seen in the other NPP models. The directly-observed CAFE estimates of NPP decreased more rapidly (average decrease of 7% per year, $p=0.001$), underscoring the large influence

of missing NPP data in the shelf region. Westberry et al. (2023) describe other potential causes for trends seen in NPP products (e.g. physiological changes in phytoplankton and decoupling of Chl-a and NPP), and emphasise the difficulty in identifying trends in NPP data and inferring drivers of trends. The trends seen here are sensitive to the occurrence of extremes in the early part of the time-period when there was a collapse of the Larsen B ice shelf along the Antarctic Peninsula (Peck et al., 2010).

No trend is seen in the shelf NPP when the first data point (year 2003) is removed.

Pearson correlation tests were carried out to determine the correlation between NPP and IFA. Single and multiple linear regressions were then performed to assess the individual and combined effects of open water area and time (year as well as duration of ice-free season visible to the satellite) on total annual NPP. Summer maximum IFA represents the largest area that is available for NPP to occur in a year. Furthermore, because the winter sea-ice edge in the Atlantic sector largely extends beyond

the Weddell Gyre boundaries used in this current study region (Fig. 1), the maximum IFA (minimum SIA) is also an indicator of the total area of SI retreat across the region within a year. The winter maximum sea-ice area within the study region box is much less variable than the summer maximum ice-free area (standard deviation: $6.6 \times 10^4$ and $2.3 \times 10^5$ km for the winter SIA and summer IFA respectively). The study area is almost entirely ice-covered each winter, and any variability in the actual maximum sea-ice extent occurs north of the study region and is not significantly reflected in the study box. Regression analysis

shows a significant relationship between the annual maximum IFA and total NPP (but that only explains approximately half of the variance), indicating that years with greater IFA result in more total NPP (Fig. 5). This relationship is seen in both the raw and imputed NPP estimates, but imputed NPP is shown in Fig. 5 and discussed here. IFA-NPP regressions for all NPP models can been found in Appendix Fig. A3. In the Weddell Gyre and open ocean sub-region, 42% of the inter-annual variability in total annual NPP can be explained by variability in the summer maximum IFA in each region ($p=0.002$, Fig. 5.a and b). This

relationship was strongest in the shelf region, with 55% of the variability in total NPP being explained by the yearly maximum IFA over the shelf ($p<0.001$, Fig. 5.c). Note that our annual NPP estimates are not independent of IFA, due both to NPP being set to zero under ice and to the IFA adjustment involved in gap filling. Therefore, the strength and statistical significance of these relationships are unsurprising and should be interpreted with caution. We therefore focus interpretation in the discussion on information contained in the residuals, with the unexplained variability being just as important.

Across the Weddell Gyre, and throughout the measurement period, a wide range of "satellite-visible days" was recorded. The mean number of satellite-visible days (days where the surface ocean is both ice-free and visible to MODIS-Aqua before the solar angle restricts satellite observations) across the open ocean was 94 days, but due to the size of the region and the rate of sea-ice retreat, there was a large range in the number of satellite-visible days seen at any given satellite pixel across the open ocean in any given year (0-262). The mean duration that the shelf region was visible to ocean-colour satellites was 49

days (range of 0-133 days for individual satellite pixels in individual years). We use this pixel-scale variability to further assess the relationship between ice cover (visible days) and NPP. The annual area-normalised annual NPP in any location (pixel) is significantly correlated to the length of time it is visible to the ocean-colour satellite during the year. Fig.6 shows an initial steady increase in in annual NPP with increasing satellite visible days in both the open ocean and on the shelf. However, in the open ocean, the rate of increase slows beyond ~130 ice-free days. In the shelf region, the increase in NPP when visible for

longer is greater than in the open ocean (one-tailed t-test (34 df) = -6.195,$p<0.005$), suggesting a more rapid response to earlier

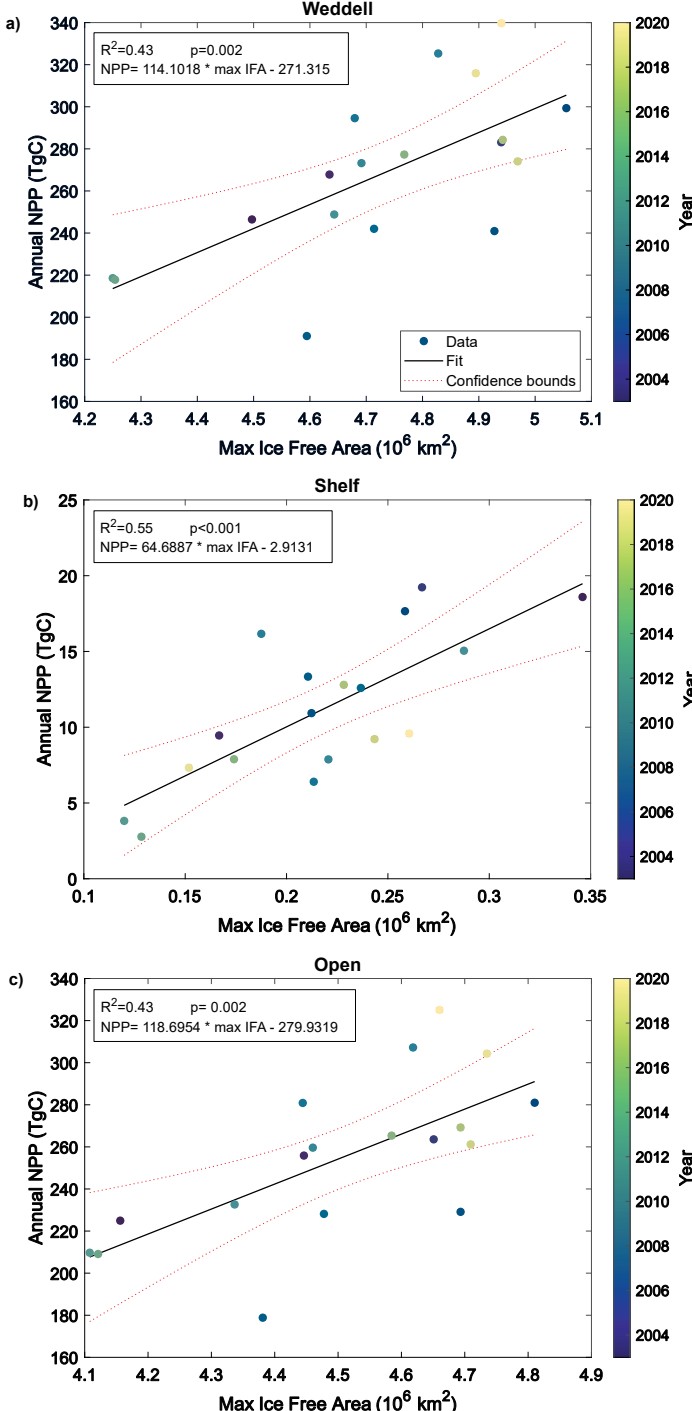

**Figure 5.** Relationship between annual maximum ice-free area and permuted annual NPP within a) the entire study region; b) the shelf region and c) the open ocean region. Dotted lines represent the 95% confidence bounds.

ice melt/longer ice-free (satellite-visible) periods. CAFE NPP on the shelf does not show signs of tapering off with increasing satellite-visible days. However, this is difficult to confirm, as very few on-shelf pixels are visible to the CAFE model for >130 days. Notably, other NPP products show similar relationships between visible days and NPP (Appendix Figures A4, A5, A6).

### 3.3 Aligning satellite and subsurface perspectives

Satellite observations indicate that, in the open ocean, the strong positive correlation between visible days and NPP degrades after around 130 visible days, indicating that other processes (e.g. grazing, nutrient availability) potentially begin to limit NPP after waters have been ice-free for more than 4 months. However, as described in Section 2.4, the ocean-color satellite loses coverage in late summer, when the solar angle decreases below 20°. As a result, it is uncertain whether further NPP is occurring, and therefore, missed after this point. Assessment of float Chl-a and POC (as proxies for phytoplankton growth/biomass)

can reduce this uncertainty by indicating whether phytoplankton are still present in the surface ocean and/or whether growth may still be occurring beyond the date when satellites lose visual coverage. Therefore we seek to address if significant growth is missed after loss of satellite coverage in late summer and whether the same relationship between ice-free days and phytoplankton growth is seen in the available float observations. Although these data come from drifting platforms, rather than fixed points, we can enquire how the seasonal cycles of Chl-a and POC unfold in each year, and specifically how they evolve relative

to light availability. It is worth noting that these data all represent open ocean conditions as floats are not deployed in regions shallower than 2000m.

Figure 7.a shows the progression of the bloom (Chl-a as proxy for growth) for each year for each float, starting at the first ice-free day. On average, 10% of the integrated annual Chl-a is recorded under the sea ice (max. 23%), so the majority of Chl-a is observed after the floats are considered to be in ice-free conditions. We evaluate the bloom progression for both the surface

(0 to 20m; analogous to satellite-observed depths; green lines in Fig.7.b) and upper 200m (blue lines in Fig.7.b), and define the bloom end as the date that Chl-a concentrations declined and remained below 50% of their seasonal maximum. Across all years and floats, median surface bloom end occurred 73 days after ice melt (range 31-134 days), while median depth-integrated bloom ended on day 90 following ice melt (range 50-165 days; Fig. 7.a). The later date of bloom end for the depth-integrated values indicates that primary production could be continuing in the water column at depth until later in the season than at the

surface. Certain float years in Figure A2 (panels a-d and k) see a concurrent increase in Chl-a and POC in the depth integrals that indicates that active production/growth is taking place, as opposed to changes in phytoplankton physiology or community composition driving changes in Chl-a (Thomalla et al., 2017).

A significant decline in Chl-a consistently occurred prior to the low solar angle and substantially before the return of ice coverage (Fig. 7.b). Overall, surface bloom ends preceded low solar angle by median 50 days (range: 100 days before to 10 days

after) and preceded ice coverage by median 130 days (range: 176 days before to 0 days). Surface and depth-integrated blooms ended before the date of low solar angle and the date of sea-ice return in all but one of the 23 float-years. This result adds independent support to the satellite-derived finding that open Weddell Gyre phytoplankton blooms decline after 3-4 months ice free and extends the finding to full euphotic zone. The slightly later decline in depth-integrated Chl-a relative to surface Chl-a also suggest that factors other than light contribute to the end of the bloom.

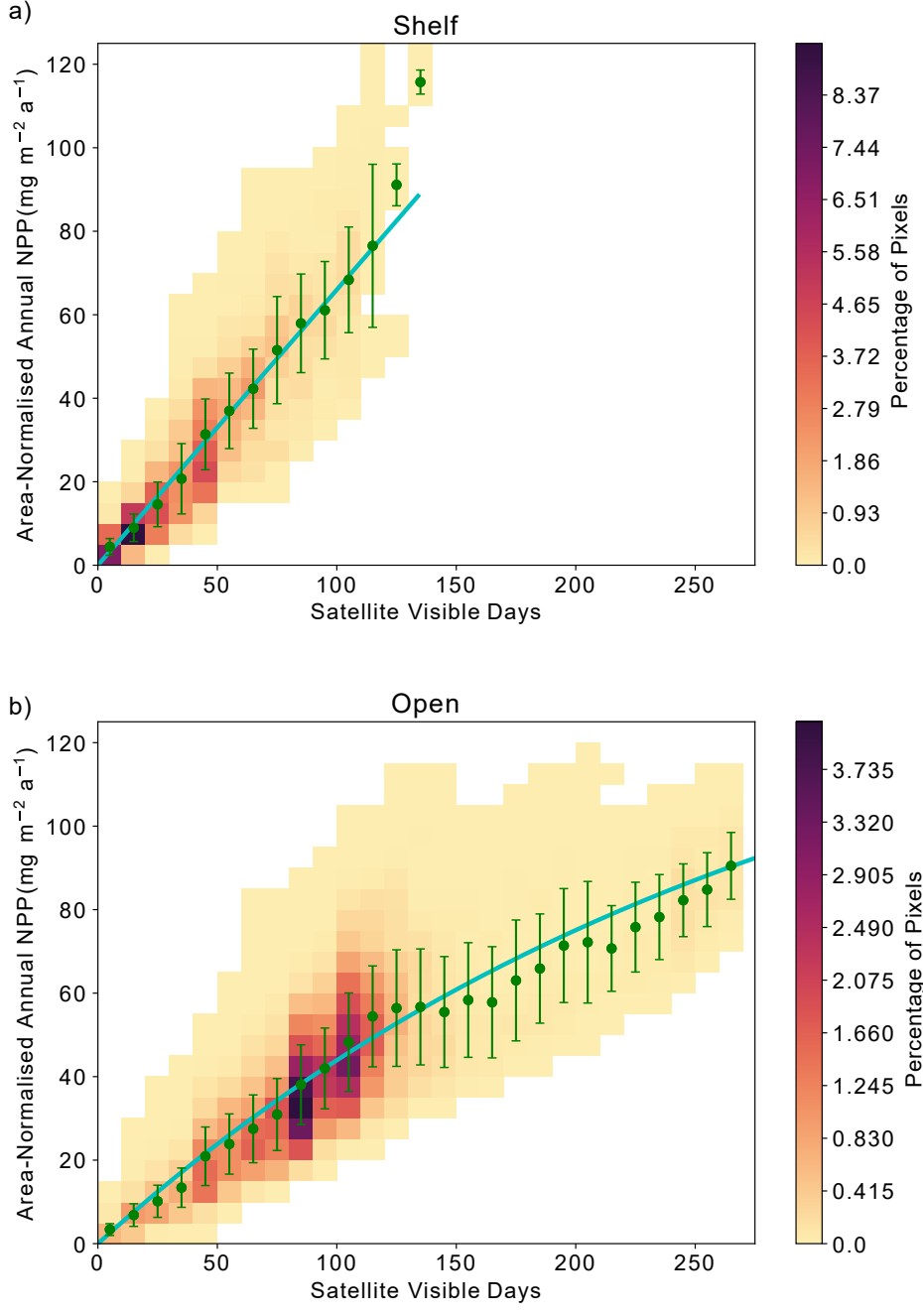

**Figure 6.** Relationship between area-normalised annual NPP and number of visible days (VD) for all satellite data pixels over the timeseries. Heatmap shows the area-normalised density distribution of data points, while the lines show the a) linear (shelf; $\mathrm{NPP} = 0.66 * \mathrm{VD}$) and b) non-linear (open ocean; $\mathrm{NPP} = 151.12 \times (1 - e^{-\mathrm{VD}\frac{0.52}{151.12}})$) relationships for the whole timeseries. VD bin means $\pm$ standard deviation are plotted in green.

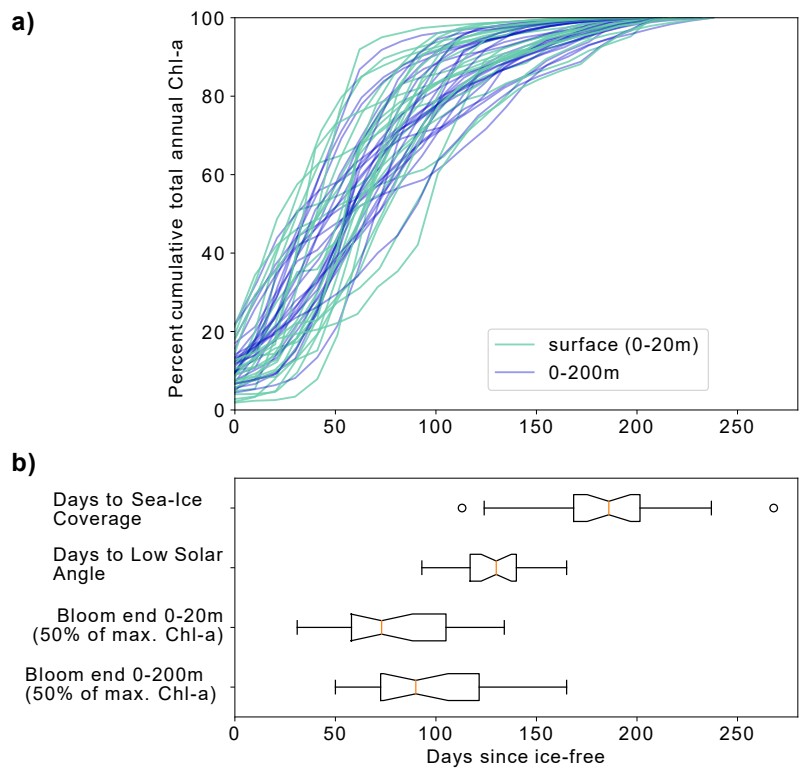

**Figure 7.** a) Percent cumulative Chlorophyll-a in the surface (0-20m) and between 0-200m for all completed annual cycles by the floats between 2015-2021. b) Box and whisker plots indicating the range in the number of days from ice melt to (i) sea-ice return, (ii) low solar angle (loss of satellite observations), (iii) surface bloom end and (iv) depth-integrated bloom end. Orange line represents the median value for each, the box represents the first and third quartiles, while the whiskers extend to 1.5 times the inter-quartile range.

## 4   Discussion

### 4.1   Biological carbon uptake in the Weddell Gyre dominated by NPP in open ocean region

This study shows that over the last 18 years, open ocean productivity has a dominant role in the biological carbon cycle of the Weddell Gyre, in agreement with synoptic ship-based studies (MacGilchrist et al., 2019; Brown et al., 2015). MacGilchrist et al. (2019) documented that the region was a net annual $CO_2$ sink only due to a large amount of carbon sourced from biological activity in the central Weddell Gyre accumulating in the local Circumpolar Deep Water; horizontal circulation of this water mass out of the Weddell Gyre then facilitates sequestration of carbon over climate-relevant (centennial to millennial) time scales. Our current study highlights the importance of the Weddell Gyre's open ocean compared to the shelf region, showing that the majority (93-96%) of the carbon uptake by phytoplankton in the Weddell Gyre occurs there, despite higher daily rates

of NPP observed on the shelf. This is due to its far greater areal extent (an order of magnitude larger than the shelf region)
and much longer ice-free growing season. In light of our results here, we highlight the need for future research in this region
to further quantify the contribution of open ocean production to the biological carbon pump. Satellite NPP values are often
underestimated due to the negative biases in spatial coverage within NPP products. Gap-filled, permuted NPP estimates were
calculated to reduce these uncertainties and biases, particularly in the shelf region where spatial coverage was impacted by
significant gaps in the data. These values are considered to be more realistic estimates of the total NPP occurring in the two
regions. It is important to note that the gap-filling correction assumes that mean NPP is equal in the visible and non visible
portions of the region and so could therefore still carry uncertainties.

## 4.2 Drivers of NPP variability

Our results show a clear relationship between sea ice and NPP on an inter-annual timescale (Fig. 6, but that explains only
40-55% of the variance in NPP. A large fraction of this arises from the role of sea-ice cover in setting light availability in the
SIZ. We discuss the nature of this relationship in the following section, before exploring other possible sources of inter-annual
variability.

### 4.2.1 Sea ice as a control of light availability

Light availability is restricted under sea ice, so when sea ice retreats, light limitation caused by sea-ice coverage is alleviated
(Twelves et al., 2021; Arrigo and Van Dijken, 2011; Arrigo et al., 2015; Rohr et al., 2017; Smith and Comiso, 2008). Therefore,
while the annual NPP cycle at high latitudes is generally driven by regular seasonal changes in solar angle/position through
the year (Ardyna et al., 2017; Arrigo et al., 2008; Park et al., 2017; Smith and Comiso, 2008), within the SIZ, the initiation of
growth and the total NPP that results is also mediated by sea-ice cover and its inter-annual variability (Rohr et al., 2017; Twelves
et al., 2021). Our basin-wide satellite-based NPP analysis assumes no NPP under ice and, as expected, finds sea-ice coverage to
be a dominant driver of inter-annual variability in NPP. Annual summer minimum SIA (summer maximum IFA) in particular
explains 40-55% of the NPP variability within the subregions and the Weddell Gyre overall (Fig. 5). The float data showing
that 2-23% of integrated Chl-a (and 7-30% of surface POC; Table. 1) is present potentially before sea-ice retreat suggests that
our the satellite analysis may over-estimate somewhat the correlation between IFA and NPP. Recent studies (Bisson and Cael,
2021; Hague and Vichi, 2021; McClish and Bushinsky, 2023) have also reported the presence of considerable amounts of Chl-a
under sea ice as well as highlighting the onset of growth prior to complete sea-ice retreat (Hague and Vichi, 2021; McClish
and Bushinsky, 2023). However, while our float observations also indicate that biomass tends to increase before complete ice
retreat, our results still clearly show IFA as a major productivity driver. Strong phytoplankton growth follows ice melt (Fig.
7) and the majority of phytoplankton biomass is found in ice-free conditions (Figs. 2, A1 and Table 1). Similarly, in McClish
and Bushinsky (2023), the break-up of sea ice initiates the increase in Chl-a and POC, highlighting the light limiting control
of sea ice on phytoplankton growth. Recent work has shown that seasonal to inter-annual variability in Antarctic sea-ice cover
exhibits substantial predictability (Libera et al., 2022; Bushuk et al., 2021). The strong relationship seen here between sea ice
and NPP therefore indicates that sea-ice predictability may translate into predictability of NPP, with consequences for fisheries

and ecosystem management. Indeed, this link between sea-ice and NPP predictability was recently shown in a perfect model context (Buchovecky et al., 2023).

Despite this strong control of IFA on total NPP, there is still a considerable amount of inter-annual variability that is not explained by the variability in the summer maximum IFA (Fig. 5). Some of the unexplained variance in NPP could be related to the spatial patterns and variability of sea-ice retreat. Daily average NPP exhibits substantial spatial variations (not shown), with "hotspots" in the eastern Weddell Gyre, particularly around Maud Rise, along the narrow shelf, and in the open ocean near the eastern boundary of the Weddell Gyre. These hotspots are thought to be set by comparatively high levels of nutrient supply (e.g. Vernet et al., 2019; Geibert et al., 2010; Arrigo et al., 2015; Moreau et al., 2023). Consequently, in any given year, spatial variations in where and when sea ice retreats will determine whether or not these hot spot regions are exposed and for how long, with concomitant impacts on integrated carbon uptake across the Weddell Gyre. Inter-annual variability in sea-ice cover is set by physical mechanisms, such as atmospheric forcing (e.g. phase of the Southern Annular Mode) and ocean forcing (e.g. sea surface temperature) (Kumar et al., 2021). Moreau et al. (2023) found that strong winds transport sea ice towards the shelf, potentially removing light limitation to the surface waters as a result.

At a local level, area-normalised NPP appears strongly related to the duration that an area has sufficient available light for satellite detection (Fig. 6; see Sect. 2.4 for a discussion on the distinction between visible days and light availability). Over shelf sea regions, sea-ice persists for longer and re-forms earlier than in the open ocean, meaning that the duration of light availability is set by both the retreat and return of sea ice (Fig. 4). The strongly linear relationship in Fig. 6.a suggests that phytoplankton on the shelf are primarily limited by light availability, itself controlled by sea-ice cover. This is consistent with the expectation that the shelf sea regions are nutrient (specifically iron) replete (Arrigo et al., 2015; Boyd et al., 2012; McGillicuddy et al., 2015; Sedwick and Ditullio, 1997), such that they do not become nutrient limited during the short ice-free season.

### 4.2.2 Nutrient limitation

In the open ocean subregion, once the sea ice is gone, other factors beyond light limitation – such as nutrient supply – determine how much NPP takes place, and for how long (Fig. 6.b). We hypothesise that the open ocean experiences a progression from light limitation when sea ice is present to nutrient limitation once the ice-free season has persisted longer than ∼80-130 days (Figures 6.b,7). This picture is consistent with other studies that identify a progression of limiting factors in the SO (Arrigo et al., 2015; Ryan-Keogh et al., 2017; Sedwick et al., 2011; Twelves et al., 2021; von Berg et al., 2020). In the open ocean, the earlier melt and later return of sea ice compared to the shelf region (Fig. 4) means that the number of visible days (and by consequence ice-free season) can be much greater (Fig. 6.b). The weakening/plateau of the relationship between the number of visible days and NPP (Fig. 6.b) indicates the exertion of a limiting control, such as nutrient availability or top-down controls, mediating NPP towards the end of the growing season. The sub-surface float data support this finding: For the majority of float years, a decline in Chl-a occurs before the solar angle is below critical (20°) in March, and well in advance of the return of sea ice to the float's location (Fig. 7). The average timing of this decline (73 days at the surface) is broadly consistent with what is seen in the satellite results. The implication is that the decline in growth after 2-3 months of ice-free conditions leads to a suppression in annually integrated NPP in areas where the ice-free season lasts longer than approximately 3-4 months.

Thus, nutrient limitation could be setting an upper limit to NPP and effectively dampen the influence of sea ice on inter-annual variability, particularly in areas that experience longer ice-free seasons.

The nutrient limitation hypothesis is well supported by existing literature. Much of the SO is macro-nutrient replete but micro-nutrient limited (primarily by iron (de Baar et al., 1995; Hauck et al., 2015) and possibly by manganese (Hawco et al., 2022)). Some areas of the SO also experience silicate limitation after the spring bloom (Lafond et al., 2020; Quéguiner, 2013), although this is unlikely in the Weddell Gyre, as high concentrations of silicate have been documented through extensive repeat sections through the region (Hoppema et al., 2015). Iron limitation has been found to be more prevalent with distance from the ice shelf, as well as later in the growing season, when the 'winter inventory' has been utilised (Twelves et al., 2021; Boyd et al., 2012; Hoppema et al., 2007). In the Weddell Gyre, the areas that have the longest ice-free/satellite-visible seasons are also generally the areas furthest from the ice shelf and continent, and therefore furthest from an abundant iron supply. We postulate therefore that iron limitation could be a major driver of the slowing/decline in NPP that occurs prior to sea-ice return in the open ocean.

Our hypothesis of iron limitation at the end of the growing season is supported by the sub-surface Chl-a and POC observed by floats (Appendix Figures A1 and A2). Changes in Chl-a concentrations can arise from several situations aside from growth/accumulation of biomass: photo-acclimation, nutrient limitation and changes in phytoplankton community composition (Thomalla et al., 2017). Comparing Chl-a to POC, we can assess what may be causing changes in Chl-a. The presence of elevated Chl-a concentrations close to or below the base of the mixed layer, often (but not always) after the cessation of the initial surface bloom (Appendix Figure A1), suggests that phytoplankton are benefiting from replenishment of nutrients from below the mixed layer through diapycnal mixing (Arrigo et al., 2015; Taylor et al., 2013). Elevated POC signals coincide with increased Chl-a in the majority of these cases, providing evidence that active production is taking place at depth (Appendix Fig. A2). Surface nutrient concentrations are thus likely to be limiting phytoplankton growth in many areas of the ice-free Weddell gyre, although float data do not allow us to quantify its net impact on NPP. Grazing pressures may also be important in driving the differences in surface and sub-surface phytoplankton dynamics (Baldry et al., 2020, also see Section 4.2.3).

The complexity of the relationship between light and nutrient limitations – and their implications for inter-annual variability in annual NPP – is highlighted by the occasional occurrence of a secondary (temporally separated) late-summer bloom (Appendix Figure A1. e.g. panels a) 5904397: 2018, 2019; b) 5904467: 2018; c) 5904468: 2018, 2019; d) 5904471: 2018; g) 5905992 2020). As seen in the matching Chl-a and POC signals at depth, the second peaks in surface and depth integrated Chl-a that suggests a late-summer bloom are matched by simultaneous POC increases at these times, implying active growth within the phytoplankton community (Appendix Fig. A2). There are four float years (Appendix Fig. A2 h) 5905994: 2020; j) 5906033: 2020; and k) 5906034: 2020, 2021) that saw small increases in Chl-a at the end of the ice-free season without a concurrent increase in POC. We conclude that the increase in Chl-a in these cases may be a result of phytoplankton photo-acclimating to the decreasing light conditions. The secondary blooms typically follow a set pattern (von Berg et al., 2020): when sea-ice melt occurs earlier, iron resources are then also depleted earlier in the growing season. The secondary bloom initiates when micro-nutrients are replenished from entrainment/diapycnal mixing (Arrigo et al., 2015; Taylor et al., 2013), before it is later cut off by a combination of sea-ice return and reduction in PAR due to a shoaling of the light penetration

depth (Appendix Figure A1). Two temporally separate blooms are not always visible, with previous studies suggesting that the timing of the first bloom may be a key driver for this (von Berg et al., 2020), otherwise the two blooms will overlap in time sufficiently as to be indistinguishable. Further iron limitation, grazing pressure, or low light levels bring an end to the second bloom. The observation of double blooms, particularly with the latter sometimes occurring at different locations in the water column, indicates that the processes limiting NPP or acting as a brake on its magnitude are not identical in all locations/water depths.

As with sea-ice cover and its impact on light limitation (Sect. 4.2.1), physical mechanisms drive inter-annual variability in nutrient supply, with plausible implications for the variability in total annual NPP. Changes in iron supply to the ocean surface and thus the alleviation of iron limitation can occur through variability in mixing (Prend et al., 2022; Biddle and Swart, 2020; Swart et al., 2020), upwelling of iron rich deep waters (Moreau et al., 2023; Twelves et al., 2021; Hoppema et al., 2015; Vernet et al., 2019), as well as supply of icebergs to warmer areas of the Eastern Weddell Gyre ($\sim$ 20 - 25 °E) (Geibert et al., 2010). Furthermore, in addition to its impact on light availability, sea-ice dynamics play an important role in supplying iron to the SIZ (Boyd et al., 2012), possibly complicating the impact of sea-ice variability on total annual NPP.

### 4.2.3 Grazing

Within the scope of this study, it has not been possible to assess additional controls on phytoplankton growth, beyond sea-ice-induced light availability (Sect. 4.2.1) and nutrients (Sect. 4.2.2). However, it is plausible that other factors could be important for the inter-annual variability of NPP, specifically top-down controls (such as zooplankton grazing or microbial activity). The presence of sea ice can act to decouple grazers and phytoplankton (Hoppema et al., 2000; Rohr et al., 2017; Smetacek et al., 2004), and as micro-nutrient availability diminishes, grazing may increasingly contribute to the decline of the NPP bloom such that, by late summer, grazer populations may be the dominant control of phytoplankton biomass/communities (Rohr et al., 2017; Smetacek et al., 2004). Vernet et al. (2019) review various studies that have highlighted the abundance of higher trophic levels in areas of the Weddell Gyre, and Kauko et al. (2022) recently highlighted krill as exerting top-down (grazing) control on diatom populations in the Kong Håkon VII Hav. However, there is a lack of widespread zooplankton research that has taken place through the Weddell Gyre open ocean, and so it is difficult to determine whether grazers are a significant control of phytoplankton. Variability in ecosystem composition is also likely a significant contribution to the temporal and spatial signal of integrated NPP (Lin et al., 2021; Trimborn et al., 2019; Mascioni et al., 2021; Takao et al., 2020).

### 4.3 Implications for the future

Sea-ice extent around Antarctica is expected to change in response to anthropogenic global warming (Kumar et al., 2021; Ludescher et al., 2019; Casagrande et al., 2023). With its strong link to NPP, as exhibited in this study, changes in sea-ice dynamics could strongly impact biological carbon uptake in the Weddell Gyre. In turn, this could affect the ecosystem health as well as the contribution that the Weddell Gyre makes to global carbon uptake and climate (Henley et al., 2020).

Our results strongly suggest that, within the SIZ, a larger ice-free area and longer ice-free season (such as might be expected in a warmer world) will lead to higher total annual NPP in many regions, assuming there are no changes to other environmental

variables such as nutrient supply and grazing. However, our results also indicate that this increase is not likely to be linear, and beyond a threshold in the length of the ice-free season, NPP will cease to increase at the same rate. Our results further imply that nutrient supply is a key control on this upper limit for NPP in the present day in the open ocean. Consequently, how NPP will change across the Weddell Gyre becomes sensitive to how iron supply will change in the future. Such changes could also be mediated by changes in sea-ice dynamics due to their impact on stratification, mixing and upwelling (Moreau et al., 2023; Hoppema et al., 2015). In future warming conditions, increased stratification, combined with freshening from melting ice could act to cut off biological productivity by reducing the vertical nutrient supply (Bronselaer et al., 2020). This will be particularly apparent in the open ocean, given its greater distance from terrestrial micro-nutrient sources. Noh et al. (2023) recently showed that, within CMIP models, Chl-a in the Arctic declines as a result of reduced nutrient supply when regions become ice-free. Despite being based in the Arctic, and thus differing physically and ecologically from the SO, this result in Noh et al. (2023) could point to a less productive Weddell Gyre in the future, should any of it become permanently ice-free. Notwithstanding changes in nutrient supply, an increasingly ice-free Weddell Gyre will see a greater expanse experiencing nutrient limitation late in the growing season. It is unclear whether the same limitations will have a similar effect on NPP in the shelf region should it become increasingly ice-free for longer than is currently seen ($\sim$ 130 days).

## 5   Conclusions

This study used a complement of satellite-derived sea ice and NPP products as well as BGC-Argo float observations of Chl-a and POC as proxies for phytoplankton biomass to assess the basin-scale relationship between sea ice and phytoplankton growth. We find that sea ice is the primary control on Weddell Gyre NPP in areas that experience fewer than 70-130 ice-free days per year. Beyond $\sim$ 130 ice-free days, float Chl-a and POC observations suggest that nutrients (likely iron) emerge as an important limit to growth, possibly co-limiting with top-down grazing control. We find that while the shelf region sustains higher instantaneous NPP during its ice-free window, the open ocean sustains 93-96% of the annual NPP of the Weddell Gyre, due to its larger area and longer ice-free season. Furthermore, while sea ice is a primary driver of inter-annual variability in total annual NPP in the Weddell Gyre, nearly half of NPP variability is still unexplained, motivating further study. We found no long-term trends in the Weddell Gyre sea-ice extent or NPP during the study period. However, our results suggest that NPP will increase if sea-ice extent decreases in the future, at least until the Weddell Gyre is ice-free for longer than 130 days, at which point, controls other than sea ice may dominate. Finally, this work has highlighted the importance of using BGC-Argo float data to complement and corroborate satellite data analysis. The study highlights the need for development of quantitative float-based NPP measurements in the region, which would likely benefit from inclusion of PAR sensors on more floats.

*Data availability.*   Raw data available from: Oregon State www.science.oregonstate.edu/ocean.productivity/ and SOCCOM December 2021 Snapshot https://doi.org/10.6075/J00R9PJW

*Code and data availability.* Processed data from data sources above and code used to process data and create figures for this paper are available at https://doi.org/10.5281/zenodo.7951184

## Appendix A: Supporting Figures

*Author contributions.* Conceptualisation and design of study were carried out by all authors. Data preparation, analysis and writing by CCD. Editing was carried out by all authors

*Competing interests.* The authors declare no competing interests.

*Acknowledgements.* This work was supported by the Natural Environmental Research Council (NE/S007210/1 & NE/X008657/1) and by a European Research Council Consolidator grant (GOCART, agreement no. 724416). GAM was supported by the NSF-funded SOCCOM project (PLR-1425989) and UKRI (MR/W013835/1).

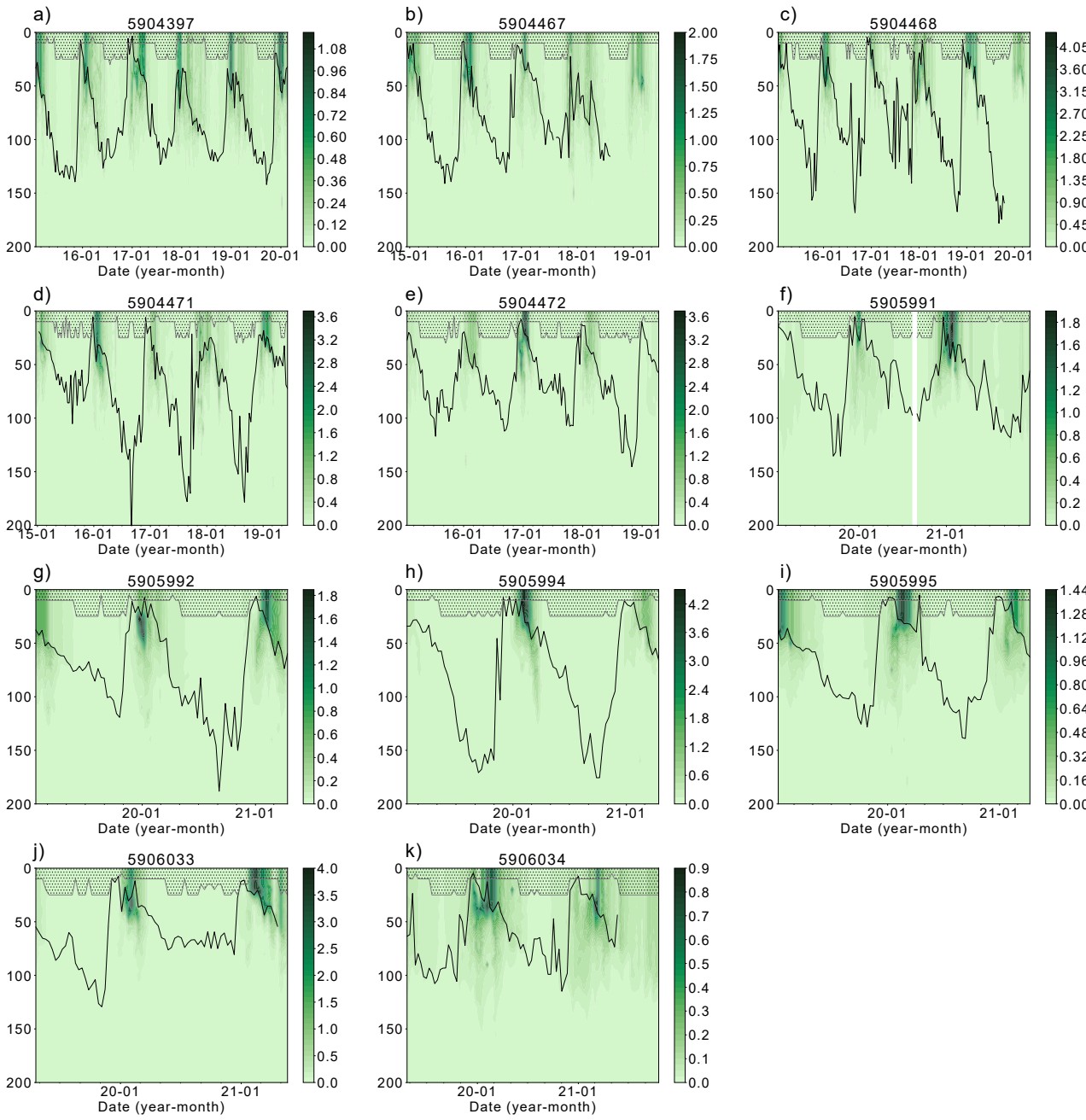

**Figure A1.** Chlorophyll-a concentration between 0-200m for each float found within the Weddell Gyre study region between 2014-2021. Black line represents the mixed layer depth and shaded areas represent the data extrapolated to the surface from the shallowest float measurement. Note colour scales vary between floats

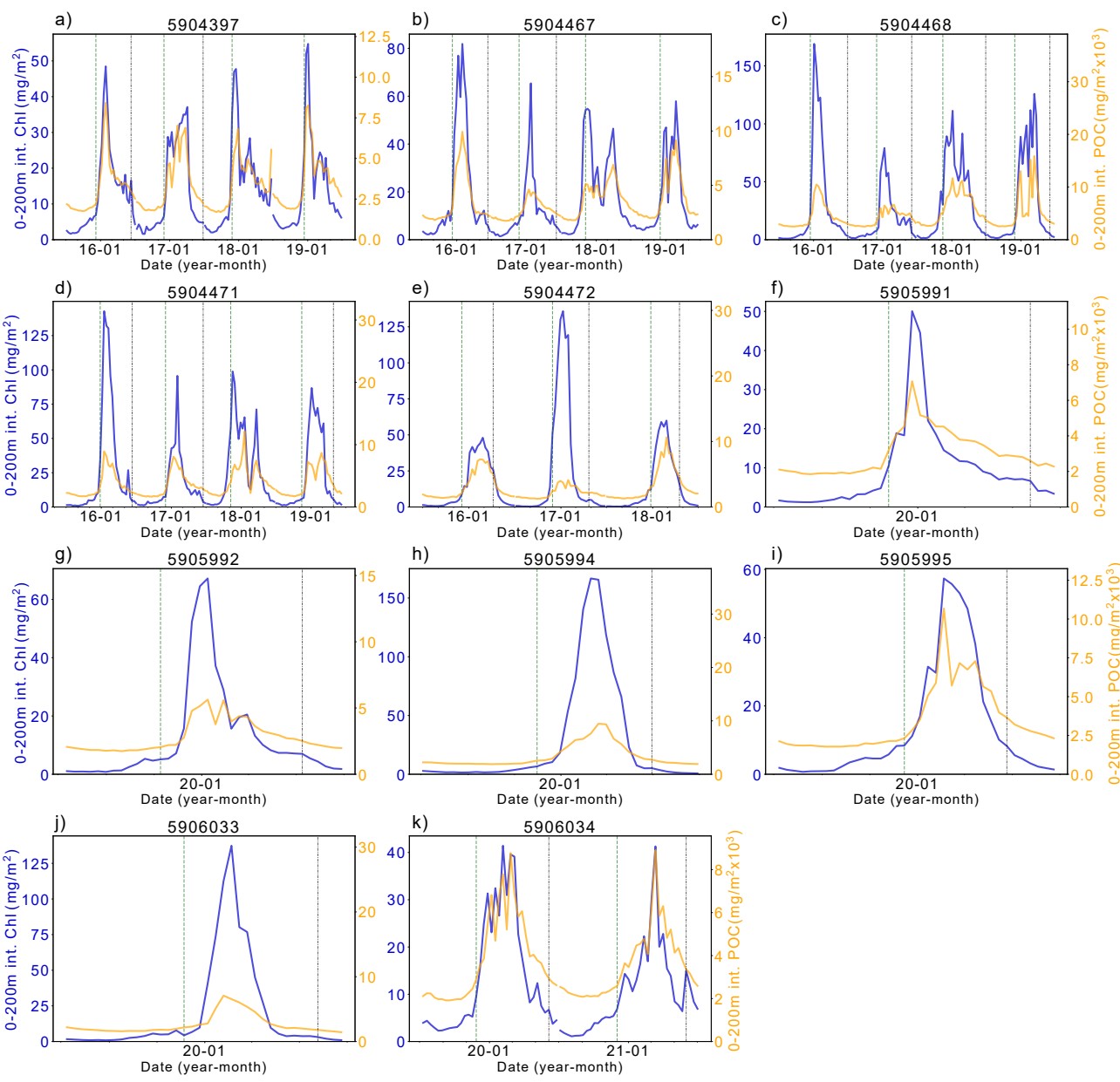

**Figure A2.** Chlorophyll-a (Chl-a) and particulate organic carbon (POC) comparison. Left y-axis shows the integrated Chl-a between 0-200m (dark blue line). Right y-axis shows the integrated POC between 0-200m Vertical dashed green line indicates start of ice-free season; vertical dash dot blue line indicates start of ice-season.

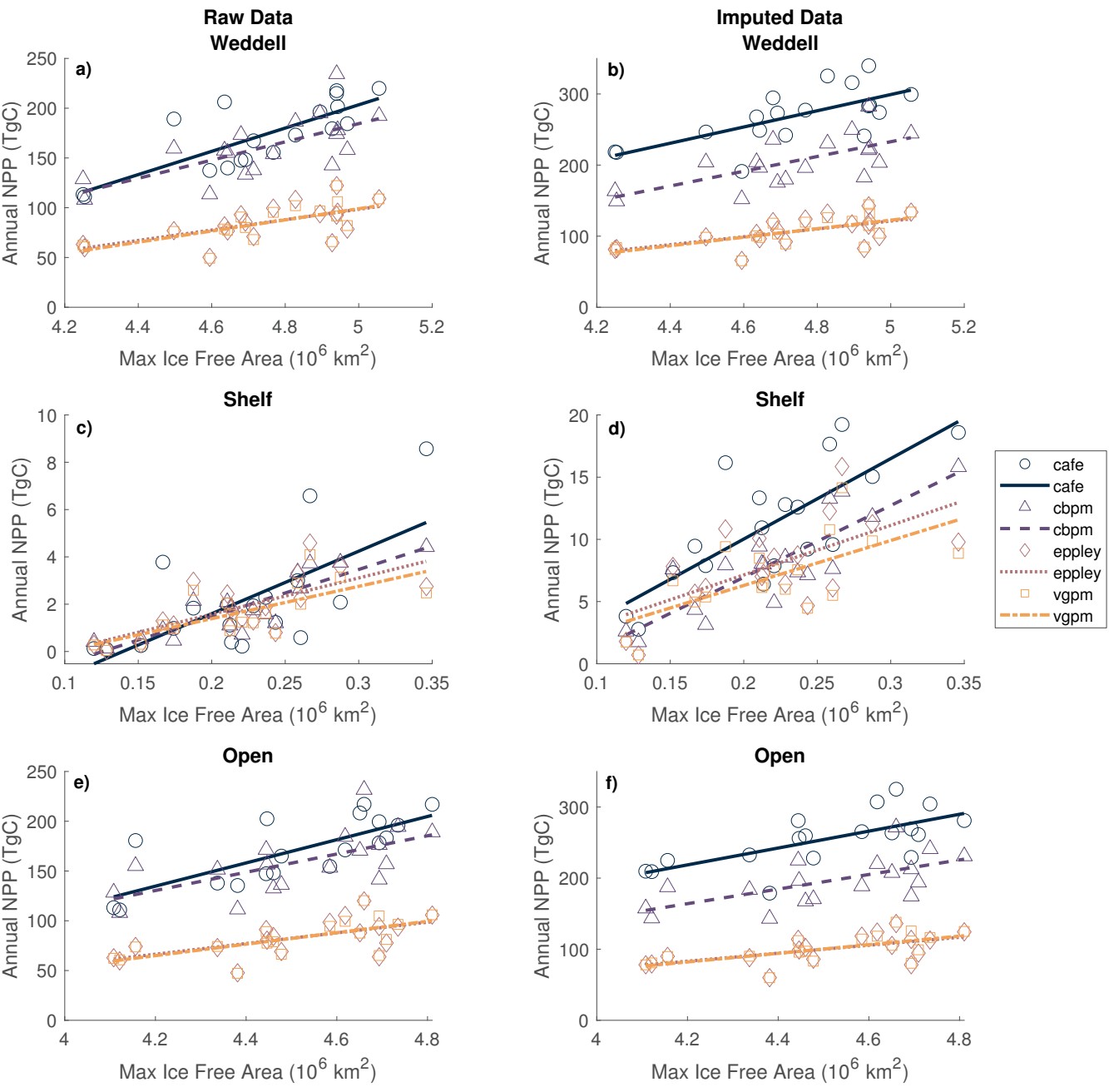

**Figure A3.** Relationship between annual maximum ice-free area and annual NPP for all four NPP models for the entire Weddell Gyre (a-b), Shelf region (c-d) and Open Ocean region (e-f). Panels a), c) and e) show the regressions using the directly-observed raw estimates of total annual NPP. Panels b), d) and f) show the regressions using the imputed annual NPP estimates.

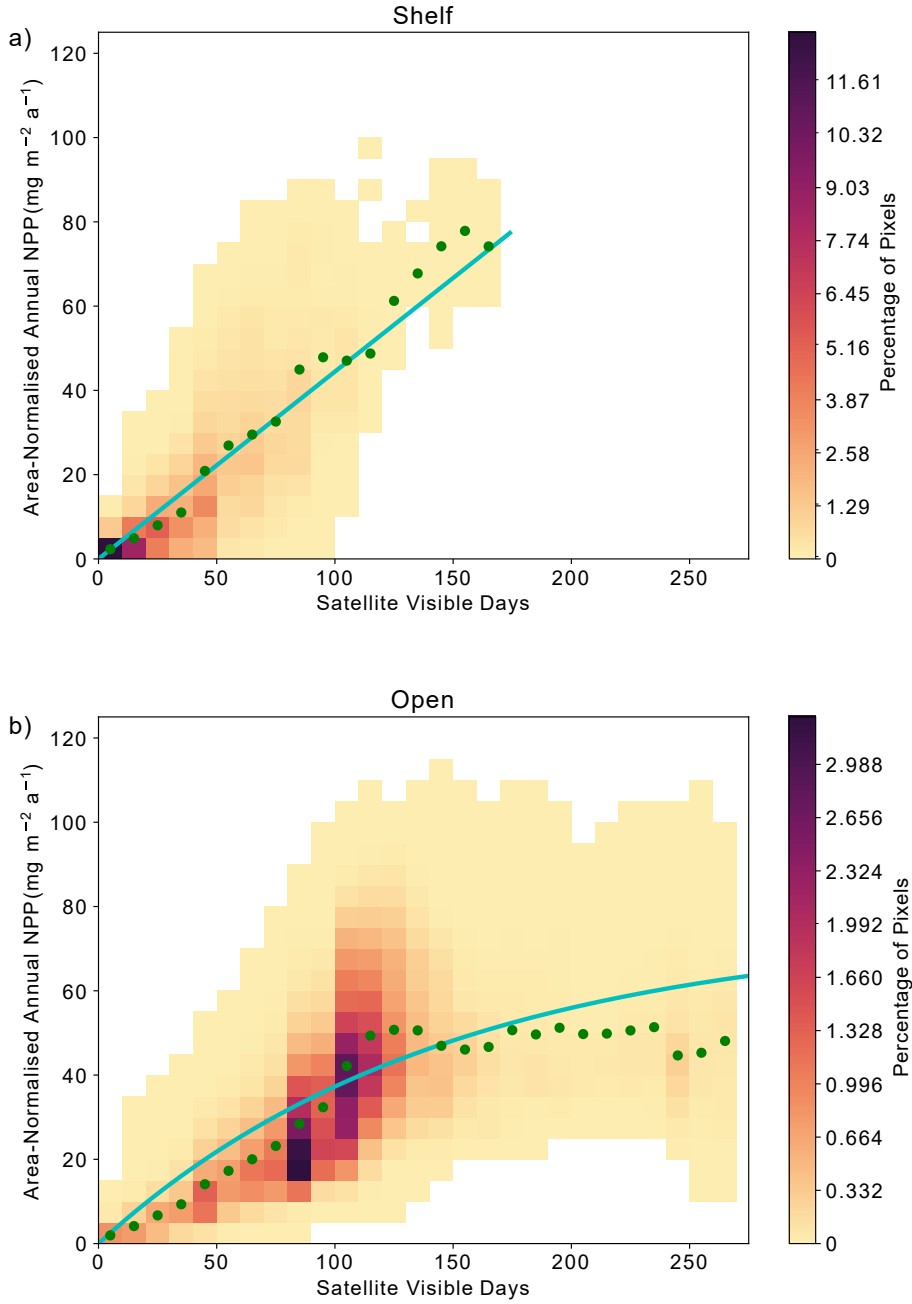

**Figure A4.** Relationship between area-normalised annual NPP and number of visible days (VD) for all satellite data pixels over the timeseries as reported in the CbPM NPP product. Heatmap shows the area-normalised density distribution of data points, while the lines show the (a) linear (shelf) and (b) non-linear (open ocean) relationships for the whole timeseries. VD bin means are plotted in green.

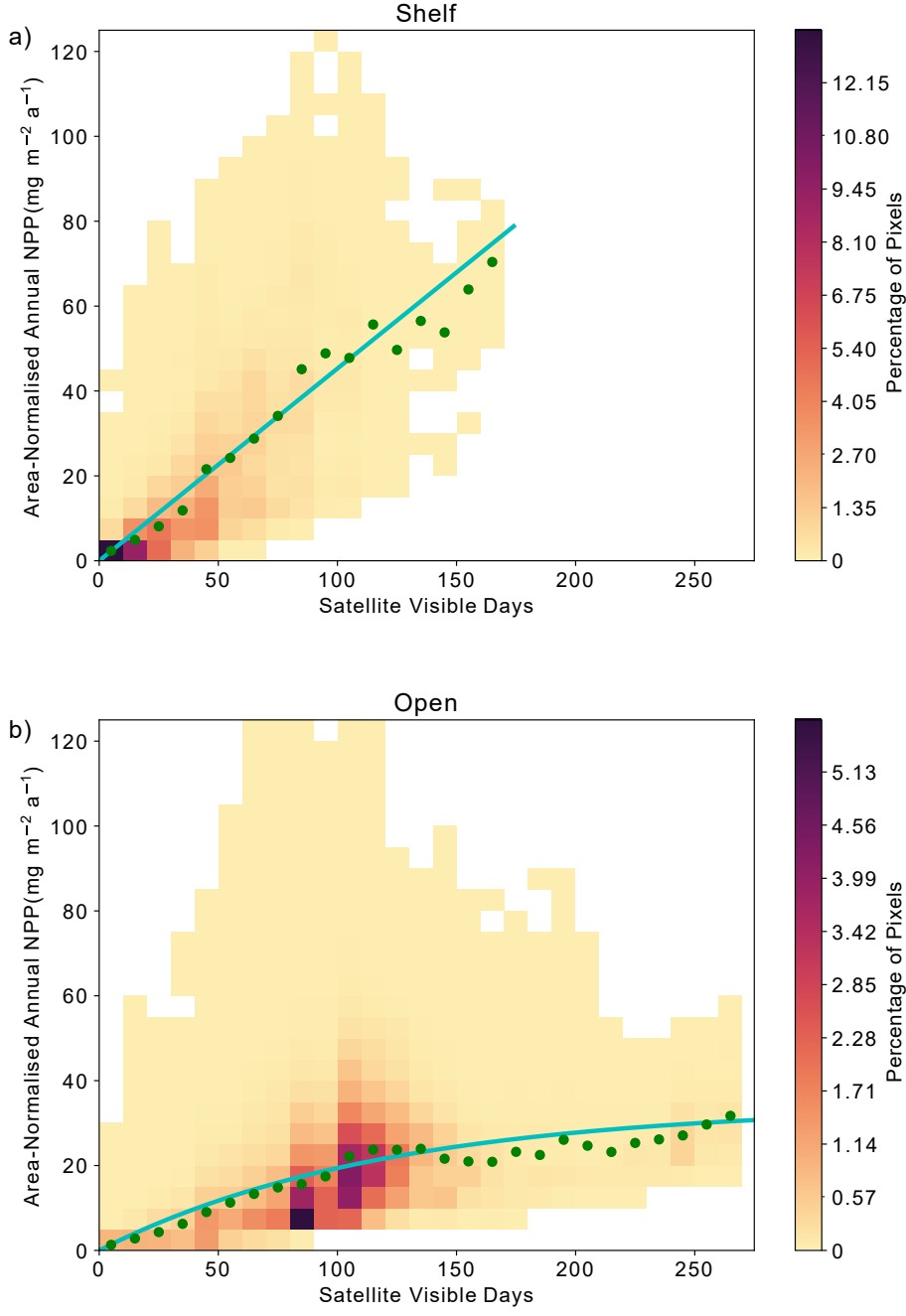

**Figure A5.** Relationship between area-normalised annual NPP and number of visible days (VD) for all satellite data pixels over the timeseries as reported in the VGPM-Eppley NPP product. Heatmap shows the area-normalised density distribution of data points, while the lines show the (a) linear (shelf) and (b) non-linear (open ocean) relationships for the whole timeseries. VD bin means are plotted in green.

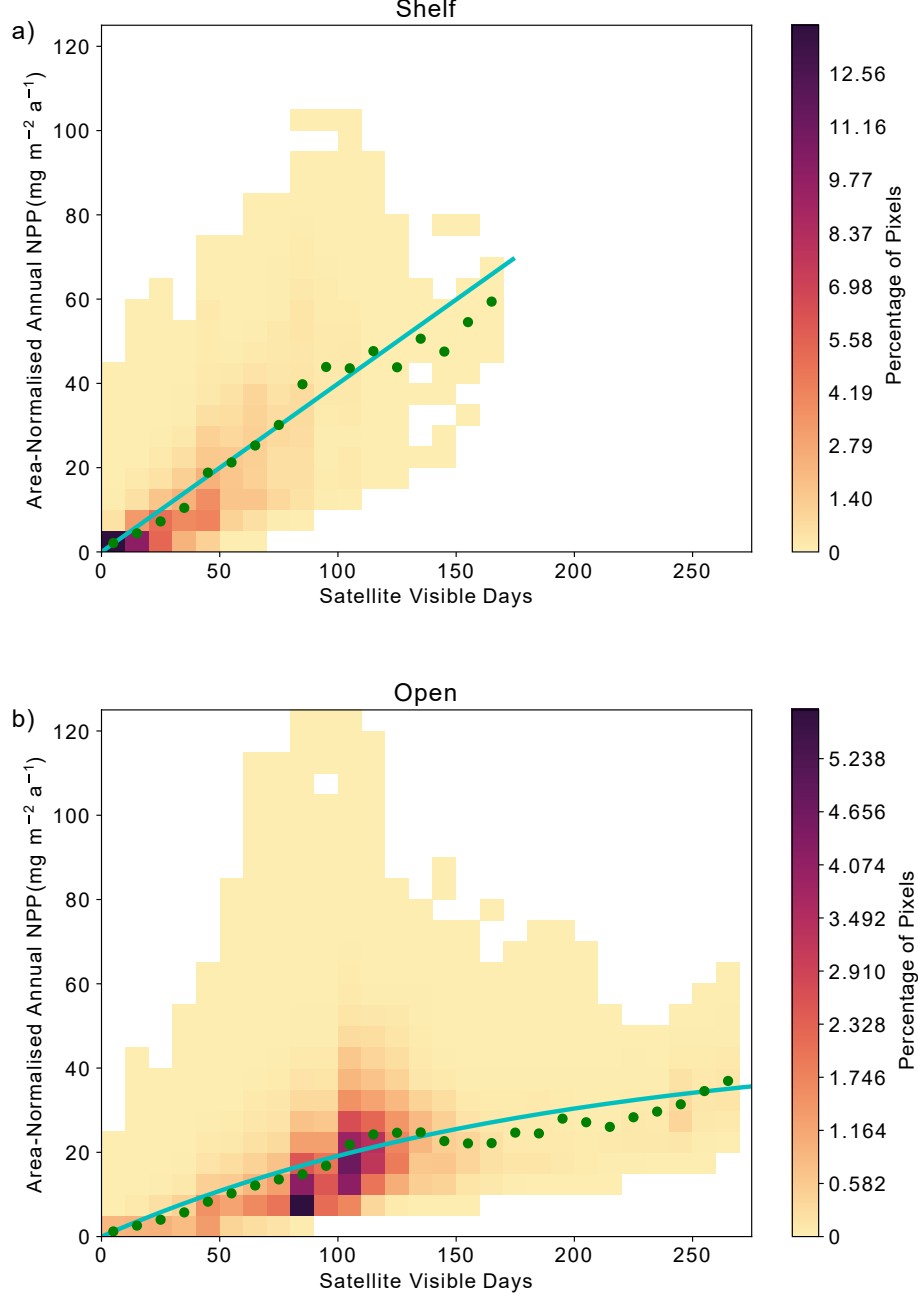

**Figure A6.** Relationship between area-normalised annual NPP and number of visible days (VD) for all satellite data pixels over the timeseries as reported in the VGPM NPP product. Heatmap shows the area-normalised density distribution of data points, while the lines show the (a) linear (shelf) and (b) non-linear (open ocean) relationships for the whole timeseries. VD bin means are plotted in green.

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
