# Peer review of "Exploring the relationship between sea ice and phytoplankton growth in the Weddell Gyre using satellite and Argo-float data"

_EGUsphere, 2023_

## Referee Comment (RC1)

**Review of "Exploring the relationship between sea-ice and primary production in the Weddell Gyre using satellite and Argo-float data"**

*Douglas et al. (2023)*

In this study, Douglas and coauthors use satellite-derived estimates of net primary production (NPP) and chlorophyll data from profiling floats to investigate the link between the seasonal evolution of biological activity in the Weddell Sea and sea-ice dynamics, as well as the ability of satellites to observe this high-latitude region. Over the observational period, the authors find a strong relationship between satellite visible days (reflecting both the absence of sea ice and a solar angle high enough to facilitate satellite observations) and annual NPP on the continental shelf, but not to the same extent in the open ocean, where the relationship weakens once a critical threshold of satellite visible days (~4 months) is reached. This reflects a strong light limitation of phytoplankton on the shelf throughout the ice-free/growing season, while light is no longer the strongest limiting factor in the open ocean towards the end of the season (likely switching to nutrient limitation, but other contributing factors such as grazing cannot be excluded based on the current study).

The paper is well-written overall, and the study is certainly **suitable for publication in *Ocean Science***. However, I see potential for improvement in the current version of the manuscript in the way chlorophyll and NPP data were integrated in the study and in the discussion of implications and shortcomings of the findings. **Overall, I therefore recommend major revisions before publication.** Addressing all major and minor points outlined in the following will hopefully substantially enhance the clarity of the manuscript.

**Main comments:**

1. In my view, in the current version of the manuscript, the authors do not make it clear enough for the reader what is gained from adding the analysis of float-derived chlorophyll data to the analysis of satellite-based NPP. The way I see it, almost all results could be obtained from using the satellite-derived estimates alone. This is reflected by the abstract of the paper, which mentions the floats in L. 5, but which does otherwise only summarize results of the analysis of satellite-based NPP. The authors should therefore re-consider the title of their paper and/or modify the content of the abstract (and the paper) to better integrate the analysis of the float data. I am not arguing to leave the float-based analysis out but think the paper would benefit from a better integration of this piece into the paper. Further, it should be made much clearer to the reader that the float-based chlorophyll estimates are not estimates of primary production (see e.g., title and L. 240). While their dynamics are certainly similar, their dynamics can be decoupled at times and in some places due to variations in community chlorophyll-to-carbon ratios and/or in loss terms. For example, in the context of this study, I am wondering if one could learn something (about biomass loss terms?) from the overall similar behavior of chlorophyll and NPP in relation to sea-ice dynamics.

2. What is lacking in the current version of the paper is a discussion of the results in the context of possible shortcomings of the underlying data, in particular data gaps in the satellite-derived estimates of NPP. One of the main findings of the authors is the overwhelming contribution of the open ocean to basin-wide estimates of NPP. Yet, at the same time, as the authors state themselves, NPP in highly productive coastal polynyas might be substantially underestimated (L. 152-156 of the submitted paper), leaving the reader wonder what the impact of this bias on all means and trends reported in the

paper might be. Given that chlorophyll appears to be detected for at least some of these polynyas (while NPP is set to zero in the CAFE model), can this information be used to somehow quantify an upper bound of NPP for the shelf region? Is the area affected on the shelves similar for all years or is there interannual variability? I realize that the main findings of the study will remain unaffected and that any attempt to quantify this bias might be associated with its own uncertainty, but the impact of this bias should at least be qualitatively discussed. Only mentioning this once in the method section is not enough in my view and might lead to a false certainty on quantitative estimates reported throughout the paper. I have highlighted several locations in my specific comments below, where I think the paper would benefit from such a discussion.

3. While the overall writing of the authors is clear and easy to follow, the writing could be improved further in several parts of the manuscript. In particular, the discussion section could be written more concisely, as it currently includes some redundancies with other sections of the paper. For example, it sometimes repeats content from the introduction, making it overall too loosely tied to the new findings of the manuscript. I think more clearly focusing on these new results and getting rid of too lengthy repetitions of the motivation of the study will make for a stronger discussion section. Similarly, a revised conclusion section should be less generic and should more clearly highlight the knowledge gain and implications from the study by Douglas and coauthors.

**Specific comments:**

L. 4: I suggest adding the time period of the analysis to the abstract.

L. 6-8: In my view, a more nuanced statement would be more accurate here given that the continental shelf region is likely affected more by e.g., data gaps, than the open ocean region (see also L. 155 of the manuscript). I realize that the open ocean will remain the dominant area whatever one would do to fill data gaps on the continental shelf but adding some upper/lower bounds (if possible) to the number (95%) or acknowledging the uncertainty of that number some other way would be helpful.

L. 12: From what I have understood in your paper, you have not ruled out grazing, have you? If true, I suggest adding "grazing" to the abstract for a more accurate representation of your discussion.

L. 18: Very minor comment: It is a bit uncommon to see a reference "*b*" before "*a*".

L. 27/30: When first reading this part, I read it as "Here is modeling studies that have looked at the relationship (REF1, REF2), but observational evidence is lacking". I realize that this is *not* what the authors have actually written but want to encourage the authors to rephrase this critical piece to make it clearer to the reader that it is *basin-scale* studies that have been missing until now. I further suggest splitting the cited references to more clearly indicate the ones that are modeling-based, float-based, satellite-based etc. Lastly, I am aware of three more references that might be relevant in this context (Briggs et al., 2017, Uchida et al., 2917, Arteaga et al., 2020; based on a non-exhaustive search).

L. 36/37: The part "will follow in the coming decades" of this sentence is a little unclear to me (will follow what exactly?) – can you rephrase to increase clarity?

L. 40: In the context of deep/bottom water formation, I would find "Weddell Sea" more accurate than "Weddell Gyre", given that the formation also involves processes other than the Gyre itself.

L. 43: Nissen et al. (2022) could be added here.

L. 48/49: It would be good to be quantitative in your comparison between rates of NPP on the continental shelf/in polynyas and in the open ocean. Can you add numbers?

Fig. 1: Given the strong focus on the role of sea ice seasonality, the authors should consider adding lines for the max/min sea ice extent in the area. From what I could see, the Southern Boundary is not referred to at all in the text and could be deleted (to make room for contours of sea ice extent). Another fairly minor point: Personally, I found it confusing that the contour line denoting the continental shelf is in a color from the colorbar of the chlorophyll map in the background. Please consider changing one of the two to avoid confusion. Lastly, please add a citation to the CAFE model to the figure caption.

L. 62: closing ")" missing

L. 63: From what I understood, "its" should refer to the Gyre here. Grammatically, it refers to "water" I think – please double-check.

L. 64: This is where I first thought that adding sea ice contours to Fig. 1 might be helpful.

L. 83: I am not sure I understand the reasoning behind using parentheses here and would advise against them for section titles.

L. 87: I suggest specifying here what these input data are (not every reader will be familiar with the model).

L. 98: corresponds *to*

L. 99/100: It might be helpful here for the reader to state more explicitly why these don't (necessarily) agree.

L. 105: I suggest moving this sentence further up to where this quantity is described. It feels out of place here.

L. 124: I suggest citing the paper by Klatt et al. (2007) here, who first suggested the use of such a criterion.

L. 127/128: Have you assessed whether the ice-free period determined this way from the floats corresponds to the ice-free period based on satellite observations in that same region? I would expect some mismatches based on the very different criteria used (see also Hague & Vichi 2021, who have used a revised sea-ice detection algorithm for floats, i.e., also including salinity).

L. 111-130: Just reading up to this section, I do not think it is sufficiently clear yet for the reader what is gained from looking at float-based chlorophyll data in addition to the satellite data. Based on the paper title and e.g., last paragraph of the introduction, the reader was set up for a study on the links between NPP and sea ice, so that the use of chlorophyll might come as a surprise here. In my view, the use of parentheses for chlorophyll in the section title does not

help emphasizing why the use of float-based chlorophyll data is useful for the study question, so I suggest clarifying this.

L. 144: I assume that for this assessment, you used the temperature criterion to determine whether a float is under ice. If this is indeed what you did, please mention that here. But coming back to an earlier comment of mine: How sensitive is this to how you define the time "under sea ice"? Given that the temperature criterion is probably a conservative criterion, I would assume a rather small sensitivity. Nonetheless, it could make sense to check this here.

L. 145: Please also mention in the text (not only in the Table) that you're assessing chlorophyll averaged over 0-20m here.

L. 152: I am not sure I fully understand: How can the smaller spatial coverage of the NPP product than of the chl-a product be due to "satellite observation limitations" if the latter is also a remote-sensing product? Can you clarify this?

L. 153: The number you give here is difficult to put into context. I suggest to also give the number relative to the whole Weddell Sea area in the text (not only in the Table).

L. 155/156: In relation to my comment on L. 152, could you elaborate on what field causes the polynyas to not show up in the NPP products and why?

L. 162: If your intention with Figure A2 was to highlight that satellite-based chlorophyll estimates often underestimate concentrations relative to float-based estimates, I suggest stating that more explicitly in the text. I do not find the current, rather generic statement of "showing similarities and dissimilarities" very helpful. Additionally, please increase the font sizes in this Figure.

L. 169: I would delete the "area-normalized" here for clarity. It took me a moment to realize that the contrast in this sentence was between *daily* and *annual*, not between *integrated* and *area-normalized*.

L. 170: In the text you say "mg C" but the y-axis label of Fig. 2 says "g C" – can you double-check which one is right?

L. 170: How large is the difference in the ice-free season? It might be of interest to the reader, so please consider adding this information.

L. 170-173: This is when I first wondered what the impact of the missing data on the shelf is. Given that you're missing NPP in coastal polynyas (see your method section), this bias might be worth mentioning, even though it might not change that much on a basin-scale.

L. 172: In the abstract, it says 95%. Should it say 99%?

L. 174-178: For all numbers in this paragraph, I suggest adding the "% of total area" as it is rather difficult to put the numbers into context otherwise.

Fig. 2: I found it a little confusing to have a legend with "IFA" in panel a, even though this quantity is only shown in panel b. Maybe move the legend to outside of panel a, e.g., on top of or below the Figure? In panel b, is the maximum IFA shown for the whole Weddell Sea? Please clarify in the y-axis label and the Figure caption. Looking at the Figure and assuming this is indeed whole-Weddell Sea-IFA, I was wondering why this is not split into open ocean and shelf

as well (for consistency). Lastly, the way NPP is currently plotted, hardly anything can be seen for NPP on the shelf. I realize that this might have been your point, but for any reader interested in the evolution of NPP on the shelf, this would be more easily visible if plotted on two separate y-axes.

L. 190-194: What you write in this paragraph is very hard to see in the Figure. Also, unless I missed something, you do not currently show the temporal evolution of sea ice on the shelf anywhere, do you? Please see my comments on Fig. 2 above for suggestions. Additionally, given your statements in the method section that you might miss NPP in coastal polynyas, have you checked in the chlorophyll and sea ice data whether there are any trends in "possibly missing NPP in polynyas" over time? Can you assume this bias to be constant in time or would it modulate any trends in NPP on the shelf reported here? I think this is an important aspect to check and report to add confidence or uncertainty to the identified trends in NPP.

L. 196: I am not sure I understand what "time" you used in the regression here. Do you mean the total duration of open water area? Please clarify.

L. 197-199: For this statement to hold, it assumes that the maximum sea ice area is far less variable over the years than the maximum ice free aera, doesn't it? I would assume this is true (and assume that you have checked this), but this could be explicitly stated for clarity.

L. 203-207: Again, the reader is left wondering what role the potential NPP underestimation due to missing NPP in (some) coastal polynyas plays for this finding. It would be helpful if you elaborated on this here (or in the discussion). Do you really think that sea ice is a less strong predictor of NPP in shelf seas or do you think that the caveats associated with the data used here complicate the comparison between the role of sea ice on the continental shelf and in the open ocean?

L. 208-213: I am not sure I find this paragraph particularly useful as it is. I found the per-pixel analysis in the subsequent paragraph much more insightful. I am wondering if the main message of this paragraph ("Over the whole open ocean, the area-normalized annual mean NPP is not correlated with mean IFA, sea-ice retreat or mean visible days, but it is on the shelf") would not be better embedded into this subsequent paragraph to improve the flow. Unless I misunderstood something, it is the breakdown of the relationship in the open ocean beyond ~120 days that also causes/contributes to the absence of correlation on a basin scale. Do you agree?

L. 211: How does this statement fit to the finding in the previous paragraph, i.e., the low correlation between the maximum ice-free area and NPP?

L. 214: a wide range *was* recorded

L. 215: Whatever you decide to do about my suggestion of the paragraph in L. 208-213, I suggest moving the definition of "visible days" to where these are first mentioned in the result section.

Fig. 5: The y-axis label might be easier to read if given in %.

Fig. 6: In the caption, please describe panel a before panel b and use the panel labels in the caption (instead of "upper panel", same goes for in the main text). Further, for panel a, please add what the whiskers, the orange line etc. represent (Median? Mean? Which percentiles?) in a

legend and in the caption. I further suggest finding clearer y-axis labels for the current panel a. For example, for "days to 50% 0-200m", I suggest specifying that you're referring to chlorophyll and that this is the bloom end. This should also become clear from the figure itself, not only from the caption and the text).

L. 231: I suggest rephrasing "after this much exposure". Maybe "after waters have been ice-free for more than 4 months"? For me, "exposure" is too unclear.

L. 231: Can you specify in what way the use of floats *deepens* the analysis? It would be helpful for the reader if you stated explicitly what can be gained from additionally analyzing the float data. The current statement is rather vague.

L. 234: I suggest rephrasing "depth restrictions" here. This made me think of the limitation that floats only sample the top 2000m – which is not the shortcoming in the context of your analysis. I assume you're referring to the fact that because they sample the top 2000m, they are not deployed in regions shallower than 2000m, i.e., the shelf regions. Can you clarify this in the text?

L. 235: If you're referring to the bottom panel first, I suggest switching the order of the panels in the figure. It is always easier for the reader if you refer to panel a first.

L. 236: It might be helpful here to add "blue lines in Fig. 6a" [or panel b, depending on whether you decide to switch the order or not] and "green lines in Fig. 6a" to the text to guide the reader.

L. 237: Do you have a reference for this definition of the bloom end? This is different from the more typical bloom definition in the literature (see e.g., Siegel et al., 2002 or Soppa et al., 2016, Hague & Vicchi 2018), and I am wondering why you did not use a criterion that makes use of the complete annual information. Please clarify.

L. 240: Please be careful here not to equate chlorophyll concentrations, i.e., a proxy for carbon biomass concentrations, with primary production. Chl:C ratios might vary substantially across the growing season, both in response to a changing light and nutrient environment and changing community composition. Further, as a proxy for biomass, chlorophyll concentrations also integrate loss terms.

L. 242-247: Related to my comment above, I am left wondering how sensitive this assessment is to how you define the bloom end. Have you tested different bloom metrics?

L. 254: What do you consider "long"? As this is subjective to the reader, I suggest being more specific here.

L. 255-262: A lot of this reads more like information for the introduction, not the discussion. Please revise to make this a more concise discussion of your findings. In general, I would always argue for the "one idea/message per paragraph" structure. In section 4.1., I have trouble identifying one message per paragraph – to me, it is the same message in each (see also section title). Maybe combine into a single paragraph?

L. 263-266: As stated above, I think it is very important to at least mention shortcomings related to data coverage in the NPP data set here.

L. 269: Why "likely"?

L. 280: Are you referring to your own work here or somebody else's with the comment in the parentheses? This is not clear to me, as I do not find this information anywhere in your result section.

L. 281: Weddell *Sea* instead of Weddell *Gyre*

L. 282: *the* instead of *a* dominant driver

L. 285: I find the formulation "provide more space" odd in this context. Can you rephrase?

L. 297: Please add "in review" or "submitted" after this reference.

L. 298: I suggest adding "for satellite detection" after "sufficient available light" to increase clarity here as this is what you show in Figure 5.

L. 305ff: Some of the references mentioned further down (L. 318/319) should be mentioned here already to point out that other authors have reported this transitioning of limiting factors in the high-latitude Southern Ocean.

L. 315-320: This feels repetitive with previous section. Consider deleting/shortening to reduce redundancies.

L. 330-335: While it is very likely that iron is indeed the limiting factor for growth, differences in grazing pressure might also play a role in explaining differences between surface and subsurface bloom dynamics of phytoplankton chlorophyll. I suggest slightly adapting the language here to represent more accurately what you can be sure about and what only appears likely.

L. 336: Do you mean phytoplankton community composition? If so, could you elaborate on how you conclude this from the chlorophyll data?

L. 355: What *warmer* areas of the Gyre are you referring to here? I suggest using geographic descriptors in this context.

L. 364: Grazer populations exert top-down control on phytoplankton communities whenever they are present, not only by late summer. I suggest rephrasing to "may dominantly control phytoplankton biomass/communities" or such.

L. 378: As stated by the authors in the abstract, this statement only holds as long as the other environmental variables do not change (nutrient availability, grazing). I suggest adding this information/assumption here.

L. 380-383: Please check for redundancy with first half of the paragraph.

L. 384: As you only infer this from your results and don't actually show it, I suggest saying "imply" instead of "indicate". I further suggest deleting "particularly", as you only infer this for the open ocean and not at all for the shelf. These changes would reflect your findings more precisely in my opinion.

L. 385: Again: unless nutrient supply changes.

L. 391: The float data do not give estimates of NPP, please be precise. Additionally, since you do not only look at sea ice but also visible days, I suggest rephrasing to something that better synthesizes what you have done.

L. 392: "It is clear" – This makes it sound like it was clear already before your study. I do agree with this interpretation (there was a body of work demonstrating the link before this paper), but I am not sure this is what you actually refer to here.

L. 394: Please add the number here instead of saying "to a high degree".

L. 397: I disagree with the authors that the float data demonstrate the iron limitation – it might seem plausible (and is probably true), but this has not been explicitly shown in this paper. Please elaborate on this or rephrase.

L. 398-405: I find a lot of these statements rather generic, and as such, they do not represent strong concluding sentences based on the results and discussion presented in this paper. I suggest re-working the conclusion section.

Figure A3: All font sizes are way too small.

**Additional references**

Arteaga, L. A., Boss, E., Behrenfeld, M. J., Westberry, T. K., & Sarmiento, J. L. (2020). Seasonal modulation of phytoplankton biomass in the Southern Ocean. *Nature Communications*, *11*(1), 5364. https://doi.org/10.1038/s41467-020-19157-2

Briggs, E. M., Martz, T. R., Talley, L. D., Mazloff, M. R., & Johnson, K. S. (2018). Physical and biological drivers of biogeochemical tracers within the seasonal sea ice zone of the Southern Ocean from profiling floats. *Journal of Geophysical Research: Oceans*, 123, 746– 758. https://doi.org/10.1002/2017JC012846

Hague, M., & Vichi, M. (2018). A Link Between CMIP5 Phytoplankton Phenology and Sea Ice in the Atlantic Southern Ocean. *Geophysical Research Letters*, *45*(13), 6566–6575. https://doi.org/10.1029/2018GL078061

Klatt, O., Boebel, O., & Fahrbach, E. (2007). A Profiling Float's Sense of Ice. *Journal of Atmospheric and Oceanic Technology*, *24*(7), 1301–1308. https://doi.org/10.1175/JTECH2026.1

Siegel, D. a, Doney, S. C., & Yoder, J. a. (2002). The North Atlantic spring phytoplankton bloom and Sverdrup's critical depth hypothesis. *Science (New York, N.Y.)*, *296*(5568), 730–733. https://doi.org/10.1126/science.1069174

Soppa, M., Völker, C., & Bracher, A. (2016). Diatom Phenology in the Southern Ocean: Mean Patterns, Trends and the Role of Climate Oscillations. *Remote Sensing*, *8*(5), 420. https://doi.org/10.3390/rs8050420

Nissen, C., Timmermann, R., Hoppema, M., Gürses, Ö., & Hauck, J. (2022). Abruptly attenuated carbon sequestration with Weddell Sea dense waters by 2100. *Nature Communications*, *13*(1), 3402. https://doi.org/10.1038/s41467-022-30671-3

Uchida, T., Balwada, D., Abernathey, R., Prend, C. J., Boss, E., & Gille, S. T. (2019). Southern Ocean phytoplankton blooms observed by biogeochemical floats. *Journal of Geophysical Research: Oceans*, 124, 7328– 7343. https://doi.org/10.1029/2019JC015355

---

## Referee Comment (RC2)

Douglas et al primarily investigates interannual fluctuations in satellite-derived NPP as they relate to sea ice variability in the Weddell Sea. The main result is that annual NPP and annual maximum ice-free area are correlated at interannual timescales. They also contrast the shelf and open ocean regions. For example, they show that in the open ocean, an increase in satellite visible days corresponds to an increase in annual NPP up to a certain point only. This presumably reflects a shift from light to nutrient limitation over the course of the growing season. I think this paper will be a useful contribution to the community, but some points need clarification before the paper is suitable for publication. In light of this, I'm suggesting major revisions. My general and detailed comments are provided below.

**General comments:**
First, I find the usage of gyre confusing in the context of this manuscript. First off, the boundaries of the study region are hydrographic transects that have nothing to do with the actual gyre dynamics. Second, the gyre is typically not thought to extend all the way onto the continental shelf, e.g. see map of mean dynamic ocean topography in Fig. 5a of Armitage et al. (2018). So the division into open ocean and shelf regions seems to apply to the Weddell Sea rather than the Weddell Gyre. I would consider replacing "Weddell Gyre" with "Weddell Sea" in the title and throughout most of the manuscript.

Second, since satellites cannot see through sea ice, it seems inevitable that the annual NPP over the entire region (as derived from satellites) will necessarily be higher when there's greater ice-free area simply because you have the ability to detect the NPP? For example, the ice-free area is correlated with the total annual NPP (Fig. 2b) but not with the area-normalized annual NPP (Fig. 2a). So isn't this suggesting that the greater annual NPP is simply due to there being more ice-free pixels (with non-zero NPP)?

A critic might argue that if there were significant under-ice NPP that is undetectable by satellite, the correlation between total annual NPP and ice-free area is an artifact related to the limitations of the satellite data. I actually don't think this is the case, since the floats show that a very small percentage of annual NPP occurs under the ice. But I think this point should still be addressed explicitly, and furthermore, this could help better integrate the float data analysis into the rest of the paper (i.e. if you frame the float analysis as a response to this imagined critic). In general, in the current version of the manuscript, the float data feels unnecessary to the main results of the paper.

Regarding the float data, I also think you need to more explicitly mention the differences between chlorophyll and NPP since it feels like they're used interchangeably at many places in the manuscript. I'm also wondering why you didn't use the POC estimates derived from the float backscatter? Backscatter-based POC is a somewhat better indicator of biomass than chlorophyll and perhaps more comparable to NPP than chlorophyll.

**Detailed comments:**
**Title:** This title doesn't convey any of the actual results of the paper, so consider changing.

**Lines 10-12:** "…additional factors such as nutrient availability **or top-down controls** limit NPP."

**Lines 30-39:** These sentences feel repetitive and are a bit hard to follow in terms of the actual writing. Consider condensing/rephrasing for clarity.

**Line 40:** I would say "The Weddell **Sea**…" rather than "The Weddell Gyre…" I think this applies to most of the manuscript (except for some other places in the Introduction that are explicitly related to the actual gyre), but I will not continue to point it out.

**Figure 1:** The Southern Boundary is hard to see on this map and I don't think it's referenced anywhere in the manuscript. Add contours showing the average annual maximum and minimum sea ice extent? There are no maps showing sea ice concentration so it's hard for the reader to visualize.

**Line 64:** Missing the closing parenthesis that starts at Line 62.

**Lines 116-119:** You haven't properly introduced the relationship of chlorophyll and NPP so this subsection feels out of place as written. Also, as I stated in my general comments, why not look at the POC derived from backscatter?

**Lines 125-127:** It's worth mentioning that float timeseries reflect both temporal and spatial variability. The language here implies that the floats can be treated as the timeseries of a bloom at a particular location, but this may or may not be the case given the small decorrelation length scales for chlorophyll.

**Lines 150-152:** I don't understand this statement, which input data have less extensive spatial coverage than Chl-a?

**Lines 154-156:** This seems like a limitation to the partitioning of total NPP on the shelf vs open ocean, which is framed as one of the main results of the paper. Obviously there's not much that can be done to address this, but it feels like it should at least be discussed later on in the paper.

**Lines 170-173:** You should mention explicitly that the area of the open ocean is significantly larger than the shelf, which seems to be dominating the partitioning of the total annual NPP between the two regions.

**Line 172:** The abstract says 95%, but here it says 99%.

**Lines 186-188:** Consider discussing some of the relevant forcings that drive interannual variability of NPP? This entire subsection is very descriptive, and you don't really discuss any of the mechanisms at play. I realize that you go into depth on the drivers in the Discussion section, but at least a sentence or two mentioning some of the controls on NPP might help the reader.

**Lines 190-191:** As I said above, some discussion of the mechanisms feels absent. Why might NPP on the shelf be declining? Speculation is fine, but I think some mention of the underlying dynamics is helpful. Otherwise the reader is left wondering whether this trend is just due to aliasing associated with the limitations of the satellite data on the shelf.

**Lines 203-204:** Is this the yearly maximum IFA over the entire region or over the sub-regions separately? Because the area of the open ocean is so much larger than the area of the shelf, so the yearly maximum IFA over the entire region will be dominated by changes in the open ocean. In other words, if you're considering the yearly maximum IFA over the whole region, this could lead to a smaller correlation with the NPP on the shelf (compared to if you used the yearly maximum IFA on the shelf). It just seems strange to me that sea ice would be less important on the shelf.

**Lines 229-232:** I think more could be done to introduce the objectives of the float data analysis so that it feels better integrated with the rest of the paper.

**Lines 236-237:** Where does this definition of bloom end come from?

**Lines 284-285:** Larger areas of ice-free water also provide more space for satellites to detect NPP. As I said in my general comment, I think you should use the float data as evidence that there is not significant NPP occurring underneath the ice, so that you can rule out the possibility that the correlation between ice-free area and NPP is not simply due to the greater number of pixels with non-zero NPP since satellite can't see through the ice.

**Line 286:** I know you cite it later on, but some discussion of Moreau et al. (2023) seems warranted in this paragraph.

**Line 297:** add "in review" for this reference and also link to the preprint in the References section at the end of the paper.

**Lines 335-337:** Can you elaborate on how float data show differences in type/composition?

**Lines 378-380:** I don't understand this statement? Why would a region becoming permanently ice-free cause NPP to decrease? Are you suggesting that the sea ice is an important source of iron to the system? Or that freshwater fluxes associated with sea ice melt/refreeze are important in setting the stratification that favors growth? Give some possible mechanism because "by analogy to the permanently open ocean regions in the present-day Southern Ocean" is not very convincing since it's not clear what regions you're even referring to. There are many sources of variability besides just ice vs. no ice that lead to heterogeneity in NPP.

**Lines 390-405:** I found the Conclusions section to be a bit weak and I suggest rewriting. Some of the statements are well-known from existing literature (e.g. it is clear that sea-ice dynamics are important in driving NPP in this region), while other statements are speculative and don't stem from the actual analysis conducted (e.g. substantial spatial variability undoubtedly contributes

to the variance in NPP…). As a result, the reader is left feeling uncertain about what contribution has been made by this study.

Channing Prend

**References**

Armitage, T.W.K., R. Kwok, A.F. Thompson, & G. Cunningham (2018). Dynamic topography and sea level anomalies of the Southern Ocean: Variability and teleconnections. *Journal of Geophysical Research: Oceans*, **123**, 613-630.

---

## Author Comment (AC1)

**Review of "Exploring the relationship between sea-ice and primary production in the Weddell Gyre using satellite and Argo-float data"**
**Douglas et al. (2023)**

*We are grateful to the two reviewers for their positive responses and constructive suggestions and comments. In the following,* reviewers' comments are in regular typeface *and our responses are in blue italics.*

In this study, Douglas and coauthors use satellite-derived estimates of net primary production (NPP) and chlorophyll data from profiling floats to investigate the link between the seasonal evolution of biological activity in the Weddell Sea and sea-ice dynamics, as well as the ability of satellites to observe this high-latitude region. Over the observational period, the authors find a strong relationship between satellite visible days (reflecting both the absence of sea ice and a solar angle high enough to facilitate satellite observations) and annual NPP on the continental shelf, but not to the same extent in the open ocean, where the relationship weakens once a critical threshold of satellite visible days (~4 months) is reached. This reflects a strong light limitation of phytoplankton on the shelf throughout the ice-free/growing season, while light is no longer the strongest limiting factor in the open ocean towards the end of the season (likely switching to nutrient limitation, but other contributing factors such as grazing cannot be excluded based on the current study). The paper is well-written overall, and the study is certainly suitable for publication in Ocean Science. However, I see potential for improvement in the current version of the manuscript in the way chlorophyll and NPP data were integrated in the study and in the discussion of implications and shortcomings of the findings. Overall, I therefore recommend major revisions before publication. Addressing all major and minor points outlined in the following will hopefully substantially enhance the clarity of the manuscript.

**Main comments:**
1. In my view, in the current version of the manuscript, the authors do not make it clear enough for the reader what is gained from adding the analysis of float-derived chlorophyll data to the analysis of satellite-based NPP. The way I see it, almost all results could be obtained from using the satellite-derived estimates alone. This is reflected by the abstract of the paper, which mentions the floats in L. 5, but which does otherwise only summarize results of the analysis of satellite-based NPP. The authors should therefore re-consider the title of their paper and/or modify the content of the abstract (and the paper) to better integrate the analysis of the float data. I am not arguing to leave the float-based analysis out but think the paper would benefit from a better integration of this piece into the paper.
*We are grateful for the reviewer's request here and in the specific comments to better integrate the float data into the paper. The sentiment here, concerning better integration of the float data in the manuscript, is shared by both reviewers. We therefore broadly address their comments here, and refer back to this when addressing specific reviewer comments. The floats are an important component of this work for two reasons: 1) They allow us to assess the uncertainty in the satellite data (namely in quantifying what the satellite misses due to sea-ice cover and low solar angle). 2) They allow us to observe the seasonal progression in Chl-a and calculate a*

*quantitative timeline of activity from ice melt through the growing season, providing a complementary perspective to that gained from the satellite data.*

*As an overview of the changes we have made:*
1) *Altered the title to better encompass the use of Chl-a and POC*
2) *The abstract has been modified to integrate the float results more explicitly.*
3) *A sentence explaining the inclusion of float data has been added to the beginning of the autonomous floats methods section.*
4) *The uncertainties section has been modified to improve flow and clarity.*
5) *A paragraph has been added to the satellite results section to emphasise uncertainties in the data and lead into the floats results, emphasising the importance of the addition of float data to this paper.*
6) *The importance of the float data in supporting and expanding on the satellite-based results are also emphasised in the discussion*

*Details:*
1) *The title has been changed to:*

> *Exploring the relationship between sea ice and phytoplankton growth in the Weddell Gyre using satellite and Argo-float data*

2) *Abstract:*

> *Some of the highest rates of primary production across the Southern Ocean occur in the seasonal ice zone (SIZ), making this a prominent area of importance for both local ecosystems and the global carbon cycle. There, the annual advance and retreat of ice impact light and nutrient availability, as well as the circulation and stratification, thereby imposing a dominant control on phytoplankton growth. In this study, the drivers of variability in phytoplankton growth between 2002-2020 in the Weddell Gyre SIZ were assessed using satellite net primary production (NPP) products alongside chlorophyll-a and particulate organic carbon (POC) data from autonomous biogeochemical floats. Although the highest daily rates of NPP are consistently observed in the continental shelf region (water depths shallower than 2000m), the open ocean region's larger size and longer ice-free season mean that it dominates biological carbon uptake within the Weddell Gyre, accounting for 93-96% of the basin's total annual NPP. Variability in the summer maximum ice-free area is the strongest predictor of inter-annual variability in total NPP across the Weddell Gyre, with greater ice-free area resulting in greater annual NPP, explaining nearly half of the variance ($R^2$=42%). In the shelf region, the return of sea-ice cover controls the end of the productive season. In the open ocean, however, both satellite NPP and float data show that a decline in NPP occurs before the end of the ice-free season (~ 80 to 130 days after sea-ice retreat). Evidence of concurrent increases in float-observed chlorophyll-a and POC suggest that, later*

*in the summer season, additional factors such as micro-nutrient availability or top-down controls (e.g. grazing) could be limiting NPP. These results indicate that in a warmer and more ice-free Weddell Gyre, notwithstanding compensating changes in nutrient supply, NPP is likely to be enhanced only up to a certain limit of ice-free days.*

3) *Autonomous Floats methods section now starts with an introduction as to why we include analysis of float data and it's value in supporting and expanding on the satellite results (lines 134 - 136):*

    *We use BGC-Argo float data to evaluate the data recovery attributes of satellite data, estimate associated uncertainties (Section 2.4), and also to assess the seasonal progression of phytoplankton growth in the water column, using Chl-a as a proxy for photosynthetic potential and particulate organic carbon (POC) as a proxy for biomass.*

4) *We have substantially reordered and expanded our uncertainty section to include more explicit reasoning for the value added by the float data. Additionally, we have included POC measurements to partially bridge the gap between chlorophyll and primary production. The uncertainties section now reads (lines 172 to 208):*

    *Without comprehensive* in situ *measurements of NPP, it is not possible to directly quantify what is missed by satellites. However, BGC-Argo float data can be used to provide insight into these limitations, as they provide sub-surface, year-round, under-ice observations of key parameters related to NPP. Floats data have their own limitations - floats are Lagrangian autonomous observing platforms, so observations reflect both temporal and spatial variability. Additionally, sensor calibrations may vary and sensors sometimes drift towards the end of the float deployment. We did not attempt to estimate water-column integrated NPP from float data, as the floats in the study region lacked PAR (Photosynthetically Active Radiation) sensors, and, as far as we are aware, there are not yet methods for calculating NPP from float data that have been robustly validated for widespread use. Instead, estimates of Chl-a and POC can be used as conservative proxies for phytoplankton growth/biomass to assess the importance of missed phytoplankton. While Chl-a does not directly equate to NPP, satellite Chl-a is used in all satellite NPP products, and as such we can compare float and satellite Chl-a observations. Figure \ref{fig:chlmatchup} highlights the similarities and dissimilarities in float and satellite Chl-a estimates. The absolute values of Chl-a differ between the satellite and the float because of the way Chl-a is derived from observations on these platforms - satellite Chl-a is derived from reflectance, whereas fluorescence is used to calculate float-observed Chl-a. Nonetheless, the relative changes in Chl-a over the timeseries should be seen in both datasets. In most cases, the satellite Chl-a at each float location peaks and troughs at the same time as the float Chl-a. On some occasions, the spring peak/bloom is not*

*fully seen by the satellites, and in several cases (Figure \ref{fig:chlmatchup}: b) and c) in 2018, f) and k) in 2020), it is missed entirely. As such, the addition of float data in this study expands on the results derived from the satellite products and allows assessment of the seasonal patterns in Chl-a. Notably, it is clear that the satellite Chl-a product does not span the entire ice-free/growing season and the floats are valuable in providing a year-round perspective on the seasonal changes in Chl-a and POC.*

*We use float data to determine how much of the annual-integrated Chl-a and POC (in the surface 20m) is missed by satellites, reasoning that this offers a reasonable estimate for the fraction of missed NPP. The proportions of annually integrated Chl-a and POC measured by each float observed when the float was (1) under sea ice (using the Klatt et al., 2007 temperature criterion mentioned above) and (2) when the solar angle is less than 20°, limiting satellite view at the end of summer (from mid-March) are summarised in Table 1. At the float locations, a small proportion of annually integrated Chl-a is present under sea ice (median 10%, range 2-23%), while up to 19% (median 9%) of annually integrated Chl-a is observed by floats during the time when the ocean-colour satellite was unable to view the surface ocean due to low solar angle (Table 1). Up to 30% of the annually integrated POC is recorded under-ice (minimum 7%, median 19%) and an average of 12% (range 4-20%) of the surface POC is recorded after the solar angle is too low.*

*Ocean-colour satellites are only able to view the surface of the ocean, meaning Chl-a satellite products do not represent the full euphotic zone/water column inventory (Pope et al., 2017). The CAFE algorithm applies a light saturation model for the mixed layer and assumes a co-varying relationship between the phytoplankton absorption coefficient and light saturation below the mixed layer in order to estimate the vertical structure of Chl-a (and NPP; Silsbe et al., 2016). As a result, the subsurface Chl-a inventory may not be accurately resolved and may also miss subsurface Chl-a maximum layers, which contribute substantially to productivity and promote enhanced carbon export in the SO (Baldry et al., 2020). Again, floats enable assessment of the importance of phytoplankton missed by satellites, in this case, below 20m. The depth integrated Chl-a signal seen in Figure 2 declines later than the surface Chl-a, indicating that concentrations of Chl-a remain elevated at depth for longer than at the surface.*

5) *A paragraph has been added to the satellite results section to emphasise uncertainties in the data and lead into the floats results, emphasising the importance of the addition of float data to this paper:*

   *As described in Section. 2.4 and summarised in Table. 1, BGC Argo float estimates of Chl-a and POC both under ice and at low sun angle suggest that substantial amount of phytoplankton biomass may be missed by satellite NPP*

*products in the open ocean (medians of 20% Chl and 39% POC for all float locations). We did not attempt to further adjust our NPP estimates for these missed data, because float biomass proxies are not directly proportional to NPP and because float coverage, while broad, was not even throughout the Weddell Gyre.*

*6) The importance of the float data in supporting and expanding on the satellite-based results are also emphasised in the discussion:*

*The weakening/plateau of the relationship between the number of visible days and NPP (Fig. 6.b) indicates the exertion of a limiting control, such as nutrient availability or top-down controls, mediating NPP towards the end of the growing season. The sub-surface float data support this finding: For the majority of float years, a decline in Chl-a occurs before the solar angle is below critical (20°) in March, and well in advance of the return of sea ice to the float's location (Fig. 7). The average timing of this decline (73 days at the surface) is broadly consistent with what is seen in the satellite results. The implication is that the decline in growth after 2-3 months of ice-free conditions leads to a suppression in annually integrated NPP in areas where the ice-free season lasts longer than approximately 3-4 months.*

Further, it should be made much clearer to the reader that the float-based chlorophyll estimates are not estimates of primary production (see e.g., title and L. 240). While their dynamics are certainly similar, their dynamics can be decoupled at times and in some places due to variations in community chlorophyll-to-carbon ratios and/or in loss terms. For example, in the context of this study, I am wondering if one could learn something (about biomass loss terms?) from the overall similar behavior of chlorophyll and NPP in relation to sea-ice dynamics.

*We chose to look at Chl-a from the floats because we could compare it to the satellite Chl-a observations. We have now made it clearer in the text that Chl-a is used here as a proxy for growth, and that it does not equate to estimates of NPP. In our description of the CAFE model, we state that Chl-a is used in the model to calculate NPP. Additionally, at the start of the "Autonomous Floats" methods section, we state that we are "using Chl-a as a proxy for growth". We have also now added analysis of particulate organic carbon (discussion of results lines 427-447) (estimated from backscatter float data) as an estimate/proxy for biomass. We have calculated Chl-a:POC ratios to improve our interpretation of the seasonal progression in phytoplankton activity.*

2. What is lacking in the current version of the paper is a discussion of the results in the context of possible shortcomings of the underlying data, in particular data gaps in the satellite-derived estimates of NPP. One of the main findings of the authors is the overwhelming contribution of the open ocean to basin-wide estimates of NPP. Yet, at the same time, as the authors state themselves, NPP in highly productive coastal polynyas might be substantially underestimated (L. 152-156 of the submitted paper), leaving the reader wonder what the impact of this bias on all means and trends reported in the paper might be. Given that chlorophyll appears to be

detected for at least some of these polynyas (while NPP is set to zero in the CAFE model), can this information be used to somehow quantify an upper bound of NPP for the shelf region?

Is the area affected on the shelves similar for all years or is there interannual variability?

I realize that the main findings of the study will remain unaffected and that any attempt to quantify this bias might be associated with its own uncertainty, but the impact of this bias should at least be qualitatively discussed. Only mentioning this once in the method section is not enough in my view and might lead to a false certainty on quantitative estimates reported throughout the paper. I have highlighted several locations in my specific comments below, where I think the paper would benefit from such a discussion.

*We have followed this suggestion and added a substantial analysis of the impact of potential satellite NPP coverage gaps on our estimates of the relative contribution of coastal regions to annual NPP. Upon further investigation into the data used to calculate the CAFE NPP product, we found that the absorption and backscatter input variables have reduced spatial coverage compared to Chl-a and the other remaining variables used in the CAFE model. In particular, absorption and backscatter coverage is sparse during times of rapid sea ice retreat, and declines in coverage/quality from 2010 onwards. There is inter-annual variability in the area that is affected on the shelves: The proportion of ice-free water in the shelf region that is visible in the CAFE model ranges from 0-65% seasonally (mean 10%). Furthermore, there is a declining trend seen in the proportion of ice-free water that is visible on the shelf in the CAFE NPP product (2.6 % decline per year, p = 0.001). This trend in the bias is not present in the other NPP models. As a result, it is uncertain whether the declining trend in NPP seen in CAFE on the shelf is robust. A weak(er) decreasing trend (smaller slope and $R^2$) in raw annual NPP on the shelf is seen in the VGPM and VGPM-Eppley models ($R^2$=0.2, p = 0.036 for both), but not in the CBPM model ($R^2$=0.16, p = 0.058). In the open ocean region, the spatial coverage of CAFE is, at times (mainly in December), lower than the other NPP products due to the sparseness of the absorption and backscatter coverage, but at the time of peak IFA, the CAFE product sees over 90% of the IFA.*

*We continue to use CAFE as the main NPP model, as it is the most comprehensive model available on the Ocean Productivity website and robustly captures the baseline productivity in the Southern Ocean (Silsbe et al., 2016, Ryan-Keogh, preprint, Westberry et al., 2023). The shelf region clearly suffers from problems in data availability and trends in errors, so in the paper, we now refer to the supplementary figures which show the results from the other models and demonstrate that the main results remain consistent across the NPP models.*

*We have also introduced a method to account for these data gaps, to generate improved estimates of annually-integrated NPP, reporting them alongside the original values in the Fig.2 bar charts to show the effect of data coverage on NPP values.*

*This is achieved by assuming that the NPP in the areas missed in the CAFE model can be represented as the average NPP reported in the visible areas of that region for each timepoint. This approach thus provides an estimate of how much NPP could be occurring across the entire ice-free area instead of just the area visible in the NPP product.  In the CAFE model, total annual NPP on the shelf increases from 2 +/- 2 Tg C to 11 +/- 5 Tg C and in the open ocean, increases from 170 +/-33 Tg C to 255 +/- 38 Tg C. With these improved estimates, the*

*percentage contribution of the shelf region remains small: 4 +/- 2 % – with the open ocean accounting for 96 +/- 2 % of the annual NPP in the Weddell Sea. Although the absolute values of NPP in the other models differ, the percentage contribution of the shelf and open ocean regions to the Weddell Gyre study area are consistent with the CAFE results (mean contribution ranges from 93-96% and the range across all years in all models spans 87-99%). The open ocean region accounts for 85% of the study region. As such, even when taking into account uncertainties in NPP estimates, our result - that the open ocean contributes a disproportionately proportion of the Weddell Gyre NPP - stands. Imputed and directly observed estimates of NPP are included in the main text from line 230 - 245.*

*The following was added/expanded on in the Uncertainties section:*

> *In addition to missing satellite NPP data under ice and at low solar zenith angle, there are further empty grid points in the mapped NPP and Chl-a data that mean that the IFA does not equate to the area of the NPP or Chl-a products where data is recorded. NPP in these areas is considered to be zero, leading to an underestimate of the regions' total NPP. Some of these gaps could be attributed to the "adjacency effect" along the ice edge and also where the cloud-fill algorithm did not complete successfully. In addition to this, there is also a disparity in the spatial coverage of the NPP products and the Chl-a input data used to derive CAFE NPP (Table 1). Some of the input data (absorption due to gelbstoff and detritus, absorption due to phytoplankton and backscatter spectral slope parameter) used in the CAFE algorithm to derive NPP have less extensive spatial coverage than the Chl-a input data. This means that there are some areas in the NPP product that imply there is no NPP occurring despite Chl-a being observed by the satellite. Over the timeseries, between 10-100% (mean 78%, median 85%) of the area with Chl-a data also has coverage in the CAFE NPP product (lowest values occur as the satellite view declines at the start of winter). The spatial coverage of the CAFE product in relation to IFA ranges between 0-94% over the seasonal cycle. When ice-free and with sufficient solar elevation for ocean-colour observations, on average 47% of the IFA was visible in the CAFE product. This difference between the IFA and area where NPP is observed in the CAFE product is most acute in the shelf region, where 0-65% (mean 10%) of IFA has NPP data when conditions are suitable for ocean-colour observations, and many coastal polynyas that are visible in the input Chl-a data are not translated into the NPP data. Therefore shelf integrated NPP values are expected to be significantly underestimated. To estimate the potential total NPP occurring across the entire IFA in both the shelf and open ocean regions, data gaps were filled with the mean daily rate for each 8-day time-step within a region. As with the original calculation, the total annual NPP is then calculated by integrating the new 8-day totals for the austral year. The difference between the directly observed (raw) and imputed (gap-filled) estimates can be seen in Figure 3.b-d).*

3. While the overall writing of the authors is clear and easy to follow, the writing could be improved further in several parts of the manuscript. In particular, the discussion section could be written more concisely, as it currently includes some redundancies with other sections of the paper. For example, it sometimes repeats content from the introduction, making it overall too loosely tied to the new findings of the manuscript. I think more clearly focusing on these new results and getting rid of too lengthy repetitions of the motivation of the study will make for a stronger discussion section. Similarly, a revised conclusion section should be less generic and should more clearly highlight the knowledge gain and implications from the study by Douglas and coauthors.

*We have removed areas of repetition/redundant sentences following specific comments from both reviewers. We have also adapted the conclusions to better summarise the results from this study.*

**Specific Comments**
L. 4: I suggest adding the time period of the analysis to the abstract.
*"Between 2002-2020" has been added*

L. 6-8: In my view, a more nuanced statement would be more accurate here given that the continental shelf region is likely affected more by e.g., data gaps, than the open ocean region (see also L. 155 of the manuscript). I realize that the open ocean will remain the dominant area whatever one would do to fill data gaps on the continental shelf but adding some upper/lower bounds (if possible) to the number (95%) or acknowledging the uncertainty of that number some other way would be helpful.
*Our assessment, including quantification of uncertainties associated with satellite spatial coverage, finds that across all NPP models, the mean open ocean annual NPP contribution to the Weddell Sea ranged between 93-96% and the range across all years in all models spans 87-99%.*

L. 12: From what I have understood in your paper, you have not ruled out grazing, have you? If true, I suggest adding "grazing" to the abstract for a more accurate representation of your discussion
*"or top-down controls (grazing)" has been added*

L. 18: Very minor comment: It is a bit uncommon to see a reference "b" before "a".
*To the Editor: This seems to be an automatic citation issue associated with the Ocean Sciences template, can it be sorted in post-processing?*

L. 27/30: When first reading this part, I read it as "Here is modeling studies that have looked at the relationship (REF1, REF2), but observational evidence is lacking". I realize that this is not what the authors have actually written but want to encourage the authors to rephrase this critical piece to make it clearer to the reader that it is basin-scale studies that have been missing until now. I further suggest splitting the cited references to more clearly indicate the ones that are modeling-based, float-based, satellite-based etc. Lastly, I am aware of three more references

that might be relevant in this context (Briggs et al., 2017, Uchida et al., 2917, Arteaga et al., 2020; based on a non-exhaustive search).

*We appreciate the reviewer highlighting this point of confusion and these papers. The text now reads as:*

> *Sea-ice dynamics play a critical role in primary production in the SIZ by attenuating light and altering stratification, mixing, and nutrient delivery (McGillicuddy et al., 2015; Gupta et al., 2020; Twelves et al., 2021). Many observational (e.g. Arteaga et al., 2020; Bisson and Cael, 2021; Hague and Vichi, 2021; von Berg et al., 2020; Giddy et al., 2023; Briggs et al., 2018; Uchida et al., 2019; McGillicuddy et al., 2015) and model-based (e.g. Schultz et al., 2021; Taylor et al., 2013; Briggs et al., 2018; Twelves et al., 2021; McGillicuddy et al., 2015; Gupta et al., 2020) studies have enhanced our understanding of these impacts on regional (SO-wide) scales and at small, local scales. However, our understanding of the spatiotemporal relationship between sea ice and net primary production (NPP) on basin-scales is lacking.*

L. 36/37: The part "will follow in the coming decades" of this sentence is a little unclear to me (will follow what exactly?) – can you rephrase to increase clarity?

*We appreciate the reviewer requesting clarity. Following additional request from reviewer 2 to condense this paragraph, the paragraph now reads as:*

> *Climate models from Coupled Model Intercomparison Project Phase 5 and 6 (CMIP5 and CMIP6) project a decline in Antarctic sea-ice area and concentration as a response to anthropogenic climate change (Casagrande et al., 2023). However, low confidence in projections, due to the complexity of ocean-ice-atmosphere systems, means that exact estimates of decline are uncertain (Casagrande et al., 2023; Meredith et al., 2019). Therefore, in light of these observed and anticipated changes in the climate of the SIZ (Kumar et al., 2021; Ludescher et al., 2019; Casagrande et al., 2023), the need for a deeper understanding of the relationship between sea ice and NPP is pressing, as changes in sea ice will have concomitant impacts on carbon uptake and ecosystem health. However, the crucial gaps in our understanding of the drivers of NPP in the SIZ mean that large uncertainty remains about the nature and extent of these changes (Campbell et al., 2019; Henley et al., 2020; Kim and Kim, 2021; Pinkerton et al., 2021; Séférian et al., 2020; Henson et al., 2022).*

L. 40: In the context of deep/bottom water formation, I would find "Weddell Sea" more accurate than "Weddell Gyre", given that the formation also involves processes other than the Gyre itself.

*We have changed this instance of Weddell Gyre to Weddell Sea to better reflect the area where deep/bottom water formation occurs.*

L. 43: Nissen et al. (2022) could be added here.

*Reference added*

L. 48/49: It would be good to be quantitative in your comparison between rates of NPP on the continental shelf/in polynyas and in the open ocean. Can you add numbers?
*Numbers have been added and the language modified. The text now reads:*

> *However, the offshore (open ocean) area of the Weddell Sea has been shown to contribute greatly to the region's annual primary production, with annual NPP in the offshore (open ocean) marginal ice zone (73.7 g C m$^{-2}$ a$^{-1}$) exceeding that seen in the shelf marginal ice zone (65.2 g C m$^{-2}$ a$^{-1}$ ), and almost matching the annual NPP exhibited in the shelf region (77.0 g C m$^{-2}$ a$^{-1}$) (Arrigo et al., 2008). In this way, the Weddell Sea marginal ice zone was identified as the largest and most productive marginal ice zone of all the geographic sectors investigated (25% greater than the Ross Sea marginal ice zone; Arrigo et al., 2008).*

Fig. 1: Given the strong focus on the role of sea ice seasonality, the authors should consider adding lines for the max/min sea ice extent in the area. From what I could see, the Southern Boundary is not referred to at all in the text and could be deleted (to make room for contours of sea ice extent). Another fairly minor point: Personally, I found it confusing that the contour line denoting the continental shelf is in a color from the colorbar of the chlorophyll map in the background. Please consider changing one of the two to avoid confusion. Lastly, please add a citation to the CAFE model to the figure caption.
*The Silsbe et al. 2016 CAFE citation has been added to figure caption. Lines to indicate the maximum and minimum sea ice extent have been added, the shelf region line was changed to orange and the open ocean to dark blue and the Southern Boundary line was removed. The figure now appears as:*

[Figure]

*Figure 1. Location of the Weddell Gyre and study subregions (dark blue: open ocean; orange: shelf). The 18-year mean area-normalised annual NPP (g C m-2) climatology derived from MODIS-Aqua satellite measurements using the Carbon, Absorption and Fluorescence Euphotic-resolving (CAFE) model (Silsbe et al., 2016) is represented by the yellow to green colourmap. White areas represent no data (permanent sea ice present). SOCCOM float trajectories from 11 BGC-Argo floats are shown in greyscale and labelled with WMO ID. Profiles from float SD5905991 located north of the study region were not included in the analysis. The dashed and solid light blue lines denote the summer minimum (March 2006) and winter maximum (September 2010) sea ice extent respectively.*

L. 62: closing ")" missing
*Closing ")" has been added*

L. 63: From what I understood, "its" should refer to the Gyre here. Grammatically, it refers to "water" I think – please double-check.
*"its" has been changed to "the Gyre's"*

L. 64: This is where I first thought that adding sea ice contours to Fig. 1 might be helpful.
*A reference to Fig 1 has been added.*

L. 83: I am not sure I understand the reasoning behind using parentheses here and would advise against them for section titles.
*Parentheses have been removed*

L. 87: I suggest specifying here what these input data are (not every reader will be familiar with the model).

*The paragraph has been modified to read:*

> *Satellite-derived NPP products calculated from MODIS (Moderate-Resolution Imaging Spectroradiometer)-Aqua chlorophylla (Chl-a) data were obtained from the Ocean Productivity website (www.science.oregonstate.edu/ocean.productivity/ ; Oregon State University, 2019). We use the NPP product derived using the Carbon, Absorption and Fluorescence Euphotic-resolving (CAFE) model (Silsbe et al., 2016; Westberry and Behrenfeld, 2013). The CAFE model is an absorption-based model that considers the amount of energy absorbed by phytoplankton, photoacclimation, and the efficiency with which energy is converted into biomass. CAFE was chosen over other NPP models as it is currently the most comprehensive open-source NPP model and is the most robust in capturing the baseline productivity of the SO (Silsbe et al., 2016; Ryan-keogh et al., 2023; Westberry et al., 2023). The CAFE NPP product is calculated using MODIS-derived products: Chl-a, PAR (photosynthetically available radiation), sea surface temperature; MODIS-derived absorption and backscattering variables configured using the Generalized Inherent Optical Property (GIOP) model; and mixed layer depth provided by the HYbrid Coordinate Ocean Model (HYCOM). Input data to the CAFE model are cloud-filled via spatial and temporal interpolation prior to calculating NPP. The resolution of the NPP data is 8-day averages on a 1/12° latitude-longitude grid (2-4km zonal resolution, 8-10km meridional resolution in this region) for the period 2002-2020. Cloud-filled MODIS-Aqua Chl-a concentration data were also obtained at the same resolution as the NPP data to later compare to BGC-Argo float data as proxies for growth. Three other NPP models were analysed to assess the robustness of the CAFE results. The Carbon-based Productivity Model (CbPM; Westberry et al., 2008; Behrenfeld et al., 2005) relates NPP to phytoplankton carbon biomass (derived from particulate backscatter received by MODIS-Aqua). Both the Vertically Generalized Production Model (VGPM; Behrenfeld and Falkowski, 1997) and Eppley-adjusted VGPM (eppley; Eppley, 1972) products are calculated based on the relationship between chlorophyll and temperature derived growth rates. Eppley differs from VGPM by parameterising the relationship between temperature and phytoplankton growth rate using the positive exponential function described by Eppley (1972); Morel (1991).*

L. 98: corresponds to.
*"to" has been added*

L. 99/100: It might be helpful here for the reader to state more explicitly why these don't (necessarily) agree.

*The reason for a difference in the IFA and the visible area has been added, with a reference to the uncertainty section for further detail. The sentence now reads as:*

> *Note that this area is not exactly equivalent to the IFA derived from the sea-ice data because of the "adjacency effect" of the ice edge on restricting ocean colour data near 120 sea ice (Pope et al., 2017) as well as areas where there are data missing for other reasons (cloud-fill algorithm, input data: See Section 2.4)*

L. 105: I suggest moving this sentence further up to where this quantity is described. It feels out of place here.

*The sentence that started on L105 (about the annual means of the spatially-averaged NPP) was adapted to introduce the definition of the austral year and moved to the end of the previous paragraph (now lines 121-122).*

L. 124: I suggest citing the paper by Klatt et al. (2007) here, who first suggested the use of such a criterion.

*Citation has been added*

L. 127/128: Have you assessed whether the ice-free period determined this way from the floats corresponds to the ice-free period based on satellite observations in that same region? I would expect some mismatches based on the very different criteria used (see also Hague & Vichi 2021, who have used a revised sea-ice detection algorithm for floats, i.e., also including salinity).

*While this is an interesting suggestion, the under-ice threshold is a decision made by the BGC-Argo community and investigating this in detail is beyond the scope of this study. We acknowledge that there is uncertainty in the exact timing when the waters become ice-free at float locations. However profiles that are considered under-ice by the ice detection criterion will not have an accurate location to compare to the satellite observations, and so uncertainty will also arise from this.*

L. 111-130: Just reading up to this section, I do not think it is sufficiently clear yet for the reader what is gained from looking at float-based chlorophyll data in addition to the satellite data. Based on the paper title and e.g., last paragraph of the introduction, the reader was set up for a study on the links between NPP and sea ice, so that the use of chlorophyll might come as a surprise here. In my view, the use of parentheses for chlorophyll in the section title does not help emphasizing why the use of float-based chlorophyll data is useful for the study question, so I suggest clarifying this.

*We appreciate the request for clarity and acknowledge that our explanation for including float data in order to assess uncertainties associated with satellite data appears at the end of the methods, after the float data has already been introduced.*

- *The satellite NPP and Chl-a section more explicitly introduces Chl-a as one of the variables used to calculate NPP:*

*The CAFE NPP product is calculated using MODIS-derived products: Chl-a, PAR (photosynthetically available radiation), sea surface temperature; MODIS-derived absorption and backscattering variables configured using the Generalized Inherent Optical Property (GIOP) model; and mixed layer depth provided by the HYbrid Coordinate Ocean Model (HYCOM). Input data to the CAFE model are cloud-filled via spatial and temporal interpolation prior to calculating NPP. The resolution of the NPP data is 8-day averages on a 1/12° latitude-longitude grid (2-4km zonal resolution, 8-10km meridional resolution 105 in this region) for the period 2002-2020. Cloud-filled MODIS-Aqua Chl-a concentration data were also obtained at the same resolution as the NPP data to later compare to BGC-Argo float data as proxies for growth.*

- *We have also added a sentence to the start of the Autonomous Floats methods section to introduce the reasons for including float data in our study:*

  *We use BGC-Argo float data to evaluate the data recovery attributes of satellite data, estimate associated uncertainties (Section 2.4), and also to assess the seasonal progression of phytoplankton growth in the water column, using Chl-a as a proxy for photosynthetic potential and particulate organic carbon (POC) as a proxy for biomass*

- *We have also adjusted the first sentence in the final paragraph of the introduction to read:*

  *Given the importance of biological carbon uptake in the Weddell Gyre for local ecosystems and the global carbon cycle, this paper characterises and quantifies the basin-scale relationship between sea ice and phytoplankton growth using both satellite net primary production (NPP) products and in situ observations of chlorophyll-a (Chl-a) and particulate organic carbon (POC) from Biogeochemical (BGC) Argo floats.*

L. 144: I assume that for this assessment, you used the temperature criterion to determine whether a float is under ice. If this is indeed what you did, please mention that here.
*This has now been mentioned.*
But coming back to an earlier comment of mine: How sensitive is this to how you define the time "under sea ice"? Given that the temperature criterion is probably a conservative criterion, I would assume a rather small sensitivity. Nonetheless, it could make sense to check this here.
*Thank you for the question. Given that the temperature criterion is conservative to ensure floats don't surface into ice, we could assume the possibility that a float surfaces several profiles after sea-ice has retreated above the float location. We investigated the sensitivity of the "under sea-ice" definition by setting the last 2 profiles that the float considered ice-covered as ice-free (assuming a potential uncertainty of 1 - 2 weeks (or 1 - 2 profiles). We calculated the annual Chl-a present under-ice assuming the sea-ice had retreated up to 20 days (2 profiles) earlier than the surfacing of the float. Across all float years, the median Chl-a under-ice decreased from*

*10 % to 7 % and the maximum seen decreased from 17 % to 15 % in the surface ocean (0 - 20 m). Therefore, we consider this to be a minimal effect and the overall results are not changed. (Note for the Editor, the numbers in the uncertainties table have been updated to corrected values that align with the cumulative Chl-a figure.)*

*As stated above, the under-ice threshold is a decision made by the BGC-Argo community and investigating this in detail is beyond the scope of this study. We acknowledge that there will be uncertainty in the start dates of the ice-free and ice-covered seasons and that estimates of under-ice Chl-a will be an upper estimate.*

L. 145: Please also mention in the text (not only in the Table) that you're assessing chlorophyll averaged over 0-20m here.
*Added to text*

L. 152: I am not sure I fully understand: How can the smaller spatial coverage of the NPP product than of the chl-a product be due to "satellite observation limitations" if the latter is also a remote-sensing product? Can you clarify this?
*Modified in text:*

> *Some of the input data (absorption due to gelbstoff and detritus, absorption due to phytoplankton and backscatter spectral slope parameter) used in the CAFE algorithm to derive NPP have less extensive spatial coverage than the Chl-a input data.*

L. 153: The number you give here is difficult to put into context. I suggest to also give the number relative to the whole Weddell Sea area in the text (not only in the Table).
*Investigation into the cause of data gaps/incomplete spatial coverage produced estimates of the discrepancies between Chl-a data coverage and the coverage of data in the NPP product. We have also calculated the percentage of ice-free area (IFA) that the CAFE NPP product has data for. The text now reads as:*

> *This means that there are some areas in the NPP product that imply there is no NPP occurring despite Chl-a being observed by the satellite. Over the timeseries, between 10-100% (mean 78%, median 85%) of the area with Chl-a data also has coverage in the CAFE NPP product (lowest values occur as the satellite view declines at the start of winter). The spatial coverage of the CAFE product in relation to IFA ranges between 0-94% over the seasonal cycle. When ice-free and with sufficient solar elevation for ocean-colour observations, on average 47% of the IFA was visible in the CAFE product. This difference between the IFA and area where NPP is observed in the CAFE product is most acute in the shelf region, where 0-65% (mean 10%) of IFA has NPP data when conditions are suitable for ocean-colour observations, and many coastal polynyas that are visible in the input Chl-a data are not translated into the NPP data.*

L. 155/156: In relation to my comment on L. 152, could you elaborate on what field causes the polynyas to not show up in the NPP products and why?

*The adjacency effect associated with sea ice and ocean-colour satellites as well as the absorption and backscatter variables used to calculate CAFE NPP (mentioned in the L152 comment response) cause polynyas not to show up in the CAFE product. The CbPM, VGPM and VGPM-Eppley models have also now been included in the supplementary figures to assess the robustness of the CAFE results on the shelf.*

L. 162: If your intention with Figure A2 was to highlight that satellite-based chlorophyll estimates often underestimate concentrations relative to float-based estimates, I suggest stating that more explicitly in the text. I do not find the current, rather generic statement of "showing similarities and dissimilarities" very helpful.

*We have moved the sentences referring to Figure A2 earlier in the uncertainties section and have added more detail about the similarities and differences between the satellite and float Chl-a observations. It now reads as:*

> *Figure 2 highlights the similarities and dissimilarities in float and satellite Chl-a estimates. The absolute values of Chl-a differ between the satellite and the float because of the way Chl-a is derived from observations on these platforms - satellite Chl-a is derived from reflectance, whereas fluorescence is used to calculate float-observed Chl-a. Nonetheless, the relative changes in Chl-a over the timeseries should be 185 seen in both datasets. In most cases, the satellite Chl-a at each float location peaks and troughs at the same time as the float Chl-a. On some occasions, the spring peak/bloom is not fully seen by the satellites, and in several cases (Figure 2: b) and c) in 2018, f) and k) in 2020), it is missed entirely. As such, the addition of float data in this study expands on the results derived from the satellite products and allows assessment of the seasonal patterns in Chl-a. Notably, it is clear that the satellite Chl-a product does not span the entire ice-free/growing season and the floats are valuable in providing a year-round perspective on 190 the seasonal changes in Chl-a and POC*

Additionally, please increase the font sizes in this Figure.

*The font sizes have been enlarged in all figures where it was necessary.*

L. 169: I would delete the "area-normalized" here for clarity. It took me a moment to realize that the contrast in this sentence was between daily and annual, not between integrated and area normalized.

*This has now been removed*

L. 170: In the text you say "mg C" but the y-axis label of Fig. 2 says "g C" – can you double check which one is right?

*The unit should be "g C", this has been changed in the text.*

L. 170: How large is the difference in the ice-free season? It might be of interest to the reader, so please consider adding this information.

*The following has been added to the results section:*

> *The sea-ice product shows that areas in the outer North-East edge of the open ocean are at the outer extent of the SIZ and can be ice-free for entire years, while on average, the whole open ocean region is ice-free for 139±13 days per year. The longest any of the shelf region is ice-free is 157 days, while the mean is 37±13 days.*

L. 170-173: This is when I first wondered what the impact of the missing data on the shelf is. Given that you're missing NPP in coastal polynyas (see your method section), this bias might be worth mentioning, even though it might not change that much on a basin-scale.

*As described in response to general comment 2, we found that a considerable proportion of the IFA does not have NPP values and that there is a trend in the missing data bias that would likely impact our estimates and trends seen in NPP, particularly on the shelf. As a result, we have calculated an estimate of NPP across the ice-free area and these are reported in paper, with the results and discussion text modified in line with the recalculated results.*

L. 172: In the abstract, it says 95%. Should it say 99%?

*Thank you for picking up on this, the value should have been 99%. Although, the main text and abstract now reflect the updated values (93-96%) following calculation of uncertainties.*

L. 174-178: For all numbers in this paragraph, I suggest adding the "% of total area" as it is rather difficult to put the numbers into context otherwise.

*We have now added these numbers to add context:*

> *The summer maximum IFA for the entire Weddell Gyre averaged ± standard deviation 4.73±0.23 x 106 km2 between 2003 and 2020 (80±4% of the total study region). The mean IFA per year for the entire Weddell Gyre averaged 2.18±0.24 x 106 km$^2$ between 2003 and 2020. On average in the shelf region, summer (maximum) IFA was 2.17±0.57 x 105 km$^2$ (25±7% of the shelf area), the mean IFA was 8.10±1.67 x 104 km$^2$ . In the open ocean region region, the average summer (maximum) IFA was 4.50±0.22 x 10$^6$ km$^2$ (89±4% of the open ocean area), mean IFA averaged 2.08±0.23 x 10$^6$ km$^2$*

*(The absolute values of ice-free area in the Weddell and Shelf regions are slightly different as the previously reported values included an area that was removed prior to submitting the manuscript. All other values have been checked and updated where relevant.)*

Fig. 2: I found it a little confusing to have a legend with "IFA" in panel a, even though this quantity is only shown in panel b. Maybe move the legend to outside of panel a, e.g., on top of or below the Figure? In panel b, is the maximum IFA shown for the whole Weddell Sea? Please clarify in the y-axis label and the Figure caption. Looking at the Figure and assuming this is

indeed whole-Weddell Sea-IFA, I was wondering why this is not split into open ocean and shelf as well (for consistency). Lastly, the way NPP is currently plotted, hardly anything can be seen for NPP on the shelf. I realize that this might have been your point, but for any reader interested in the evolution of NPP on the shelf, this would be more easily visible if plotted on two separate y-axes.

*The figure has been modified following the reviewer suggestions and to include the scaled-up NPP estimates:*

[Figure]

*Figure 3. CAFE NPP: a) Mean daily NPP. Orange bars represent the shelf region value and blue bars for open ocean values; b) AreaNormalised Annual NPP; c) Shelf Region Total Annual NPP shown on the left axis and Annual (Summer) Maximum Ice Free Area on the right axis; d) Open Ocean Total Annual NPP shown on the left axis and Annual (Summer) Maximum Ice Free Area on the right axis. The light bars represent the raw (directly observed) values integrated from the CAFE NPP product, the dark bars indicate the imputed values calculated to consider data gaps in the spatial coverage of the CAFE product.*

L. 190-194: What you write in this paragraph is very hard to see in the Figure. Also, unless I missed something, you do not currently show the temporal evolution of sea ice on the shelf anywhere, do you? Please see my comments on Fig. 2 above for suggestions.

*A panel for shelf region has been added (see above), and reference in text has been included*
Additionally, given your statements in the method section that you might miss NPP in coastal polynyas, have you checked in the chlorophyll and sea ice data whether there are any trends in "possibly missing NPP in polynyas" over time? Can you assume this bias to be constant in time or would it modulate any trends in NPP on the shelf reported here? I think this is an important

aspect to check and report to add confidence or uncertainty to the identified trends in NPP. Have to check if the bias is constant in time

*As described in the response to General comment 2:*

*Absorption and backscatter coverage is sparse during times of rapid sea ice retreat, and appears to decline in coverage/quality from 2010 onwards. There is inter-annual variability in the area that is affected on the shelves: The proportion of ice-free water that is visible in the CAFE model is 28% +/- 20% (mean +/- standard deviation) and there is a declining trend seen in the proportion of ice-free water that is visible in the CAFE NPP product (5 % decline per year, p = 0.001). This trend in the bias is not present in the other NPP models. As a result, it is uncertain whether the declining trend in NPP seen on the shelf is true. A weak(er) decreasing trend in annual NPP on the shelf is seen in the VGPM and VGPM-Eppley models (p = 0.036 for both), but not in the CBPM model (p = 0.058).*

*We have calculated a "scaled to ice-free area" estimate of annual NPP and reported that alongside the original values in the Fig.2 bar charts and Fig.4 and Appendix regressions. The timeseries trend in annual NPP on the shelf becomes less significant when using these values, as now described in the text:*

> *In contrast, a trend in NPP is seen in the shelf region. In the CAFE model, imputed NPP declined by 3% per year, p=0.02 (Fig. 3.b). A similar rate of decline, although less statistically significant, is seen in the other NPP models. The directly-observed CAFE estimates of NPP decreased more rapidly (average decrease of 7% per year, p=0.001), underscoring the large influence of missing NPP data in the shelf region. Westberry et al.,2023 describe other potential causes for trends seen in NPP products (e.g. physiological changes in phytoplankton and decoupling of Chl-a and NPP), and emphasise the difficulty in identifying trends in NPP data and inferring drivers of trends. The trends seen here are sensitive to the occurrence of extremes in the early part of the time-period when there was a collapse of the Larsen B ice shelf along the Antarctic Peninsula (Peck et al., 2010). No trend is seen in the shelf NPP when the first data point (year 2003) is removed.*

L. 196: I am not sure I understand what "time" you used in the regression here. Do you mean the total duration of open water area? Please clarify.

*We appreciate the request for clarity. We have added "(duration of ice-free season visible to the satellite)" to the sentence.*

L. 197-199: For this statement to hold, it assumes that the maximum sea ice area is far less variable over the years than the maximum ice free area, doesn't it? I would assume this is true (and assume that you have checked this), but this could be explicitly stated for clarity.

*This is the case and we have now included the following:*

> *The winter maximum sea-ice area within the study region box is much less variable than the summer maximum ice-free area (standard deviation: $6.6 \times 10^4$ and $2.3 \times 10^5$ km for the winter SIA and summer IFA respectively). The study*

*area is almost entirely ice-covered each winter, and any variability in the actual maximum sea-ice extent occurs north of the study region and is not significantly reflected in the study box.*

L. 203-207: Again, the reader is left wondering what role the potential NPP underestimation due to missing NPP in (some) coastal polynyas plays for this finding. It would be helpful if you elaborated on this here (or in the discussion). Do you really think that sea ice is a less strong predictor of NPP in shelf seas or do you think that the caveats associated with the data used here complicate the comparison between the role of sea ice on the continental shelf and in the open ocean?
*Following the other comments about the data biases and uncertainties, and the calculation of a total NPP estimate that is scaled-up to account for the data missing in some of the ice-free waters, we find that the relationship is now strongest in the shelf region (R2=0.55, p<0.001). We can conclude that the data biases had a confounding effect on the previously reported result.*

L. 208-213: I am not sure I find this paragraph particularly useful as it is. I found the per-pixel analysis in the subsequent paragraph much more insightful. I am wondering if the main message of this paragraph ("Over the whole open ocean, the area-normalized annual mean NPP is not correlated with mean IFA, sea-ice retreat or mean visible days, but it is on the shelf") would not be better embedded into this subsequent paragraph to improve the flow. Unless I misunderstood something, it is the breakdown of the relationship in the open ocean beyond ~120 days that also causes/contributes to the absence of correlation on a basin scale. Do you agree?
*Upon further reflection from this comment, we think the inclusion of this result complicates the message without adding additional insight that isn't already present in the subsequent paragraph. We have therefore removed this paragraph.*

L. 211: How does this statement fit to the finding in the previous paragraph, i.e., the low correlation between the maximum ice-free area and NPP?
*As above, the correlation between maximum IFA and NPP on the shelf region is greater when NPP is scaled-up to account for data gaps. Conversely (and interestingly), the R2 for the open ocean has decreased with this scaled-up NPP estimate. We think this lends more support to the conclusion that while the maximum IFA will influence total NPP to an extent because it dictates how much of the open ocean/a region is alleviated of light limitation, there are other factors that will determine the upper limit for NPP in the open ocean.*

L. 214: a wide range was recorded
*Grammar corrected*

L. 215: Whatever you decide to do about my suggestion of the paragraph in L. 208-213, I suggest moving the definition of "visible days" to where these are first mentioned in the result section.
*This remained where it was.*

Fig. 5: The y-axis label might be easier to read if given in %.
The colourbar axis was changed to percentages. The figure now appears as:

[Figure]

Fig. 6: In the caption, please describe panel a before panel b and use the panel labels in the caption (instead of "upper panel", same goes for in the main text). Further, for panel a, please add what the whiskers, the orange line etc. represent (Median? Mean? Which percentiles?) in a legend and in the caption. I further suggest finding clearer y-axis labels for the current panel a. For example, for "days to 50% 0-200m", I suggest specifying that you're referring to chlorophyll and that this is the bloom end. This should also become clear from the figure itself, not only from the caption and the text).

*In text referenced to the figure now refer to the panels by letter. Changes have been made to the figure and caption:*

[Figure]

*Figure 7. a) Percent cumulative Chlorophyll-a in the surface (0-20m) and between 0-200m for all completed annual cycles by the floats between 2015-2021. b) Box and whisker plots indicating the range in the number of days from ice melt to (i) sea-ice return, (ii) low solar angle (loss of satellite observations), (iii) surface bloom end and (iv) depth-integrated bloom end. Orange line represents the median value for each, the box represents the first and third quartiles, while the whiskers extend to 1.5 times the inter-quartile range.*

L. 231: I suggest rephrasing "after this much exposure". Maybe "after waters have been ice-free for more than 4 months"? For me, "exposure" is too unclear.
*We have adopted this suggestion into the text.*

L. 231: Can you specify in what way the use of floats deepens the analysis? It would be helpful for the reader if you stated explicitly what can be gained from additionally analyzing the float data. The current statement is rather vague.
*First paragraph in the "Aligning satellite and subsurface perspectives" section has been changed to:*

*Satellite observations indicate that, in the open ocean, the strong positive correlation between visible days and NPP degrades after around 130 visible days, indicating that other processes (e.g. grazing, nutrient availability) potentially begin to limit NPP after waters have been ice-free for more than 4 months. However, as described in Section 2.4, the ocean-color satellite loses coverage in late summer, when the solar angle decreases below 20°. As a result, it is uncertain whether further NPP is occurring, and therefore, missed after this point. Assessment of float Chl-a and POC (as proxies for phytoplankton growth/biomass) can reduce this uncertainty by indicating whether phytoplankton are still present in the surface ocean and/or whether growth may still be occurring beyond the date when satellites lose visual coverage. Therefore we seek to address if significant growth is missed after loss of satellite coverage in late summer and whether the same relationship between ice-free days and phytoplankton growth is seen in the available float observations. Although these data come from drifting platforms, rather than fixed points, we can enquire how the seasonal cycles of Chl-a and POC unfold in each year, and specifically how they evolve relative to light availability. It is worth noting that these data all represent open ocean conditions as floats are not deployed in regions shallower than 2000m.*

L. 234: I suggest rephrasing "depth restrictions" here. This made me think of the limitation that floats only sample the top 2000m – which is not the shortcoming in the context of your analysis. I assume you're referring to the fact that because they sample the top 2000m, they are not deployed in regions shallower than 2000m, i.e., the shelf regions. Can you clarify this in the text?
*Thank you for the suggestion, we have amended the text to:*

> *It is worth noting that these data all represent open ocean conditions as floats are not deployed in regions shallower than 2000m.*

L. 235: If you're referring to the bottom panel first, I suggest switching the order of the panels in the figure. It is always easier for the reader if you refer to panel a first.
*We have implemented this suggestion - see response to Fig.6 comment (above)*

L. 236: It might be helpful here to add "blue lines in Fig. 6a" [or panel b, depending on whether you decide to switch the order or not] and "green lines in Fig. 6a" to the text to guide the reader.
*Added:*

> *We evaluate the bloom progression for both the surface 325 (0 to 20m; analogous to satellite-observed depths; green lines in Fig.7.b) and upper 200m (blue lines in Fig.7.b),*

L. 237: Do you have a reference for this definition of the bloom end? This is different from the more typical bloom definition in the literature (see e.g., Siegel et al., 2002 or Soppa et al., 2016,

Hague & Vicchi 2018), and I am wondering why you did not use a criterion that makes use of the complete annual information. Please clarify.

*Hague & Vichi, 2021 used the time derivative/rate of change in Chl-a within their definition of growth initiation. Our definition was informed by this prior use of time derivatives to define bloom dynamics.We noted the plateauing nature of the cumulative increase in Chl-a, and aimed to define the timing of that plateau. Notably, the values of Chl-a observed by the floats varied considerably, and it was not possible to use an absolute value (such as 1 mg m$^{-3}$) as a threshold from blooms and bloom termination. The "bloom-end" term used here is a subjective definition in order to quantify the slowing/decline in growth/increase in Chl-a.*

L. 242-247: Related to my comment above, I am left wondering how sensitive this assessment is to how you define the bloom end. Have you tested different bloom metrics?

*As mentioned in the response above, the definition of bloom end was created to identify the point at which growth/the rate of increase in Chl-a slows and levels off. Other bloom metrics, such as thresholds of absolute values, were not applicable for this purpose due to the range in absolute Chl-a values observed by floats. For example, using 1 mg m$^{-3}$, would miss some blooms completely, and not accurately identify the decline in the rate of change in Chl-a concentrations.*

L. 240: Please be careful here not to equate chlorophyll concentrations, i.e., a proxy for carbon biomass concentrations, with primary production. Chl:C ratios might vary substantially across the growing season, both in response to a changing light and nutrient environment and changing community composition. Further, as a proxy for biomass, chlorophyll concentrations also integrate loss terms.

*Following comments from reviewer 2, we have now included analysis of backscatter derived POC estimates. We assessed changes in Chl-a in relation to concurrent changes in POC.*

● *The sentence in question has been modified and followed up with a sentence commenting on the POC results:*

  *The later date of bloom end for the depth-integrated values indicates that primary production could be continuing in the water column at depth until later in the season than at the 330 surface. Certain float years in Figure A2 (panels a-d and k) see a concurrent increase in Chl-a and POC in the depth integrals that indicates that active production/growth is taking place, as opposed to changes in phytoplankton physiology or community composition driving changes in Chl-a (Thomalla et al., 2017).*

● *Interpretation of these results have been added to the discussion to support the interpretation that increases in Chl-a indicate occurrent of primary production:*

  *Our hypothesis of iron limitation at the end of the growing season is supported by the sub-surface Chl-a and POC observed by floats (Appendix Figures A1 and A2). Changes in Chl-a concentrations can arise from several situations aside*

*from growth/accumulation of biomass: photo-acclimation, nutrient limitation and changes in phytoplankton community composition (Thomalla et al., 2017). Comparing Chl-a to POC, we can assess what may be causing changes in Chl-a. The presence of elevated Chl-a concentrations close to or below the base of the mixed layer, often (but not always) after the cessation of the initial surface bloom (Appendix Figure A1), suggests that phytoplankton are benefiting from replenishment of nutrients from below the mixed layer through diapycnal mixing (Arrigo et al., 2015; Taylor et al., 2013). Elevated POC signals coincide with increased Chl-a in the majority of these cases, providing evidence that active production is taking place at depth (Appendix Fig. A2). Surface nutrient concentrations are thus likely to be limiting phytoplankton growth in many areas of the ice-free Weddell gyre, although float data do not allow us to quantify its net impact on NPP. Grazing pressures may also be important in driving the differences in surface and sub-surface phytoplankton dynamics (Baldry et al., 2020, also see Section 4.2.3).*

*The complexity of the relationship between light and nutrient limitations – and their implications for inter-annual variability in annual NPP – is highlighted by the occasional occurrence of a secondary (temporally separated) late-summer bloom (Appendix Figure A1. e.g. panels a) 5904397: 2018, 2019; b) 5904467: 2018; c) 5904468: 2018, 2019; d) 5904471: 2018; g) 5905992 2020). As seen in the matching Chl-a and POC signals at depth, the second peaks in surface and depth integrated Chl-a that suggests a late-summer bloom are matched by simultaneous POC increases at these times, implying active growth within the phytoplankton community (Appendix Fig. A2). There are four float years (Appendix Fig. A2 h) 5905994: 2020; j) 5906033: 2020; and k) 5906034: 2020, 2021) that saw small increases in Chl-a at the end of the ice-free season without a concurrent increase in POC. We conclude that the increase in Chl-a in these cases may be a result of phytoplankton photoacclimating to the decreasing light conditions.*

L. 254: What do you consider "long"? As this is subjective to the reader, I suggest being more specific here.
*Amended to say:*

*climate-relevant (centennial to millennial)*

L. 255-262: A lot of this reads more like information for the introduction, not the discussion. Please revise to make this a more concise discussion of your findings. In general, I would always argue for the "one idea/message per paragraph" structure. In section 4.1., I have trouble identifying one message per paragraph – to me, it is the same message in each (see also section title). Maybe combine into a single paragraph?
*We appreciate the request to make the first subsection of the discussion more concise. It now reads as:*

*This study shows that over the last 18 years, open ocean productivity has a dominant role in the biological carbon cycle of the Weddell Gyre, in agreement with synoptic ship-based studies (MacGilchrist et al., 2019; Brown et al., 2015). MacGilchrist et al. (2019) documented that the region was a net annual CO2 sink only due to a large amount of carbon sourced from biological activity in the central Weddell Gyre accumulating in the local Circumpolar Deep Water; horizontal circulation of this water mass out of the Weddell Gyre then facilitates sequestration of carbon over climate-relevant (centennial to millennial) time scales. Our current study highlights the importance of the Weddell Gyre's open ocean compared to the shelf region, showing that the majority (93-96%) of the carbon uptake by phytoplankton in the Weddell Gyre occurs there, despite higher daily rates of NPP observed on the shelf. This is due to its far greater areal extent (an order of magnitude larger than the shelf region) and much longer ice-free growing season. In light of our results here, we highlight the need for future research in this region to further quantify the contribution of open ocean production to the biological carbon pump.*

L. 263-266: As stated above, I think it is very important to at least mention shortcomings related to data coverage in the NPP data set here.

*We have added the following to the end of the first paragraph in the discussion:*

*Satellite NPP values are often underestimated due to the negative biases in spatial coverage within NPP products. Gap-filled, permuted NPP estimates were calculated to reduce these uncertainties and biases, particularly in the shelf region where spatial coverage was impacted by significant gaps in the data. These values are considered to be more realistic estimates of the total NPP occurring in the two regions. It is important to note that the gap-filling correction assumes that mean NPP is equal in the visible and non visible portions of the region and so could therefore still carry uncertainties.*

L. 269: Why "likely"?

*"likely" removed*

L. 280: Are you referring to your own work here or somebody else's with the comment in the parentheses? This is not clear to me, as I do not find this information anywhere in your result section.

*In the Uncertainties section, we presented the proportion of annual Chl-a observed by floats whilst under sea-ice in the text and in the table. We realise that we did not explicitly refer to these values within the results nor discussion section where this comment refers.*

*We have added this result to the "Climatological NPP and Sea-Ice" result section:*

*As described in Section. 2.4 and summarised in Table. 1, BGC Argo float estimates of Chl-a and POC both under ice and at low sun angle suggest that substantial amount of phytoplankton biomass may be missed by satellite NPP*

*products in the open ocean (medians of 20% Chl and 39% POC for all float locations).*

*And we have adapted the discussion to read:*

> *The float data showing that 2-23% of integrated Chl-a (and 7-30% of surface POC; Table. 1) is present potentially before sea-ice retreat suggests that our the satellite analysis may over-estimate somewhat the correlation between IFA and NPP. Recent studies (Bisson and Cael, 2021; Hague and Vichi, 2021; McClish and Bushinsky, 2023) have also reported the presence of considerable amounts of Chl-a under sea ice as well as highlighting the onset of growth prior to complete sea-ice retreat (Hague and Vichi, 2021; McClish 375 and Bushinsky, 2023). However, while our float observations also indicate that biomass tends to increase before complete ice retreat, our results still clearly show IFA as a major productivity driver. Strong phytoplankton growth follows ice melt (Fig. 7) and the majority of phytoplankton biomass is found in ice-free conditions (Figs. 2, A1 and Table 1). Similarly, in McClish and Bushinsky (2023), the break-up of sea ice initiates the increase in Chl-a and POC, highlighting the light limiting control of sea ice on phytoplankton growth.*

L. 281: Weddell Sea instead of Weddell Gyre
*We believe that Gyre is the correct term to use to refer to our study region (and in this particular sentence "our basin-scale analysis of the Weddell Gyre"), as the outer boundaries have been defined and used in previous studies to delineate the extent of the Weddell Gyre specifically (Akhoudas et al., 2021, Brown et al., 2014, Brown et al., 2015, Jullion et al., 2014, MacGilchrist et al., 2019).*

L. 282: the instead of a dominant driver
*changed*

L. 285: I find the formulation "provide more space" odd in this context. Can you rephrase?
*We removed this sentence when we restructured this part of the discussion*

L. 297: Please add "in review" or "submitted" after this reference.
*This paper has now been accepted, so citation and reference have been updated.*

L. 298: I suggest adding "for satellite detection" after "sufficient available light" to increase clarity here as this is what you show in Figure 5.
*"for satellite detection" added*

L. 305: Some of the references mentioned further down (L. 318/319) should be mentioned here already to point out that other authors have reported this transitioning of limiting factors in the high-latitude Southern Ocean.

*In response to the next comment, this paragraph towards the end of the "Light" section was combined with the start of the "Iron" section to reduce repetition*

L. 315-320: This feels repetitive with the previous section. Consider deleting/shortening to reduce redundancies.
*The last two paragraphs at the end of the "Light" section were integrated into the start of the "Iron" section to reduce repetition. The "Light" section ends at the description of the shelf region being primarily light limited as there is a plentiful supply of nutrients to the region.*

L. 330-335: While it is very likely that iron is indeed the limiting factor for growth, differences in grazing pressure might also play a role in explaining differences between surface and subsurface bloom dynamics of phytoplankton chlorophyll. I suggest slightly adapting the language here to represent more accurately what you can be sure about and what only appears likely.
*We have taken this into account and amended the language such that the first sentence referred to here now says:*

> *We postulate therefore that iron limitation could be a major driver of the slowing/decline in NPP that occurs prior to sea-ice return in the open ocean*

*And we have added a mention of grazing pressures in this section:*

> *Grazing pressures may also be important in driving the differences in surface and sub-surface phytoplankton dynamics (Baldry et al., 2020, also see Section 4.2.3).*

L. 336: Do you mean phytoplankton community composition? If so, could you elaborate on how you conclude this from the chlorophyll data?
*In their review of the occurrence of SO sub-surface Chl-a maxima, Baldry et al., 2020 describe the dominance of diatoms in sub-surface Chl-a maxima and a shift/difference in phytoplankton community between the surface and sub-surface. We have carried out assessment of float POC While Chl-a and POC (and Chl:POC) do vary with depth, we acknowledge that we cannot say for sure whether these changes are due to community composition differences, or physiological changes due to photo-acclimation, or nutrient limitation. We have removed the statement and added a reference to Baldry et al., 2020 in relation to surface and subsurface differences.*

L. 355: What warmer areas of the Gyre are you referring to here? I suggest using geographic descriptors in this context.
*Amended to:*
> *warmer areas in the Eastern Weddell Sea (~ 20 - 25°E)*

L. 364: Grazer populations exert top-down control on phytoplankton communities whenever they are present, not only by late summer. I suggest rephrasing to "may dominantly control phytoplankton biomass/communities" or such.
*This suggestion was applied to the sentence, such that it reads:*

> *The presence of sea ice can act to decouple grazers and phytoplankton (Hoppema et al., 2000; Rohr et al., 2017; Smetacek et al., 2004), and as micro-nutrient availability diminishes, grazing may increasingly contribute to the decline of the NPP bloom such that, by late summer, grazer populations may be the dominant control of phytoplankton biomass/communities (Rohr et al., 2017; Smetacek et al., 2004).*

L. 378: As stated by the authors in the abstract, this statement only holds as long as the other environmental variables do not change (nutrient availability, grazing). I suggest adding this information/assumption here.
*We have added:*

> *assuming there are no changes to other environmental variables such as nutrient supply and grazing*

L. 380-383: Please check for redundancy within the first half of the paragraph.
*These lines have been removed*

L. 384: As you only infer this from your results and don't actually show it, I suggest saying "imply" instead of "indicate". I further suggest deleting "particularly", as you only infer this for the open ocean and not at all for the shelf. These changes would reflect your findings more precisely in my opinion.
*Thank you for these suggestions, these changes have been made.*

L. 385: Again: unless nutrient supply changes.
*The sentences in this paragraph have been rearranged for clarity. It now reads as:*

> *Our results further imply that nutrient supply is a key control on this upper limit for NPP in the present day in the open ocean. Consequently, how NPP will change across the Weddell Gyre becomes sensitive to how iron supply will change in the future. Such changes could also be mediated by changes in sea-ice dynamics due to their impact on stratification, mixing and upwelling (Moreau et al., 2023; Hoppema et al., 2015). In future warming conditions, increased stratification, combined with freshening from melting ice could 490 act to cut off biological productivity by reducing the vertical nutrient supply (Bronselaer et al., 2020). This will be particularly apparent in the open ocean, given its greater distance from terrestrial micro-nutrient sources. Noh et al. (2023) recently showed that, within CMIP models, Chl-a in the Arctic declines as a result of reduced nutrient supply when regions become ice-free. Despite being based in the Arctic, and thus differing physically and ecologically from the SO, this result in Noh et al. (2023) could point to a less productive Weddell Gyre in the future, should any of it become permanently ice-free. Notwithstanding changes 495 in nutrient supply, an increasingly ice-free Weddell Gyre will see a greater expanse experiencing nutrient limitation late in the growing season. It is unclear whether the same*

*limitations will have a similar effect on NPP in the shelf region should it become increasingly ice-free for longer than is currently seen (~ 130 days).*

L. 398-405: I find a lot of these statements rather generic, and as such, they do not represent strong concluding sentences based on the results and discussion presented in this paper. I suggest re-working the conclusion section.

*This sentiment was shared by both reviewers. As such we have amended the conclusions section to better highlight the findings, implications of this study and suggest avenues for future research. It now reads:*

*This study used a complement of satellite-derived sea ice and NPP products as well as BGC-Argo float observations of Chl-a and POC as proxies for phytoplankton biomass to assess the basin-scale relationship between sea ice and phytoplankton growth. We find that sea ice is the primary control on Weddell Gyre NPP in areas that experience fewer than 70-130 ice-free days per year. Beyond ~ 130 ice-free days, float Chl-a and POC observations suggest that nutrients (likely iron) emerge as an important limit to growth, possibly co-limiting with top-down grazing control. We find that while the shelf region sustains higher instantaneous NPP during its ice-free window, the open ocean sustains 93-96\% of the annual NPP of the Weddell Gyre, due to its larger area and longer ice-free season. Furthermore, while sea ice is a primary driver of inter-annual variability in total annual NPP in the Weddell Gyre, nearly half of NPP variability is still unexplained, motivating further study. We found no long-term trends in the Weddell Gyre sea-ice extent or NPP during the study period. However, our results suggest that NPP will increase if sea-ice extent decreases in the future, at least until the Weddell Gyre is ice-free for longer than 130 days, at which point, controls other than sea ice may dominate. Finally, this work has highlighted the importance of using BGC-Argo float data to complement and corroborate satellite data analysis. The study highlights the need for development of quantitative float-based NPP measurements in the region, which would likely benefit from inclusion of PAR sensors on more floats.*

*We have taken into consideration the additional comments below when re-writing the conclusions.*

L. 391: The float data do not give estimates of NPP, please be precise. Additionally, since you do not only look at sea ice but also visible days, I suggest rephrasing to something that better synthesizes what you have done.

L. 392: "It is clear" – This makes it sound like it was clear already before your study. I do agree with this interpretation (there was a body of work demonstrating the link before this paper), but I am not sure this is what you actually refer to here.

L. 394: Please add the number here instead of saying "to a high degree".

L. 397: I disagree with the authors that the float data demonstrate the iron limitation – it might seem plausible (and is probably true), but this has not been explicitly shown in this paper. Please elaborate on this or rephrase.

Figure A3: All font sizes are way too small.
*The font sizes have been enlarged in all figures where it was necessary.*

---

## Author Comment (AC2)

**Author Comment 2 - Responses to Reviewer 2**
**"Exploring the relationship between sea-ice and primary production in the Weddell Gyre using satellite and Argo-float data"**

*We are grateful to the two reviewers for their positive responses and constructive suggestions and comments. In the following,* reviewers' comments are in regular typeface *and our responses are in blue italics.*

Douglas et al primarily investigates interannual fluctuations in satellite-derived NPP as they relate to sea ice variability in the Weddell Sea. The main result is that annual NPP and annual maximum ice-free area are correlated at interannual timescales. They also contrast the shelf and open ocean regions. For example, they show that in the open ocean, an increase in satellite visible days corresponds to an increase in annual NPP up to a certain point only. This presumably reflects a shift from light to nutrient limitation over the course of the growing season. I think this paper will be a useful contribution to the community, but some points need clarification before the paper is suitable for publication. In light of this, I'm suggesting major revisions. My general and detailed comments are provided below.

**General comments:**
1. First, I find the usage of gyre confusing in the context of this manuscript. First off, the boundaries of the study region are hydrographic transects that have nothing to do with the actual gyre dynamics. Second, the gyre is typically not thought to extend all the way onto the continental shelf, e.g. see map of mean dynamic ocean topography in Fig. 5a of Armitage et al. (2018). So the division into open ocean and shelf regions seems to apply to the Weddell Sea rather than the Weddell Gyre. I would consider replacing "Weddell Gyre" with "Weddell Sea" in the title and throughout most of the manuscript.
*We have changed Weddell Gyre to Weddell Sea in the introduction in the contexts of deep-water formation and when citing work by Arrigo et al., 2008 that referred to the Weddell basin/sector as the Weddell Sea. However in the context of our study region we have continued to refer to the Weddell Gyre. The hydrographic transects, used during ANDREX cruises, have been shown to broadly align with the boundaries of the gyre (Akhoudas et al., 2021, Brown et al., 2014). In Armitage et al., 2018, they show that the Weddell Sea refers to only the western area of our study region and that the gyre extends to ~30 deg E, where our eastern boundary is. For consistency with the terminology used in previous papers that use the hydrographic transects as their study area boundaries (Brown et al., 2015, Jullion et al., 2014, MacGilchrist et al., 2019), we use Weddell Gyre.*

2. Second, since satellites cannot see through sea ice, it seems inevitable that the annual NPP over the entire region (as derived from satellites) will necessarily be higher when there's greater icefree area simply because you have the ability to detect the NPP? For example, the ice-free area is correlated with the total annual NPP (Fig. 2b) but not with the area-normalized annual NPP (Fig. 2a). So isn't this suggesting that the greater annual NPP is simply due to there being more ice-free pixels (with non-zero NPP)?

*A larger IFA may not always result in greater annual NPP, for instance in an area that is not light-limited and instead primarily controlled by other factors. Instead, the correlation indicates that a larger area over which light limitation from sea-ice cover is alleviated results in more NPP. As the reviewer stated below, the majority of phytoplankton biomass (Chl-a and POC) is seen by floats after the waters are ice-free, supporting the assumptions in place because of the limitations of the ocean-colour satellite that assume no under-ice NPP. Secondly, although the relationship between summer IFA and annual NPP is significant, it does not account for all of the variance in NPP. We have emphasised this and discussed drivers of unexplained variance in the discussion.*

A critic might argue that if there were significant under-ice NPP that is undetectable by satellite, the correlation between total annual NPP and ice-free area is an artifact related to the limitations of the satellite data. I actually don't think this is the case, since the floats show that a very small percentage of annual NPP occurs under the ice. But I think this point should still be addressed explicitly, and furthermore, this could help better integrate the float data analysis into the rest of the paper (i.e. if you frame the float analysis as a response to this imagined critic).
In general, in the current version of the manuscript, the float data feels unnecessary to the main results of the paper.
*The sentiment here, concerning better integration of the float data in the manuscript, is shared by both reviewers. A detailed description of the changes we have made can be found in response to Main Comment #1 by the first reviewer.*

*To summarise: The floats are an important component of this work for two reasons: 1) They allow us to assess the uncertainty in the satellite data (namely in quantifying what the satellite misses due to sea-ice cover and low solar angle). 2) They allow us to observe the seasonal progression in Chl-a and calculate a quantitative timeline of activity from ice melt through the growing season, providing a complementary perspective to that gained from the satellite data.*

*As an overview of the changes we have made:*
1) *The abstract has also been modified to integrate the float results more explicitly.*
2) *A sentence explaining the inclusion of float data has been added to the beginning of the autonomous floats methods section.*
3) *The uncertainties section has been modified to improve flow and clarity.*
4) *A paragraph has been added to the satellite results section to emphasise uncertainties in the data and lead into the floats results, emphasising the importance of the addition of float data to this paper.*
5) *The importance of the float data in supporting and expanding on the satellite-based results are also emphasised in the discussion*

*More details of how we have justified the inclusion of float data is provided in the response to reviewer 1.*

3. Regarding the float data, I also think you need to more explicitly mention the differences between chlorophyll and NPP since it feels like they're used interchangeably at many places in

the manuscript. I'm also wondering why you didn't use the POC estimates derived from the float backscatter? Backscatter-based POC is a somewhat better indicator of biomass than chlorophyll and perhaps more comparable to NPP than chlorophyll.

*We chose to look at Chl-a from the floats because we could compare it to the satellite Chl-a observations. We have now made it clearer in the text that Chl-a is used here as a proxy for growth, and that it does not equate to estimates of NPP. In our description of the CAFE model, we state that Chl-a is used in the model to calculate NPP. Additionally, at the start of the "Autonomous Floats" methods section, we state that we are "using Chl-a as a proxy for growth". We have also now added analysis of particulate organic carbon (discussion of results lines 427-447) (estimated from backscatter float data) as an estimate/proxy for biomass. We have calculated Chl-a:POC ratios to improve our interpretation of the seasonal progression in phytoplankton activity.*

**Detailed comments:**

Title: This title doesn't convey any of the actual results of the paper, so consider changing.

*Due to the changes recommended in the body of the manuscript that we have undertaken, we feel that the title is now more fully representative of the contents of the paper.*

*We have changed "primary production" to "phytoplankton growth" to better encompass the use of Chl-a and POC*

Lines 10-12: "…additional factors such as nutrient availability or top-down controls limit NPP."

*"additional factors such as nutrient availability or top-down controls (e.g. grazing) could be limiting NPP" added*

Lines 30-39: These sentences feel repetitive and are a bit hard to follow in terms of the actual writing. Consider condensing/rephrasing for clarity.

*We appreciate the request for further clarity. We have rearranged this section, which now reads:*

> *Climate models from Coupled Model Intercomparison Project Phase 5 and 6 (CMIP5 and CMIP6) project a decline in Antarctic sea-ice area and concentration as a response to anthropogenic climate change (Casagrande et al., 2023). However, low confidence in projections, due to the complexity of ocean-ice-atmosphere systems, means that exact estimates of decline are uncertain (Casagrande et al., 2023; Meredith et al., 2019). Therefore, in light of these observed and anticipated changes in the climate of the SIZ (Kumar et al., 2021; Ludescher et al., 2019; Casagrande et al., 2023), the need for a deeper understanding of the relationship between sea ice and NPP is pressing, as changes in sea ice will have concomitant impacts on carbon uptake and ecosystem health. However, the crucial gaps in our understanding of the drivers of NPP in the SIZ mean that large uncertainty remains about the nature and extent of these changes (Campbell et al., 2019; Henley et al., 2020; Kim and Kim, 2021; Pinkerton et al., 2021; Séférian et al., 2020; Henson et al., 2022).*

Line 40: I would say "The Weddell Sea…" rather than "The Weddell Gyre…" I think this applies to most of the manuscript (except for some other places in the Introduction that are explicitly related to the actual gyre), but I will not continue to point it out.
*We have changed this instance of Weddell Gyre to Weddell Sea to better reflect the area where deep/bottom water formation occurs. We follow our response to the general comment above with regards to the use of Weddell Gyre through the remainder of the manuscript.*

Figure 1: The Southern Boundary is hard to see on this map and I don't think it's referenced anywhere in the manuscript. Add contours showing the average annual maximum and minimum sea ice extent? There are no maps showing sea ice concentration so it's hard for the reader to visualize.
*Lines to indicate the maximum and minimum sea ice extent have been added, the shelf region line was changed to orange and the open ocean to dark blue and the Southern Boundary line was removed. The figure now appears as:*

[Figure]

*Fig 1: Location of the Weddell Sea and study subregions (dark blue: open ocean; orange: shelf). The 18-year mean area-normalised annual NPP (g C $m^2$) climatology derived from MODIS-Aqua satellite measurements using the Carbon, Absorption and Fluorescence Euphotic-resolving (CAFE) model (Silsbe et al., 2016) is represented by the yellow to green colourmap. White areas represent no data (permanent sea-ice present). SOCCOM float trajectories from 12 BGC-Argo floats are shown in greyscale and labelled with WMO ID. Profiles from float SD5905991 located north of the study region were not included in the analysis. The dashed and solid light blue lines denote the summer minimum (March 2006) and winter maximum (September 2010) sea ice extent respectively.*

Line 64: Missing the closing parenthesis that starts at Line 62.
*) added*

Lines 116-119: You haven't properly introduced the relationship of chlorophyll and NPP so this subsection feels out of place as written. Also, as I stated in my general comments, why not look at the POC derived from backscatter?
*We initially chose to use float Chl-a because, while Chl-a does not directly equal NPP, satellite Chl-a is used in all four of the open-source satellite NPP products, and as such we can compare float and satellite Chl-a products.*

- *We have added a more detailed description of the CAFE model in the "NPP and Chlorophyll-a" section of the methods, stating that NPP is derived from Chl-a (among other variables) and that:*

    *Cloud-filled MODIS-Aqua Chl-a concentration data were also obtained at the same resolution as the NPP data to later compare to BGC-Argo float data as proxies for growth.*

- *In the Autonomous Floats methods section, we now start the section with:*

    *We use BGC-Argo float data to evaluate the data recovery attributes of satellite data, estimate associated uncertainties (Section 2.4), and also to assess the seasonal progression of phytoplankton growth in the water column, using Chl-a as a proxy for photosynthetic potential and particulate organic carbon (POC) as a proxy for biomass.*

- *Analysis of POC data (derived from float backscatter) has now been added to this study. The methods are described in the Autonomous Floats methods section:*

    *POC concentrations were estimated from optical backscattering data after the removal of spikes due to large particles following Briggs et al., 2011. "De-spiked" backscattering were averaged in 10 m bins in the upper 50 m and then at 50 m intervals to 200 m. As with the Chl-a data, missing surface/shallow backscatter values were extrapolated (nearest neighbor) from the shallowest data available for each profile. Backscatter was converted to POC concentrations using the conversion co-efficient $3.12 \times 10^4$ as proposed in Johnson et al., 2017. The mean and depth-integrated POC in the 0-20m and 0-200m bins are reported here.*

- *Interpretation of these results have been added to the discussion to support the interpretation that increases in Chl-a indicate occurrent of primary production:*

    *Our hypothesis of iron limitation at the end of the growing season is supported by the sub-surface Chl-a and POC observed by floats (Appendix Figures A1 and A2). Changes in Chl-a concentrations can arise from several situations aside*

*from growth/accumulation of biomass: photo-acclimation, nutrient limitation and changes in phytoplankton community composition (Thomalla et al., 2017). Comparing Chl-a to POC, we can assess what may be causing changes in Chl-a. The presence of elevated Chl-a concentrations close to or below the base of the mixed layer, often (but not always) after the cessation of the initial surface bloom (Appendix Figure A1), suggests that phytoplankton are benefiting from replenishment of nutrients from below the mixed layer through diapycnal mixing (Arrigo et al., 2015; Taylor et al., 2013). Elevated POC signals coincide with increased Chl-a in the majority of these cases, providing evidence that active production is taking place at depth (Appendix Fig. A2). Surface nutrient concentrations are thus likely to be limiting phytoplankton growth in many areas of the ice-free Weddell gyre, although float data do not allow us to quantify its net impact on NPP. Grazing pressures may also be important in driving the differences in surface and sub-surface phytoplankton dynamics (Baldry et al., 2020, also see Section 4.2.3).*

*The complexity of the relationship between light and nutrient limitations – and their implications for inter-annual variability in annual NPP – is highlighted by the occasional occurrence of a secondary (temporally separated) late-summer bloom (Appendix Figure A1. e.g. panels a) 5904397: 2018, 2019; b) 5904467: 2018; c) 5904468: 2018, 2019; d) 5904471: 2018; g) 5905992 2020). As seen in the matching Chl-a and POC signals at depth, the second peaks in surface and depth integrated Chl-a that suggests a late-summer bloom are matched by simultaneous POC increases at these times, implying active growth within the phytoplankton community (Appendix Fig. A2). There are four float years (Appendix Fig. A2 h) 5905994: 2020; j) 5906033: 2020; and k) 5906034: 2020, 2021) that saw small increases in Chl-a at the end of the ice-free season without a concurrent increase in POC. We conclude that the increase in Chl-a in these cases may be a result of phytoplankton photoacclimating to the decreasing light conditions.*

Lines 125-127: It's worth mentioning that float timeseries reflect both temporal and spatial variability. The language here implies that the floats can be treated as the timeseries of a bloom at a particular location, but this may or may not be the case given the small decorrelation length scales for chlorophyll.

*We have added this caveat to the Uncertainties section:*

> *Floats data have their own limitations - floats are Lagrangian autonomous observing platforms, so observations reflect both temporal and spatial variability. Additionally, sensor calibrations may vary and sensors sometimes drift towards the end of the float deployment. We did not attempt to estimate water-column integrated NPP from float data, as the floats in the study region lacked PAR (Photosynthetically Active Radiation) sensors, and, as far as we are aware, there are not yet methods for calculating NPP from float data that have been robustly validated for widespread use.*

Lines 150-152: I don't understand this statement, which input data have less extensive spatial coverage than Chl-a?

*Reviewer 1 also commented on the input data biases. A more detailed response can be found in the response to reviewer 1. In relation to this specific section, we have modified text to say:*

> *In addition to this, there is also a disparity in the spatial coverage of the NPP products and the Chl-a input data used to derive CAFE NPP (Table 1). Some of the input data (absorption due to gelbstoff and detritus, absorption due to phytoplankton and backscatter spectral slope parameter) used in the CAFE algorithm to derive NPP have less extensive spatial coverage than the Chl-a input data. This means that there are some areas in the NPP product that imply there is no NPP occurring despite Chl-a being observed by the satellite.*

Lines 154-156: This seems like a limitation to the partitioning of total NPP on the shelf vs open ocean, which is framed as one of the main results of the paper. Obviously there's not much that can be done to address this, but it feels like it should at least be discussed later on in the paper.

*Despite this limitation, when NPP data gaps are imputed using regional timepoint tendencies, we find that our result indicating the dominance of the open ocean to Weddell Gyre NPP still stands (93-96% contribution). The imputed values are reported in the text.*

Lines 170-173: You should mention explicitly that the area of the open ocean is significantly larger than the shelf, which seems to be dominating the partitioning of the total annual NPP between the two regions.

*This is mentioned in the discussion, but we have added the following sentence here as well:*

> *The open ocean also has a far greater area than the shelf region (50.32 x $10^5$ km$^2$ compared to 8.81 x $10^5$ km$^2$, such that the open ocean represents 85% of the Weddell study region).*

Line 172: The abstract says 95%, but here it says 99%.

*Thank you for picking up on this, the value should have been 99%. Although, the main text and abstract now reflect the updated values (93-96%) following calculation of uncertainties.*

L173: mention missing data bias

*Following the comments in the 'general' section, we calculated an estimate of NPP had the areas with data unavailable experienced the mean rates of NPP. These results have been included after the first paragraph of this results section:*

> *Total annual NPP integrated over the entire Weddell Gyre between 2003 and 2020 averaged (± standard deviation) 172±34 Tg C a$^{-1}$ before gap-filling and 269±39 Tg C a$^{-1}$ after gap-filling (adjusting for the missed IFA; see Section. \ref{sec:uncertainty}). Annual area-normalised production was on average 97±8 g C m$^{-2}$ a$^{-1}$. While the open ocean experiences lower daily rates of productivity*

*compared to the shelf region (376±33 mg C m$^{-2}$ d$^{-1}$ compared to 582±99 mg C m$^{-2}$ d$^{-1}$; Figure 3.a), annual NPP is in fact higher per unit area in the open ocean than in the shelf region (97±8 mg C m$^{-2}$ a$^{-1}$, 68±23 mg C m$^{-2}$ a$^{-1}$ respectively; Fig. 3.b). This is due to a longer mean visible ice-free season: The sea-ice product shows that areas in the outer North-East edge of the open ocean are at the outer extent of the SIZ and can be ice-free for entire years, while on average, the whole open ocean region is ice-free for 139±13 days per year. The longest any of the shelf region is ice-free is 157 days, while the mean is 37±13 days.*

*The open ocean also has a far greater area than the shelf region (50.32 x 10$^5$ km$^2$ compared to 8.81 x 10$^5$ km$^2$, such that the open ocean represents 85% of the Weddell study region). As a result, when integrated over time and area, the open ocean accounts for a significant majority of the total carbon taken up by phytoplankton in the Weddell Gyre and dominates the inter-annual variability of NPP seen in the region (Fig. 3.b and c). Before imputation, the total annual NPP in the open ocean is 170±33 Tg C compared to 2±2 Tg C in the shelf region (such that the open ocean accounts for 99±1\% of the total NPP in the Weddell Gyre). After imputation, annual NPP rises to 255±38 Tg C a$^{-1}$ in the open ocean and 11±5 Tg C a$^{-1}$ in the shelf region. Despite seeing a large increase in shelf estimates following the use of the gap-filling approach, the open ocean still accounts for 96%±2% of the imputed Weddell NPP (ranging between 93-96% depending on the NPP model chosen).*

Lines 186-188: Consider discussing some of the relevant forcings that drive interannual variability of NPP? This entire subsection is very descriptive, and you don't really discuss any of the mechanisms at play. I realize that you go into depth on the drivers in the Discussion section, but at least a sentence or two mentioning some of the controls on NPP might help the reader.
*We have added the following to provide context for the reader and point to the discussion:*

> *Potential causes of variability are multiple, including ice-free area, ice-free days, timing of ice retreat, cloudiness, wind speed and direction, sea surface temperature, vertical nutrient supply, glacial contribution. We investigate a number of these in the discussion below.*

Lines 190-191: As I said above, some discussion of the mechanisms feels absent. Why might NPP on the shelf be declining? Speculation is fine, but I think some mention of the underlying dynamics is helpful. Otherwise the reader is left wondering whether this trend is just due to aliasing associated with the limitations of the satellite data on the shelf.
*Thank you for the request for more investigation and speculation on this trend. Following this and comments from the other reviewer, we had a closer examination of the spatial coverage of the satellite NPP products and found that there is a significant decline in the spatial coverage of the CAFE NPP product compared to the IFA reported. This implies a degradation in the CAFE data. The trend is not reflected in the Chl-a data used in the CAFE model, and so we can assume that it is a decline in the coverage of the absorption and backscatter variables (as elaborated on in the uncertainties section). Despite this, we continue to use the results from the*

*CAFE model since it is the most comprehensive open-source model available and has been shown to best match the Southern Ocean in Silsbe et al., 2016. However, to support the robustness of our main conclusions, we repeat several components of our analysis with other NPP products (VGPM, VGPM-Eppley and CbPM; added to the supplementary figures) to support the CAFE results.*

*The results paragraph describing the trends in shelf NPP has been adapted to include interpretation of the gap-filled data and other NPP models (below). A weak, but still statistically significant trend is seen in all gap-filled shelf NPP data, but this trend is not seen when the first year/data point is removed.*

> *In contrast, a trend in NPP is seen in the shelf region. In the CAFE model, imputed NPP declined by 3% per year, p=0.02 (Fig. 3.b). A similar rate of decline, although less statistically significant, is seen in the other NPP models. The directly-observed CAFE estimates of NPP decreased more rapidly (average decrease of 7% per year, p=0.001), underscoring the large influence of missing NPP data in the shelf region. Westberry et al.,2023 describe other potential causes for trends seen in NPP products (e.g. physiological changes in phytoplankton and decoupling of Chl-a and NPP), and emphasise the difficulty in identifying trends in NPP data and inferring drivers of trends. The trends seen here are sensitive to the occurrence of extremes in the early part of the time-period when there was a collapse of the Larsen B ice shelf along the Antarctic Peninsula (Peck et al., 2010). No trend is seen in the shelf NPP when the first data point (year 2003) is removed.*

Lines 203-204: Is this the yearly maximum IFA over the entire region or over the sub-regions separately? Because the area of the open ocean is so much larger than the area of the shelf, so the yearly maximum IFA over the entire region will be dominated by changes in the open ocean. In other words, if you're considering the yearly maximum IFA over the whole region, this could lead to a smaller correlation with the NPP on the shelf (compared to if you used the yearly maximum IFA on the shelf). It just seems strange to me that sea ice would be less important on the shelf.

*Thank you for your request for clarity, we have amended the sentence to clarify that it is the Shelf IFA vs Shelf NPP and Open ocean IFA vs Open ocean NPP. It now reads as:*

> *In the Weddell Gyre and open ocean sub-region, 42% of the inter-annual variability in total annual NPP can be explained by variability in the summer maximum IFA in each region (p=0.002, Fig. 5.a and b). This relationship was strongest in the shelf region, with 55% of the variability in total NPP being explained by the yearly maximum IFA over the shelf (p<0.001, Fig. 5.c).*

Lines 229-232: I think more could be done to introduce the objectives of the float data analysis so that it feels better integrated with the rest of the paper.

*First paragraph in the "Aligning satellite and subsurface perspectives" section has been changed to:*

> *Satellite observations indicate that, in the open ocean, the strong positive correlation between visible days and NPP degrades after around 130 visible days, indicating that other processes (e.g. grazing, nutrient availability) potentially begin to limit NPP after waters have been ice-free for more than 4 months. However, as described in Section 2.4, the ocean-color satellite loses coverage in late summer, when the solar angle decreases below 20°. As a result, it is uncertain whether further NPP is occurring, and therefore, missed after this point. Assessment of float Chl-a and POC (as proxies for phytoplankton growth/biomass) can reduce this uncertainty by indicating whether phytoplankton are still present in the surface ocean and/or whether growth may still be occurring beyond the date when satellites lose visual coverage. Therefore we seek to address if significant growth is missed after loss of satellite coverage in late summer and whether the same relationship between ice-free days and phytoplankton growth is seen in the available float observations. Although these data come from drifting platforms, rather than fixed points, we can enquire how the seasonal cycles of Chl-a and POC unfold in each year, and specifically how they evolve relative to light availability. It is worth noting that these data all represent open ocean conditions as floats are not deployed in regions shallower than 2000m.*

Lines 236-237: Where does this definition of bloom end come from?
*Hague & Vichi, 2021 used the time derivative/rate of change in Chl-a within their definition of growth initiation. Our definition was informed by this prior use of time derivatives to define bloom dynamics. We noted the plateauing nature of the cumulative increase in Chl-a, and aimed to define the timing of that plateau. Notably, the values of Chl-a observed by the floats varied considerably, and it was not possible to use an absolute value (such as 1 mg m$^{-3}$) as a threshold from blooms and bloom termination. The "bloom-end" term used here is a subjective definition in order to quantify the slowing/decline in growth/increase in Chl-a.*

Lines 284-285: Larger areas of ice-free water also provide more space for satellites to detect NPP. As I said in my general comment, I think you should use the float data as evidence that there is not significant NPP occurring underneath the ice, so that you can rule out the possibility that the correlation between ice-free area and NPP is not simply due to the greater number of pixels with non-zero NPP since satellite can't see through the ice.
*The discussion has been condensed and reworked to improve clarity and flow. The following has be written in response to this comment:*

> *The float data showing that 2-23% of integrated Chl-a (and 7-30% of surface POC; Table. 1) is present potentially before sea-ice retreat suggests that our the satellite analysis may over-estimate somewhat the correlation between IFA and NPP. Recent studies (Bisson and Cael, 2021; Hague and Vichi, 2021; McClish and Bushinsky, 2023) have also reported the presence of considerable amounts of Chl-a under sea ice as well as highlighting the onset of growth prior to*

*complete sea-ice retreat (Hague and Vichi, 2021; McClish 375 and Bushinsky, 2023). However, while our float observations also indicate that biomass tends to increase before complete ice retreat, our results still clearly show IFA as a major productivity driver. Strong phytoplankton growth follows ice melt (Fig. 7) and the majority of phytoplankton biomass is found in ice-free conditions (Figs. 2, A1 and Table 1). Similarly, in McClish and Bushinsky (2023), the break-up of sea ice initiates the increase in Chl-a and POC, highlighting the light limiting control of sea ice on phytoplankton growth.*

Line 286: I know you cite it later on, but some discussion of Moreau et al. (2023) seems warranted in this paragraph.
*A citation for Moreau et al., 2023 was added to the sentence:*
> *These hotspots are thought to be set by comparatively high levels of nutrient supply (e.g. Vernet et al., 2019; Geibert et al., 2010; Arrigo et al., 2015; Moreau et al., 2023).*

*Later in the paragraph, we have added a sentence to supplement discussion on the physical drivers of variability in sea ice-cover:*
> *Moreau et al. (2023) found that strong winds transport sea ice towards the shelf, potentially removing light limitation to the surface waters as a result.*

Line 297: add "in review" for this reference and also link to the preprint in the References section at the end of the paper.
*This paper has now been accepted, so citation and reference have been updated.*

Lines 335-337: Can you elaborate on how float data show differences in type/composition?
*In their review of the occurrence of SO sub-surface Chl-a maxima, Baldry et al., 2020 describe the dominance of diatoms in sub-surface Chl-a maxima and a shift/difference in phytoplankton community between the surface and sub-surface. We have carried out an assessment of float POC. While Chl-a and POC (and Chl:POC) do vary with depth, we acknowledge that we cannot say for sure whether these changes are due to community composition differences, or physiological changes due to photo-acclimation, or nutrient limitation. We have removed the statement and added a reference to Baldry et al., 2020 in relation to surface and subsurface differences.*

Lines 378-380: I don't understand this statement? Why would a region becoming permanently ice-free cause NPP to decrease? Are you suggesting that the sea ice is an important source of iron to the system? Or that freshwater fluxes associated with sea ice melt/refreeze are important in setting the stratification that favors growth? Give some possible mechanism because "by analogy to the permanently open ocean regions in the present-day Southern Ocean" is not very convincing since it's not clear what regions you're even referring to. There are many sources of variability besides just ice vs. no ice that lead to heterogeneity in NPP.
*We acknowledge the weak statement and have added some sentences later to highlight the potential impact of warming on stratification and nutrient mixing in the future. This now reads:*

*In future warming conditions, increased stratification, combined with freshening from melting ice could act to cut off biological productivity by reducing the vertical nutrient supply (Bronselaer et al., 2020). This will be particularly apparent in the open ocean, given its greater distance from terrestrial micro-nutrient sources. Noh et al., 2023 recently showed that, within CMIP models, Chl-a in the Arctic declines as a result of reduced nutrient supply when regions become ice-free. Despite being based in the Arctic, and thus differing physically and ecologically from the SO, this result in Noh et al., 2023 could point to a less productive Weddell Gyre in the future, should any of it become permanently ice-free.*

Lines 390-405: I found the Conclusions section to be a bit weak and I suggest rewriting. Some of the statements are well-known from existing literature (e.g. it is clear that sea-ice dynamics are important in driving NPP in this region), while other statements are speculative and don't stem from the actual analysis conducted (e.g. substantial spatial variability undoubtedly contributes to the variance in NPP…). As a result, the reader is left feeling uncertain about what contribution has been made by this study
*This sentiment was shared by both reviewers. As such we have amended the conclusions section to better highlight the findings, implications of this study and suggest avenues for future research. It now reads:*

*This study used a complement of satellite-derived sea ice and NPP products as well as BGC-Argo float observations of Chl-a and POC as proxies for phytoplankton biomass to assess the basin-scale relationship between sea ice and phytoplankton growth. We find that sea ice is the primary control on Weddell Gyre NPP in areas that experience fewer than 70-130 ice-free days per year. Beyond ~ 130 ice-free days, float Chl-a and POC observations suggest that nutrients (likely iron) emerge as an important limit to growth, possibly co-limiting with top-down grazing control. We find that while the shelf region sustains higher instantaneous NPP during its ice-free window, the open ocean sustains 93-96\% of the annual NPP of the Weddell Gyre, due to its larger area and longer ice-free season. Furthermore, while sea ice is a primary driver of inter-annual variability in total annual NPP in the Weddell Gyre, nearly half of NPP variability is still unexplained, motivating further study. We found no long-term trends in the Weddell Gyre sea-ice extent or NPP during the study period. However, our results suggest that NPP will increase if sea-ice extent decreases in the future, at least until the Weddell Gyre is ice-free for longer than 130 days, at which point, controls other than sea ice may dominate. Finally, this work has highlighted the importance of using BGC-Argo float data to complement and corroborate satellite data analysis. The study highlights the need for development of quantitative float-based NPP measurements in the region, which would likely benefit from inclusion of PAR sensors on more floats.*

**References:**

Akhoudas, C. H., Sallée, J. B., Haumann, F. A., Meredith, M. P., Garabato, A. N., Reverdin, G., Jullion, L., Aloisi, G., Benetti, M., Leng, M. J., and Arrowsmith, C.: Ventilation of the abyss in the Atlantic sector of the Southern Ocean, Scientific reports, 11, https://doi.org/10.1038/S41598-021-86043-2, 2021

Armitage, T.W.K., R. Kwok, A.F. Thompson, & G. Cunningham (2018). Dynamic topography and sea level anomalies of the Southern Ocean: Variability and teleconnections. Journal of Geophysical Research: Oceans, 123, 613-630

Baldry, K., Strutton, P. G., Hill, N. A., and Boyd, P. W.: Subsurface Chlorophyll-a Maxima in the Southern Ocean, Frontiers in Marine Science, 7, https://doi.org/10.3389/fmars.2020.00671, 2020

Brown, P. J., Meredith, M. P., Jullion, L., Garabato, A. N., Torres-Valdés, S., Holland, P., Leng, M. J., and Venables, H.: Freshwater fluxes in the Weddell Gyre: results from $\delta18O$, Philosophical transactions. Series A, Mathematical, physical, and engineering sciences, 372, https://doi.org/10.1098/RSTA.2013.0298, 2014.

Brown, P. J., Jullion, L., Landschützer, P., Bakker, D. C., Naveira Garabato, A. C., Meredith, M. P., Torres-Valdés, S., Watson, A. J., Hoppema, M., Loose, B., Jones, E. M., Telszewski, M., Jones, S. D., and Wanninkhof, R.: Carbon dynamics of the Weddell Gyre, Southern Ocean, Global Biogeochemical Cycles, 29, 288–306, https://doi.org/10.1002/2014GB005006, 2015

Hague, M. and Vichi, M.: Southern Ocean Biogeochemical Argo detect under-ice phytoplankton growth before sea ice retreat, Biogeosciences, 18, 25–38, https://doi.org/10.5194/bg-18-25-2021, 2021

Jullion, L., Garabato, A. C., Bacon, S., Meredith, M. P., Brown, P. J., Torres-Valdés, S., Speer, K. G., Holland, P. R., Dong, J., Bakker, D., Hoppema, M., Loose, B., Venables, H. J., Jenkins, W. J., Messias, M. J., and Fahrbach, E.: The contribution of the Weddell Gyre to the lower limb of the Global Overturning Circulation, Journal of Geophysical Research: Oceans, 119, 3357–3377, https://doi.org/10.1002/2013JC009725, 2014

MacGilchrist, G. A.,Naveira Garabato, A. C., Brown, P. J., Jullion, L., Bacon, S., Bakker, D. C., Hoppema, M., Meredith, M. P., and Torres-Valdés, S.: Reframing the carbon cycle of the subpolar Southern Ocean, Science Advances, 5, eaav6410, https://doi.org/10.1126/sciadv.aav6410, 2019